# One year of aerosol refractive index measurement from a coastal Antarctic site

**Zsófia Jurányi**[1,a] **and Rolf Weller**[1]

[1]Alfred-Wegener-Institut Helmholtz Zentrum für Polar- und Meeresforschung, Bremerhaven, Germany
[a]now at: Institute for the Protection of Maritime Infrastructures, German Aerospace Center (DLR), Bremerhaven, Germany

**Correspondence:** Zsófia Jurányi (zsofia.juranyi@gmail.com)

**Abstract.**

Though the environmental conditions of the Weddell Sea region and Dronning Maud Land are still relatively stable compared to the fast-changing Antarctic Peninsula, we may suspect pronounced effects of global climate change for the near future (Thompson et al., 2011). Reducing the uncertainties in climate change modeling requires a better understanding of the aerosol optical properties, and for this we need accurate data on the aerosol refractive index (RI). Due to the remoteness of Antarctica only very few RI data are available from this region (Hogan et al., 1979; Virkkula et al., 2006; Shepherd et al., 2018). We calculate the real refractive index of natural atmospheric aerosols from number size distribution measurements at the German coastal Antarctic station Neumayer III. Given the high average scattering albedo of 0.992 (Weller et al., 2013), we assumed that the imaginary part of the RI is zero. Our method uses the overlapping size range (particle diameter D between 120 and 340 nm) of a scanning mobility particle sizer (SMPS), which sizes the particles by their electrical mobility, and a laser aerosol spectrometer (LAS), which sizes the particles by their optical scattering signal at 633nm wavelength.

Based on almost a complete year of measurement, the average effective refractive index (RI$_{\text{eff}}$, as we call our retrieved RI because of the used assumptions) for the dry aerosol particles turned out to be 1.44 with a standard deviation of 0.08, in a good agreement with the RI value of 1.47, which we derived from the chemical composition of bulk aerosol sampling measurements. At Neumayer the aerosol shows a pronounced seasonal pattern in both, number concentration and chemical composition. Despite this, the variability of the monthly averaged RI$_{\text{eff}}$ values remained between 1.40 and 1.50. Compared to the annual mean, two austral winter months (July and September) showed slightly but significantly increased values (1.50 and 1.47, respectively). The size dependency of the RI$_{\text{eff}}$ could be determined from time averaged LAS and SMPS number size distributions measured between December 2017 and January 2018. Here we calculated RI$_{\text{eff}}$ for four different particle size ranges and observed a slight decrease from 1.47 (D range 116–168 nm) to 1.37 (D range 346–478 nm).

We find no significant dependence of the derived RI$_{\text{eff}}$ values on the wind direction. Thus we conclude that RI$_{\text{eff}}$ is largely independent on the general weather situation, roughly classified in (i) advection of marine boundary layer air masses during easterly winds caused by passing cyclones in contrast to (ii) air mass transport from continental Antarctica under southern katabatic winds. Neumayer, the only relevant contamination source, is located 1.5 km north of the air chemistry observatory, where the measurements were performed. Given that northerly winds are almost absent, the potential impact of local contamination is minimized in general. Indeed our data show no impact of local contamination on RI$_{\text{eff}}$. Just in one case a temporary high contamination episode with diesel engines operating right next to the measurement site resulted in an unusual high RI$_{\text{eff}}$ of 1.59, probably caused by the high black carbon content of the exhaust fumes.

To conclude, our study revealed largely constant RI$_{\text{eff}}$ values throughout the year without any sign of seasonality. Therefore, it seems reasonable to use a single, constant RI$_{\text{eff}}$ value of 1.44 for modeling optical properties of natural, coastal Antarctic sub-µm aerosol.

## 1 Introduction

Atmospheric aerosols affect the radiative balance of planet Earth (e.g. Ramanathan et al., 2001): Directly by absorbing and scattering the sunlight (e.g. Schwartz, 1996) and indirectly through modifying the micro-physical properties of the clouds (e.g. Lohmann and Feichter, 2005). The current state of the scientific knowledge on the total (direct and indirect) aerosol effect is still considered low due to the complexity of these effects (IPCC, 2014).

The refractive index (RI) of the atmospheric aerosols is a key parameter calculating their absorption and scattering and therefore essential for the global modeling of the aerosol's radiative effects. Valenzuela et al. (2018) showed that there is still clearly a need for additional and accurate measurements of the RI. There are more existing optical software packages for the optical properties of the atmospheric particulate matter and these packages extensively use RI values of the different kind of aerosols. The OPAC (Optical Properties of Aerosols and Clouds, Hess et al., 1998) package is based on laboratory measurements, whereas the HITRAN-RI (HIgh-resolution TRANsmission Refractive Indices, Massie and Hervig, 2013) package uses both laboratory and field measurements for the different included components and allows comparisons between the products using the different RIs as well.

A common method to determine the RI of aerosol particles is an indirect method: The measurement of the absorption and/or scattering of the particles along with the knowledge of the particle's size. The absorption and the scattering of a single particle is determined by the particle's size, shape and RI. It is most often assumed that particles are spherical and for the theoretical calculations the Mie theory can be used.

Wex et al. (2009) determined the RI of secondary organic aerosol by selecting the particle size using a differential mobility analyser (DMA) and measuring the scattering signal using an optical particle counter (OPC). The same method was used by Hand and Kreidenweis (2002) on ambient aerosol. Additionally they combined the measurements from an aerodynamic particle sizer as well, in order to gain information on the particles' density. Bukowiecki et al. (2011); Zhang et al. (2013); Zieger et al. (2015) used the number size distribution with parallel nephelometer and aethalometer measurements to determine the RI of ambient aerosols. A very similar method was used by Virkkula et al. (2006) for the Antarctic site Aboa, assuming that here the imaginary part of the RI can be neglected.

Barkey et al. (2007) measured laboratory generated particles' number size distribution and light scattering by a polar nephelometer. They introduced an inversion algorithm to obtain the RI. A new and more exotic method is to use optical trapping combined with Mie spectroscopy to capture the RI of atmospheric aerosol samples in the 460–700 nm wavelength range by Shepherd et al. (2018). Cavity ring-down spectroscopy is a method to study the light extinction by aerosol particles. This method was used by Bluvshtein et al. (2012) who introduced an RI retrieval method by measuring the light extinction at two carefully selected size parameters. We have to keep in mind that all above mentioned methods are not direct measurements of the RI. All of these methods search for RI values that provide good agreement in a closure study between different measured quantities.

As we see there are plenty of existing aerosol RI measurements, but the majority of these measurements are based on laboratory generated particles and only few on ambient aerosols. And if we look for RI measurements from Antarctica we can only find very few available data. Hogan et al. (1979) collected aerosol particles at the South Pole in a size range between 0.3 and 12 μm during a 4-days period and put oils with known different RIs on them until they could not see the particles in the microscope (i.e. until the applied oil's RI matched the RI of the collected particles). They have found an RI of 1.54 for these samples. Virkkula et al. (2006) derived the RI (assuming a zero imaginary part) of the ambient aerosol at coastal Antarctica during a 12-days summer campaign and got values around 1.43–1.44. Insoluble organic aerosol collected at the Clean Air Sector Laboratory of the British Antarctic Survey station Halley was analysed by Shepherd et al. (2018). They obtained a RI of 1.47 for samples collected on 60 consecutive days during austral summer 2015.

In this paper we would like to present continuous data on the real RI at 633 nm wavelength of the dry ambient aerosol as derived from measurements of an optical particle counter and a scanning mobility particle sizer. To our knowledge this is the first time, that such long-term RI measurements of almost one year from Antarctica is presented. With this, our study aims at better understanding of the aerosol optical properties at a place where only very few such data are available with special focus on its temporal variability. Given the distinct seasonality of the aerosol composition (see Weller et al., 2008, Figs. 4 and 5 therein), we may likewise expect a seasonality of RI. To this end, continuous year-round data of RI are necessary, in particular regarding the lack of such measurements for the Antarctic realm.

## 2 Method

### 2.1 Sampling Site

The measurements presented in this paper were performed in the Air Chemistry Observatory (SPUSO from "Spurenstof-fobservatorium") of the German Antarctic station of Neumayer III between February 2017 and January 2018. The SPUSO is situated at the coast of Antarctica on the Ekström shelf-ice close to Atka Bay. This observatory is a global site of the WHO's Global Atmosphere Watch programme (World Meteorological Organisation, 2016). Detailed description of the site and of the prevailing meteorological conditions were

already presented elsewhere (Wagenbach et al., 1988; Weller et al., 2008), here we only give a brief introduction to it.

The SPUSO lies 1.5 km south from the Neumayer III station and was built on the shelf-ice which moves approximately 120 m every year to the north. The edge of the shelf-ice and with this the sea is 7-to-21 km to the north. Due to the remoteness of the measurement site, anthropogenic pollution can barely reach it and the main aerosol source is the Southern Ocean. During the austral summer the sea next to the shelf-ice edge and in the close Atka bay is ice free, whereas during the long Antarctic winter the next open water can be as far as 100 km. Towards the inside of the continent, apart from some remote nunataks there is no ice-free surface.

The only possible contamination source is the Neumayer station itself, where most of the energy is provided by diesel engines. This is the reason why the SPUSO was built 1.5 km to the south of the station in a clean air sector and its power supply is provided through a cable from the main station. At this measurement site, northerly winds are almost never present and therefore most of the time we can have a contamination free sampling. The Neumayer station is completely isolated and not accessible during the winter season which lasts 9 months.

## 2.2 Experimental Setup

The aerosol is continuously sampled through our inlet system, which has its air intake approximately 8 m above the snow surface. The inlet has an aerodynamic cut-off diameter of 7–10 μm at windspeeds of 4–10 m s$^{-1}$ (Weller et al., 2008). Due to the heated measurement container and the low ambient temperatures, aerosol entering the measurement container is dry (relative humidity, RH≪30 %, most of the time even RH<10 %) without any additional drying. The inlet system is made of electropolished stainless steel, the individual instruments are connected to the inlet via stainless steel or/and conductive silicon tubing. The meteorological data used in this study (temperature, wind direction and speed and ambient RH) was measured directly on the roof of SPUSO.

The particle number size distribution was measured with two commercial instruments. A scanning mobility particle sizer (SMPS) consisting of an electrostatic classifier (TSI 3080) and a condensational particle counter (CPC, TSI 3776) measured in the 16–960 nm particle mobility diameter range. The SMPS was operated with 2.2 L min$^{-1}$ sheath flow and 0.3 L min$^{-1}$ sample flow. The other instrument was a laser aerosol spectrometer (LAS, TSI 3340) which detects and sizes the particles by measuring the intensity of their scattered light as they pass by the 633 nm Helium-Neon active cavity laser. The optical design and the high laser intensity enables the detection of single particles down to 90 nm diameter. The sample flow of the LAS was set to 0.05 L min$^{-1}$, the sheath flow was 0.65 L min$^{-1}$. The instrument measured in the size range of 90–5000 nm and was factory cali-

brated by Polysterene Latex (PSL) particles. Both the SMPS and LAS measured with a 10-minutes time resolution, however the LAS and the SMPS detects different particles at a time. The LAS counts all the particles which pass the laser beam whereas the SMPS performed two scans within the 10-minutes time period and is only able to detect one particle size at a time, dependent on the voltage that is currently set in the instrument. Therefore if the aerosol changes significantly within 10 minutes, differences can exist between the measurements of the two instruments as well.

The particle number concentration was measured by a commercial CPC (CPC, TSI 3775) with a one-minute time resolution. A Multi-Angle Absorption Photometer (MAAP, Thermo Scientific TM Model 5012) operating at a wavelength of 637 nm (Petzold and Schönlinner, 2004) was used to measure the aerosol absorption during the measurement campaign. The absorption values were converted into equivalent black carbon (eBC, Petzold et al., 2013) mass concentration using a mass absorption efficiency of 6.6 m$^2$g$^{-1}$, and were registered also once per minute. The ionic composition of the aerosol was measured by a low volume Teflon/Nylon filter system, and the filters are analysed by ion chromatography. The filters were changed daily but not every day at the same time and therefore the time resolution of the ionic composition varies with time. The average sampling flow was ≈3.5 m$^3$h$^{-1}$, the sampled air volume varied between 30 m$^3$ and 125 m$^3$ in 2017. The filter sampling is automatically switched off in case of a possible contamination (snow drift, northerly wind direction, wind velocities below 2 m s$^{-1}$ or above 20 m s$^{-1}$, and exceedingly high particle number concentrations), see details in Weller et al. (2008). In this study we used the following main ionic species: NH$_4^+$, Na$^+$ NO$_3^-$, non sea-salt SO$_4^{2-}$ and MSA$^-$ (methanesulphonate). The CPC and the MAAP are part of the continuous measurement program of GAW.

## 2.3 Correction of the LAS losses

We have collected data from both the LAS and SMPS instruments for almost one year (09.02.2017–20.01.2018). Unfortunately, during most of this time, the LAS was positioned horizontally too far away (ca. 3 m) from the inlet such that significant amount of particles were lost in the connecting tube. This problem was first discovered in November 2017. Right after, on the 23.11.2017, the instrument was repositioned right below the inlet in order to minimize the particle losses. For this study, we were particularly interested in the particle diameter range between 120 and 340 nm because we used the number size distribution data in this diameter range for the RI determination (see section 2.6). Therefore, it was important to check whether or not we are able to correct for the particle losses before November 2017 in this diameter range.

Measuring the losses in the sampling line which was used before November 2017 ("old" setup) was a challenging task.

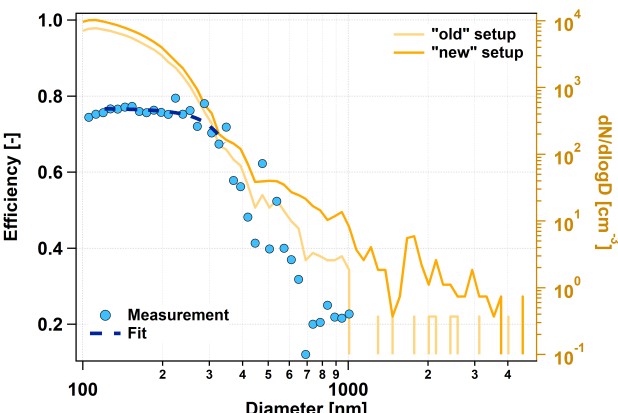

**Figure 1.** The quantification of the LAS losses in the sampling line. The two orange lines refer to the right axis and show the average room air number size distributions. "Old" setup: time average with the long horizontal tube, "new" setup: time average without the horizontal tube. The blue dots show the particle transmission efficiency through the tube, the dashed dark blue line shows a polynomial fit in the diameter range which was used for the RI calculation.

At our measurement site, no particle generator was available to perform tests with, and due to the location and isolation of the station, it was also impossible to receive any equipment for the test. Our best option was to use the room air of the measurement container to quantify the particle losses. The room air aerosol was measured by disconnecting the tubing from the inlet and sucking air from inside the measurement container. The room air provided only a low concentration, so that several hours of measurement were needed. One measurement cycle included the number size distribution measurement of the LAS of the room air aerosol in the "old" setup and right after removing the horizontal tube in the "new" setup, with the shorter, vertical tube. To make sure, that the aerosol source is stable enough during one cycle, the number size distribution measurement time was reduced to 2 times $60\,\mathrm{s}$ with some seconds in between to change between the setups.

All measured number size distributions were averaged separately for the "old" and the "new" setups, and the average number size distributions were compared. Figure 1 shows the results of this comparison. If one looks at them (Fig. 1, orange lines, right axis) or at the particle transmission efficiency (the ratio between the two size distributions, Fig. 1 blue dots, left axis) it is obvious that the losses in the "old" sampling line are significant. Almost all particles with diameters above $1\,\mu\mathrm{m}$ were lost, and therefore it is impossible to make any correction there. For this reason, the number size distribution up to $5\,\mu\mathrm{m}$ is only available after November 2017. In the diameter range of the RI determination of $120$–$340\,\mathrm{nm}$, the efficiency is between 0.77 and 0.67. The losses are significant here as well, but we consider this still as correctable. To have a continuous correction factor, the

transmission efficiency (Fig. 1, blue dots) was fit within the diameter range of interest with a polynomial line. The blue dashed line shows this polynomial fit which was used for the correction.

## 2.4    Time averaging

Due to the low aerosol number concentration in Antarctica, we performed a base time averaging of one hour of all measured data. This one hour averaging still often resulted in too noisy number size distributions, such that an RI fit was impossible. The particle number concentration at our measurement site has a strong seasonal variability with much lower concentrations in winter than in summer. This strong seasonal variability is the reason why in summer a much shorter time averaging period is sufficient to enable a successful RI fit. To keep the highest possible time resolution of the derived RI, we have chosen the length of the time averaging to be time dependent. And this length was determined by the actual particle concentration.

After performing many tests, we found, that the one hour averaged SMPS number size distributions, recorded during a time period with an average number concentration of at least $400\,\mathrm{cm}^{-3}$, showed an adequate signal to noise ratio for the RI calculation and no further averaging was needed. For all other cases with lower concentrations the hourly averaged data was further averaged until the number of particles detected by the SMPS equaled or exceeded the particle number, which is detected during a one hour SMPS scan at $400\,\mathrm{cm}^{-3}$ particle concentration. In some extreme cases in winter, the measured data had to be averaged for 15 hours, whereas in summer most of the time the original one hour or sometimes 2-hours averaging time was needed. Due to this averaging method we have the highest possible time resolution though not constant, but changing with time, depending on the total particle number concentration. This changing time resolution had to be taken into account for all further time average or statistical calculations.

## 2.5    Recalculation of the LAS number size distribution

The LAS is factory calibrated using PSL particles having an RI of 1.588 (Eidhammer et al., 2008). In order to be able to recalculate the particle number size distribution for any other RI, we need to calculate the theoretical instrument response (TIR, the signal which the instrument measures) of the LAS for both PSL particles ($\mathrm{TIR_{PSL}}$) and for particles with the specified RI ($\mathrm{TIR_{RI}}$) as function of the particle diameter. This was done by a custom-written Mie code using the LAS wavelength of $\lambda = 633\,\mathrm{nm}$ and a detection angle $\Theta$ between 22 and 158 degrees with a geometry of a round detector shape.

The LAS delivers the number size distribution ($n(D)$) as the particle number concentration ($N(D)$) sorted into diameter bins: $n(D_i) = \frac{dN(D_i)}{d\log(D_i)}$, where i denotes the i$^{th}$ diameter

bin. These bins cover the whole measurement range of the instrument leaving no gaps. Each diameter bin has a lower and a higher boundary ($D_{i,\text{lower}}$, $D_{i,\text{higher}}$). These bin boundaries correspond to the PSL calibration of the LAS. In order to recalculate the number size distribution to another RI, all bin boundary diameters have to be recalculated. This recalculation can be done by using the previously calculated TIR values: (1) For a single, PSL calibration based bin diameter ($D_{i,\text{PSL}}$) the instrument response $\text{TIR}_{\text{PSL}}(D_{i,\text{PSL}})$ is looked up. (2) Now we look at the TIR values that are calculated using the desired RI. We search at which diameter ($D_{i,\text{RI}}$) we get the same instrument response as for PSL: $\text{TIR}_{\text{RI}}(D_{i,\text{RI}}) = \text{TIR}_{\text{PSL}}(D_{i,\text{PSL}})$ and that diameter is the recalculated bin boundary diameter. We repeat this for every diameter bin.

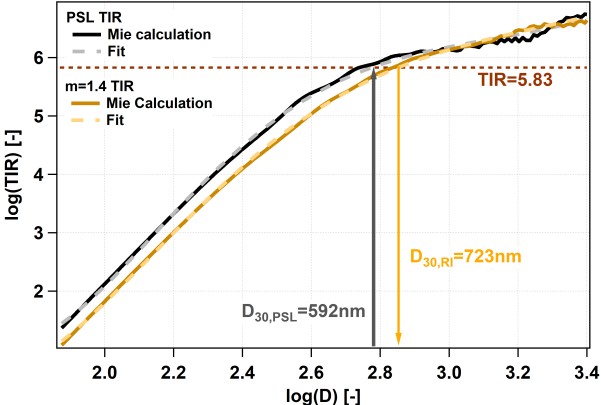

**Figure 2.** LAS Theoretical instrument responses for $m = 1.588 + 0i$ (black) and 1.40+0i (orange) as function of the particle diameter. Here we show an example, how an original LAS diameter bin border ($D_{30,\text{PSL}}$) was recalculated to the target RI ($D_{30,\text{RI}}$).

The diameter recalculation is not always straight-forward, because OPCs using a monochromatic laser often suffer from a non-monotonic instrument response at higher diameters (e.g., Hodkinson and Greenfield, 1965; Barnard and Harrison, 1988). This problem of non-monotonic instrument response was solved by smoothing the calculated instrumental response function by fitting a 5[th] grade polynomial to the logarithm of both $\text{TIR}_{\text{PSL}}$ and $\text{TIR}_{\text{RI}}$ functions. Figure 2 shows an example how a single bin boundary diameter ($D_{30,\text{PSL}}$, the 30[th] diameter bin border) is recalculated using another RI ($m = 1.4 + 0i$). The Mie calculation (solid line) and the polynomial fit (dashed line) are shown for both RIs. The 30[th] diameter bin border is 592 nm in our setup, using the original PSL calibration. One can read from Figure 2 that a PSL particle of this size detected by the LAS results in the same TIR as a particle with the RI of 1.4 and the size of $D_{30,\text{RI}} = 723$ nm. The same procedure has to be used for every bin boundary diameter and every desired index of refraction. After having the recalculated diameter borders, we can recalculate the number size distribution as well. If the original number size distribution is:

$$n_{\text{PSL}}(D_{\text{PSL}}) = \frac{dN(D_{\text{PSL}})}{d\log(D_{\text{PSL}})} \tag{1}$$

Then the recalculated number size distribution looks like this:

$$n_{\text{RI}}(D_{\text{RI}}) = \frac{dN(D_{\text{RI}})}{d\log(D_{\text{RI}})} = \frac{dN(D_{\text{RI}})}{\log(D_{\text{high,RI}}) - \log(D_{\text{low,RI}})} \tag{2}$$

where $D_{\text{high,RI}}$ is the upper and $D_{\text{low,RI}}$ is the lower boundary of the recalculated diameter bin.

## 2.6 Calculation of the effective refractive index

In order to find the aerosol refractive index, the SMPS and the LAS data in the overlapping size range has to be matched. This matching was done by recalculating the LAS number size distribution using a set of different RIs and finding the one which matches the best the SMPS number size distribution at the overlapping size range. The following expression was used after Khlystov et al. (2004) to quantise the difference between the LAS and the SMPS distribution:

$$\chi(m) = \frac{1}{N} \cdot \sum_{i=N_{\text{min}}}^{N_{\text{max}}} \left[ \log\left( n_{\text{SMPS}}\left( D_i \right) \right) - \log\left( n_{\text{LAS}}\left( m, D_i \right) \right) \right]^2 \tag{3}$$

The SMPS and the LAS has an overlapping size range between 90 and 950 nm, however only the range between 120 and 340 nm was used for the fit. The SMPS number size distribution was too noisy above 340 nm and at the lowest diameters, the LAS does not have a detection efficiency of unity. The range of the RI was chosen to be 1.3–1.8 with 0.01 steps in between. The imaginary part of the RI was kept at 0 which is an acceptable assumption considering that the absorption is very low compared to the scattering at our measurement site, with an average single scattering albedo of 0.992 (Weller et al., 2013). The $\chi(m)$ function was determined for every single $m$ value, and the minimum of this function was searched. That $m$ where $\chi$ reaches its minimum is the $m$ value we look for and we interpret as the RI of the measured aerosol. Those cases were omitted where the $\chi$ function did not have an explicit minimum or exceeded a limit. After manual inspection of many fit procedures this limit was set to the value of 0.02. Such cases might occur if too much noise is present in the data or if the size distribution was varying too much during the time period of one scan. Next to this numerical criterion every single scan was manually checked as well.

The RI derived with our method is representative for the particle diameter range of 120–340 nm, which was used for the RI calculation. If we can assume that all particles in the number size distribution have the same RI, our calculated RI is the true RI. If the chemical composition of the aerosol is changing with the particle size, it is possible that the RI is also size dependent. Hence, our derived RI might differ from the average RI which corresponds to the complete aerosol population. In addition we assumed a spherical shape of the particles and a negligible imaginary part of the RI. Therefore we term our derived RI the effective refractive index ($RI_{eff}$) from now on, and for later conclusions we have to keep in mind that the ($RI_{eff}$) might not be the true RI of an individual particle.

## 3 Results and discussion

### 3.1 Verification of the LAS correction

In order to verify the used LAS correction (see Sec. 2.3), measurement of particles with known RI and spherical shape was necessary. The lack of any particle generator left us with not many possibilities. A commercial e-cigarette (Joytech eGo) was available at the station, and we used this to generate particles for the testing purpose. E-cigarette liquid contains glycerin, propylene glycol, water, nicotine and flavourings and the formed aerosol particles are spherical liquid droplets. Pratte et al. (2016) measured the RI of many e-cigarettes of different types and got values between 1.429 and 1.436, and therefore we assume that our generated test particles had an RI of 1.43.

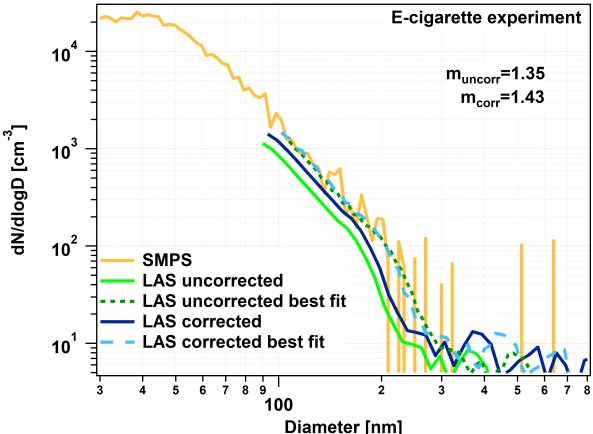

**Figure 3.** The E-cigarette experiment, showing the validation of our LAS correction. The orange line shows the measured SMPS number size distribution, the green lines the uncorrected LAS number size distribution (light: original, dark and dashed: best fit with $m_{uncorr}$ calculated RI) and the blue lines (dark: original, light and dashed: best fit with $m_{corr}$ calculated RI) are the losses corrected LAS number size distributions.

We filled a plastic bag of ≈100 L volume with particle free air, then added 2–3 puffs of the e-cigarette smoke using a small, hand-operated air pump. After that, we let the aerosol particles coagulate in the bag for 10–15 minutes in order to let the particles reach the detection diameter range of the LAS. The e-cigarette test was done with the same setup as the "old" measurement setup using the long vertical tube.

We used the method introduced in the sections 2.5 and 2.6 to calculate the RI of this e-cigarette smoke, first with the uncorrected LAS data then with applying the above introduced (Sec. 2.3) LAS correction. These values can be compared to the e-cigarette smoke's literature RI value of 1.43 to check whether the LAS correction works well or not. For this fit we have chosen a slightly different particle size range of 110–220 nm because the form of the number size distribution was different from the ambient one.

Figure 3 shows the results of the e-cigarette experiment. Without using the LAS correction on the LAS data (green lines) we get an RI of 1.35 from the best fit. This value is significantly lower than the literature RI value of 1.43 suggesting that the LAS losses had a high influence on the retrieved RI and that a correction is necessary. Using the losses corrected LAS size distribution, the best fit between the SMPS and the LAS data (blue lines) resulted in the RI of 1.43 which agrees with the literature value. This verifies our LAS correction, and we applied it on all LAS data before November 2017.

### 3.2 Sensitivity of the RI calculation on the number size distribution measurement

The accuracy of our $RI_{eff}$ calculation mainly depends on the measured input data's uncertainty, which is the uncertainty of the number size distribution measurements in our case. Here, we discuss the sensitivity of the derived $RI_{eff}$ values introduced by the measurement uncertainty. An intercomparison between many mobility particle size spectrometers showed that all of the different investigated instruments measured within an uncertainty range of ±10% (Wiedensohler et al., 2012). We use this value for our SMPS, and assume that the LAS has the same uncertainty as well.

In order to investigate the effect of this measurement uncertainty we take the worst case scenarios, by either adding 10% to the particle number concentration measured by the SMPS and subtract 10% from the LAS, or the other way around. We calculated for one month measurement period the $RI_{eff}$ values using these modified number size distributions next to the original ones. Choosing 10% higher SMPS concentration and 10% lower LAS concentration resulted in lower calculated $RI_{eff}$. On average the values were 0.045 lower compared to the original values which translates into an average 3.1% error. The other scenario results in artificially high values, which turned out to be on average 0.050 and this means an error of 3.5%. This shows that even assuming the worst case scenario would cause an acceptable

error, and most probably we can expect a lower uncertainty in reality.

### 3.3   RI calculation examples

Figure 4 shows four examples of the RI fitting procedure's performance in different cases. The first example (Fig. 4a) is from the summer season when the number concentration was high enough that a one-hour averaging period was reasonable. The orange line shows the measured SMPS scan, whereas the dark blue line shows the simultaneously measured LAS number size distribution with the factory calibration. The dark blue line lies below the SMPS line which indicates that the built-in calibration RI of 1.588 overestimates the prevailing RI. The fitting procedure verifies this and the best fit belongs to the recalculated LAS scan with the RI of 1.45 which we consider as the effective refractive index, $RI_{eff}$, of the dry aerosol at that time.

Figure 4b shows a similar situation from winter with much lower particle concentrations and therefore a longer averaging time of 11 hours. The obtained RI was quite low: 1.37 in this case. An uncommon example can be seen in figure 4c when the number size distribution was trimodal. The fit was successful again, the retrieved RI is 1.48. As the last example (Fig. 4d), we show a case where the fit was unsuccessful, we could not retrieve a valid RI. The fitting procedure returned a best fit, but the value of $\chi$ exceeded 0.02 and it is also clearly visible that this best solution does not fit very well the measured SMPS number size distribution. The reason why the fit did not work in this case was that the aerosol population was significantly changing within the duration of the SMPS scan. During the first half of the scan an aerosol plume with very high concentration reached the instruments. This appears in the SMPS scan as a very high fraction of small particles, because the SMPS selected and measured the smaller particles during the first half of the scan. Contrary, the LAS captures all particles with different diameters at the same time, and therefore this event appears as an elevated overall concentration. This was an extreme and exceptional situation where some unavoidable construction work was done around the SPUSO using machines powered by diesel engines.

### 3.4   Seasonal variability and mean value of the refractive index

We have collected data during almost a complete year (from 09.02.2017 to 20.01.2018), giving us the unique possibility to calculate the long-term $RI_{eff}$ and to analyze its seasonal variability. Figure 5 shows this seasonal variability, where some statistical values of the monthly $RI_{eff}$ are presented. The gray circles show the monthly mean values with the standard deviation (Stdev) as error bars, the black sticks the medians and the gray sticks the 25th and 75th percentiles. The orange bar chart belongs to the right axis and indicates the number of the $RI_{eff}$ values that could be retrieved for the corresponding month. The same data is also shown in Table 1 complemented with the yearly mean values.

The mean $RI_{eff}$ during our complete measurement period was 1.44 with a comparable median of 1.41. As already mentioned, there are only very few other RI measurements from Antarctica. Virkkula et al. (2006) calculated the RI values from number size distribution and scattering coefficient measurements at the Finnish Antarctic summer station Aboa. Aboa is situated approximately 300 km to the west of the Neumayer station and lies a little further away from the sea. These measurements were performed in the summer of 2000 during a 12-day period. They found a mean RI of 1.454 at $\lambda = 550$ nm and 1.460 at $\lambda = 700$ nm wavelength excluding a nucleation event where unrealistically low values (lower than the RI of water) were derived. Our average RI values have a very good agreement with their average RI values, and this agreement is even better considering only our mean $RI_{eff}$ value from January (1.45).

Concerning the monthly averages, it is interesting, that in spite of the existing strong seasonal variability of both the aerosol concentration (Jaenicke et al., 1992; Weller et al., 2011) and chemical composition (Wagenbach et al., 1988) the RI does not or only slightly show a comparable behaviour: The monthly averages of $RI_{eff}$ remain quite constant and remain within the range of 1.40–1.50. There are two winter months with higher RIs: July with a mean of 1.50 and September with 1.47. These values are significantly different from the yearly mean (determined by using a statistical T-test with a significance level of 0.01). In both cases we have only relatively few data-points due to extremely low particle concentrations and therefore we can only speculate on the reason for the slightly higher values. In winter the fraction of sea salt is higher than in summer and sea salt has a slightly higher RI than the other salts present in the aerosol phase (see Sec. 3.5).

The monthly $RI_{eff}$ distributions are quite narrow. However, due to the needed long averaging time between 1 and 20 hours, a potential higher short-term variability may not be represented. Although the Stdev of $RI_{eff}$ comprising the whole measurement period is 0.08, we observed a statistically significant seasonality in the monthly data. The winter months (June to September) seem to have a higher scatter (Fig. 5 gray sticks) and higher Stdev values (0.11 in July vs. 0.03 in January, Fig. 5 error bars). We found a similar tendency in the chemical composition with higher variability during the austral winter compared to summer. This might be one reason for the higher scatter in the $RI_{eff}$ values, apart from probably higher uncertainty of the fitting method due to extremely low wintertime particle number concentrations.

### 3.5   Link to the chemical composition

The aerosol chemical composition shows a strong seasonal variation at our measurement site. The dominant aerosol component is sea-salt with around 50 % of the total mass in summer and 86 % in winter (Weller et al., 2008). While

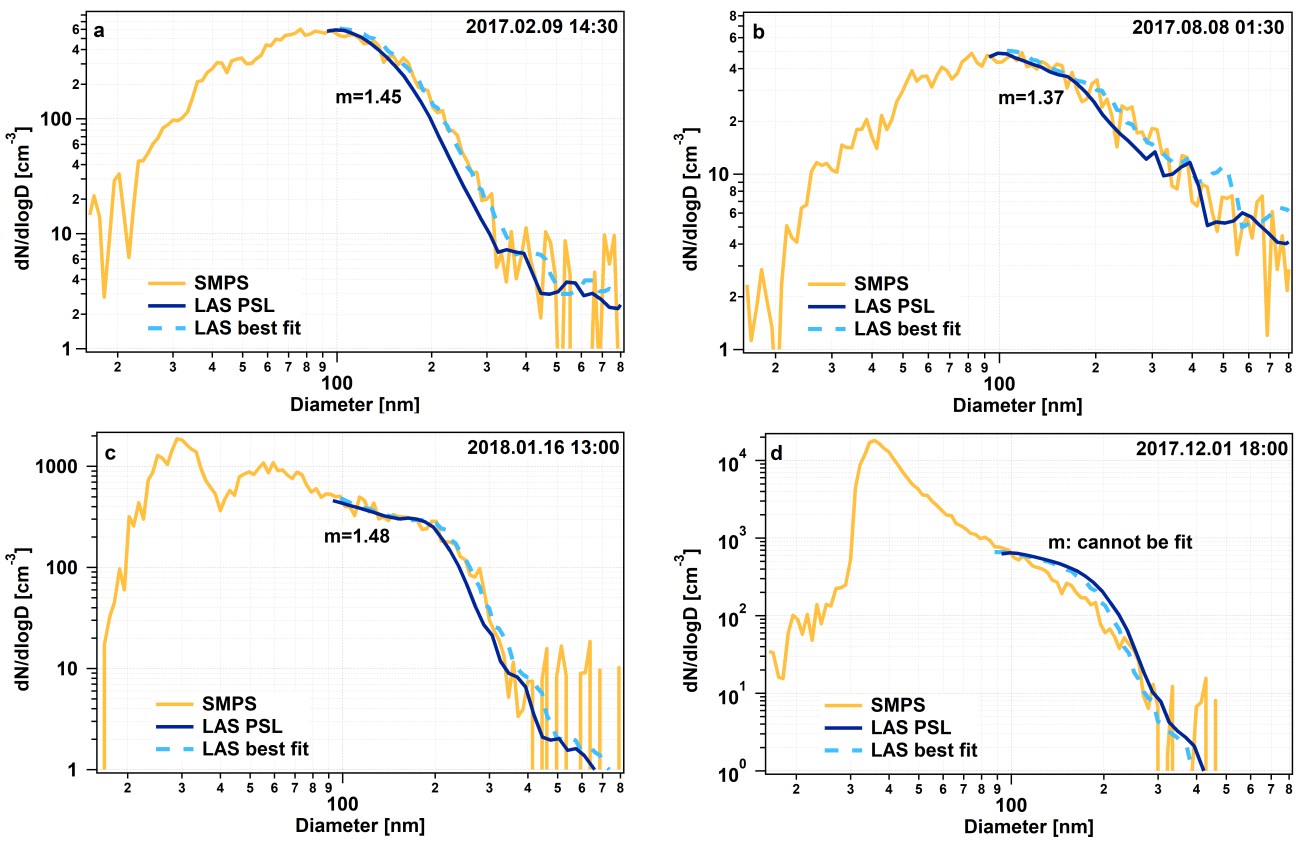

**Figure 4.** 4 examples on the refractive index fit performance. The orange line shows the measured SMPS number size distribution, whereas the blue lines (dark: PSL calibrated, light and dashed: best fit) show the LAS number size distributions.

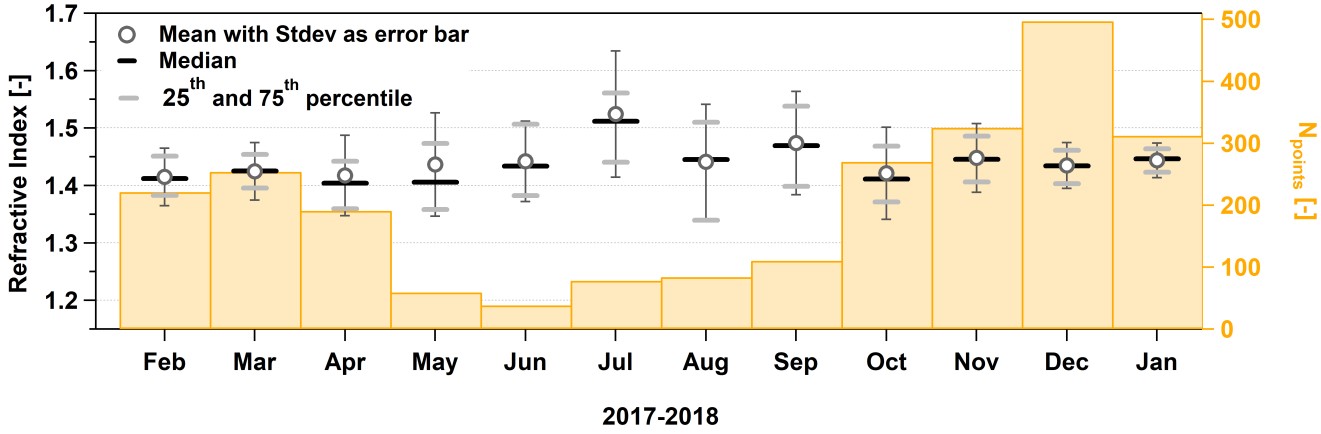

**Figure 5.** The monthly averages (with error bars as the standard deviation), medians and percentiles of $RI_{eff}$ from the coastal Antarctica, measured at $\lambda$=633 nm for dry aerosol particles. The orange bars refer to the right axis and show the number of successful RI retrievals in the corresponding month.

negligible during winter, biogenic sulphur aerosol reaches its annual maximum in austral summer between January and March (Minikin et al., 1998). At our investigated wavelength of 633 nm, sea-salt has an RI of 1.49 (Shettle and Fenn, 1979), sulfuric acid 1.42 (Palmer and Williams, 1975), ammonium sulphate 1.53 (Toon et al., 1976), ammonium bisulphate 1.47 (Chylek and Wong, 1995), sodium nitrate 1.46 (Cotterell et al., 2017), ammonium nitrate 1.52 (Toon et al.,

**Table 1.** The monthly and yearly ($\sum$) averages, standard deviations (Stdev), medians and percentiles of the $RI_{eff}$ from coastal Antarctica, measured at $\lambda$=633 nm for dry aerosol particles.

| Month | 25$^{th}$ percentile | Median | 75$^{th}$ percentile | Mean | Stdev | $N_{points}$ |
|---|---|---|---|---|---|---|
| Feb | 1.38 | 1.41 | 1.45 | 1.41 | 0.05 | 221 |
| Mar | 1.40 | 1.43 | 1.45 | 1.42 | 0.05 | 254 |
| Apr | 1.36 | 1.40 | 1.44 | 1.41 | 0.07 | 191 |
| May | 1.36 | 1.40 | 1.47 | 1.42 | 0.09 | 59 |
| Jun | 1.38 | 1.43 | 1.51 | 1.44 | 0.07 | 38 |
| Jul | 1.44 | 1.51 | 1.56 | 1.50 | 0.11 | 78 |
| Aug | 1.34 | 1.45 | 1.51 | 1.44 | 0.10 | 84 |
| Sep | 1.40 | 1.47 | 1.54 | 1.47 | 0.09 | 110 |
| Oct | 1.37 | 1.41 | 1.47 | 1.42 | 0.08 | 270 |
| Nov | 1.41 | 1.45 | 1.49 | 1.45 | 0.06 | 325 |
| Dec | 1.40 | 1.43 | 1.46 | 1.44 | 0.04 | 497 |
| Jan | 1.42 | 1.45 | 1.46 | 1.44 | 0.03 | 312 |
| $\sum$ | 1.37 | 1.41 | 1.46 | 1.44 | 0.08 | 2439 |

1976), MSA 1.43 (Virkkula et al., 2006) and black carbon 1.75+0.43i (Hess et al., 1998).

The chemical composition was determined from the daily filter measurements of the ionic composition and from the eBC measurement of the MAAP. The mass concentration of the dominant component of sea salt was calculated from the $Na^+$ ion. It was assumed that $NH_4^+$ is preferentially present as ammonium sulphate (($NH_4$)$_2SO_4$) and/or ammonium bisulphate ($NH_4HSO_4$) salt due to the high nss-$SO_4^{2-}$/$NH_4^+$ ratio of around 11.2±8 (annual mean ± Stdev). In addition, formation of ammonium nitrate ($NH_4NO_3$) has to be considered. Part of the nitrate can also be bound as $NaNO_3$. The remaining $SO_4^{2-}$ was assumed to be present as sulfuric acid.

We do not have any information on the organic carbon mass fraction for our measurement period, and therefore we could not include this component into the calculation. However, previous water soluble organic carbon (WSOC) mass concentration measurements (Weller et al., 2015) showed, that in the austral summer of 2011 the WSOC average mass fraction was less than 3% and therefore we believe that organic carbon does not have a significant influence on the resulting RI. Using this chemical composition and assuming that the aerosol is homogeneously and internally mixed, the RI can be calculated from the volume fraction and the RI of the individual components. The imaginary part of the RI was again neglected, which is a justified assumption, because the volume fraction of the eBC never exceeded 0.1 % in 2017. This amount of eBC would add at most a $\approx 4 \cdot 10^{-3}$i imaginary value to the RI.

The average RI calculated from the chemical composition in 2017 becomes 1.47, and is in a good agreement with the optically retrieved $RI_{eff}$ of 1.44. The reason for the slight discrepancy might be caused by the used assumptions. In addition and in contrast to the bulk chemical composition, the RI

calculation derived from the SMPS and OPC data are based on a limited size range between 120 and 340 nm. As discussed later in Section 3.7, RI changes slightly with the particle size.

Finally, we calculated RI separately for summer (November to February) and winter (March to October) from the aerosol chemical composition. We found higher RI values of 1.48 during austral winter compared to 1.45 during summer. This may be caused by the much higher sea salt aerosol portion during winter with the highest RI among the ionic compounds. Note also the significantly higher $RI_{eff}$ values for the winter months July and September (Fig. 5).

### 3.6 Impact of general weather situation and local contamination

Neumayer station is situated 1.5 km north of the measurement site, thus contamination during northerly winds, but also when the wind speeds are very low, has to be considered. We start with examining whether the actual wind direction influences our data in general, followed by a case study when diesel engines were operated right next to the measurement site. Contamination is mainly associated with high concentrations of black carbon. Black carbon has an RI of 1.75+0.43i (Hess et al., 1998) which is considerably higher than of any other natural chemical components of the aerosol. Note also the distinct imaginary part of the RI.

The prevailing wind direction at the SPUSO is east, associated with high wind speeds above 10 ms$^{-1}$, frequently exceeding even 20 ms$^{-1}$. Easterly wind directions, especially if they are accompanied with high wind speeds, are characteristic for the impact of passing cyclones and marine air entry. The second frequent wind direction is south, with wind speeds generally below 10 ms$^{-1}$. This weather situation is characteristic for advection of more continental

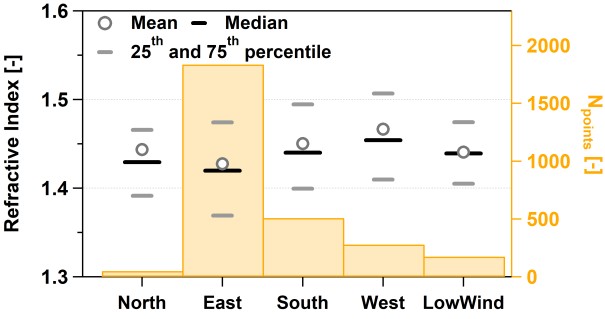

**Figure 6.** The averages, medians and percentiles of the $RI_{eff}$ from the coastal Antarctica separated by the wind direction, measured at $\lambda$=633 nm for dry aerosol particles. The orange bars refer to the right axis and show the number of successful RI retrievals.

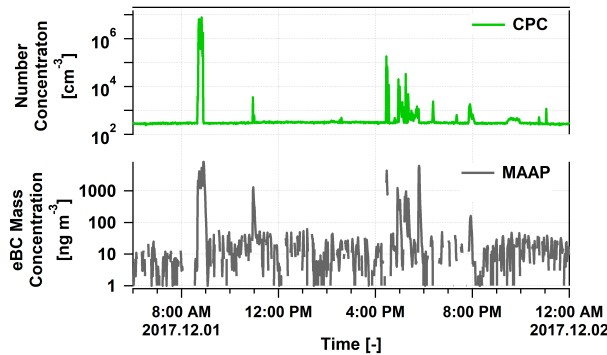

**Figure 7.** The particle number concentration (green) and the equivalent black carbon mass concentration (black) measured on 01.12.2017

air masses by katabatic winds. Westerly winds are usually caused by low-pressure systems in the southern Weddell region and associated with moderate winds speeds between 10 and 20 ms$^{-1}$. Northerly winds are virtually absent (König-
Langlo et al., 1998) and if present, mark a period of potential contamination from the station. We have separated the $RI_{eff}$ data according to different wind direction sectors to examine whether different air masses are associated with particles showing different RI. To this end, we defined the wind di-
rection sector between 315° and 45° as north, 45° and 135° as east, 135° and 225° as south and 225° and 315° as west. We categorized all data associated with wind speeds below 2 ms$^{-1}$ separately (LowWind in Figure 6).

Overall, our measurement period was representative and
meaningful for each individual sector, even for the inherently few data related to northerly wind directions. Figure 6 shows the $RI_{eff}$ values, sorted according to the mentioned categories. The gray circles show the time averages, the black sticks the medians, the gray sticks the 25$^{th}$ and 5$^{th}$ per-
centiles. In summary, no significant dependency of $RI_{eff}$ on the wind direction or wind speed is observable. We conclude that the general weather situation, just as local contamination, has no impact on $RI_{eff}$. Even adverse wind condition associated with potential contamination from the ex-
haust fumes of the main station did not cause any significant change of $RI_{eff}$.

In order to further investigate the problem of the contamination we performed a case study on a time period when planned contamination reached the SPUSO. This was the
same construction event which was already shown in Fig. 4d as an example for an unsuccessful fit when the aerosol was changing too fast. On the day of 01.12.2017, diesel engine powered machines were in operation in the very close vicinity of the measurement site.
Figure 7 shows the particle number concentration (green) and the black carbon mass concentration (black) as measured

by the CPC and MAAP, respectively during this construction episode. The highest concentrations were present during the morning and the late afternoon even exceeding $6 \cdot 10^6$ cm$^{-3}$ and $8\,\mu g\,m^{-3}$ which are 3–4 orders of magnitude higher  40 than the values without contamination (Weller et al., 2011, 2013). Unfortunately, these concentrations changed very fast, depending on whether the engine emissions were directly reaching our inlet, and therefore most of the time, we were not able to perform a fit for the RI. We have only one single  45 scan when the concentration was stable enough and elevated, allowing us to assume that we determined $RI_{eff}$ for a contaminated situation.

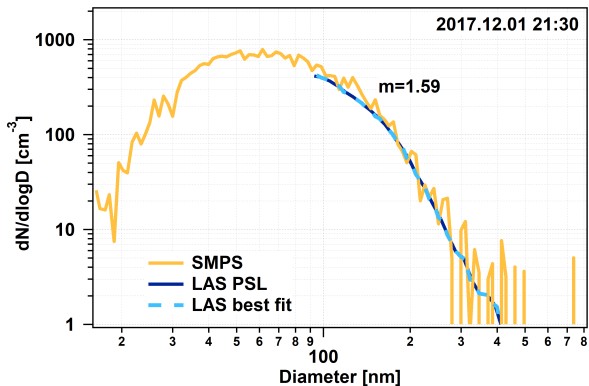

**Figure 8.** A successful RI fit from 01.12.2017 with high contamination present

Figure 8 shows this fit with the retrieved RI of 1.59. One can see that the original LAS scan fits already very well,  50 which means that the RI of the factory calibration of PSLs give us a good solution. This retrieved $RI_{eff}$ is significantly higher than the values we normally got. We can assume that the increased black carbon concentration caused this effect, and increased RI values might be an indicator for strong con-  55 tamination at this site. This time period, and any other time

period with known contamination was removed from the statistical calculations.

### 3.7 Size dependent contribution to the scattering

In the following we will calculate the contribution of the particles with different sizes to the scattering coefficient. Unfortunately, the LAS data was not usable above 600 nm during the time period when the particle losses were high, and therefore we can only do these calculations for an almost 2-months long summer period (01.12.2017-20.01.2018) when the LAS was installed right below the aerosol inlet. It was assumed that the derived $RI_{eff}$ is valid along the complete number size distribution (between 16 nm and 5000 nm) and that the particles are spherical and thus Mie calculation can be used for the determination of the single particle scattering at the wavelength of 633 nm. The scattering coefficient size distribution of the dry aerosol was calculated as follows:

$$\frac{d\sigma_s(D)}{d\log D} = C_s(D, \lambda, m) \cdot \frac{dN(D)}{d\log D} \tag{4}$$

where $\sigma_s$ is the scattering coefficient in m$^{-1}$, $m$ is the derived, time dependent $RI_{eff}$ without a unit and $C_s$ is the scattering cross section of the individual particles in m$^2$. To calculate $C_s$ we used our custom written Mie code.

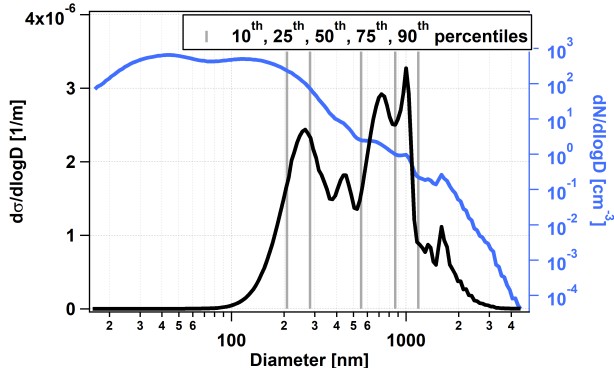

**Figure 9.** The average dry scattering coefficient size distribution (black line) at 633 nm wavelength and the corresponding particle number size distribution (blue line, right axis) as function of the particle diameter. The grey lines show the 10$^{th}$, 25$^{th}$, 50$^{th}$, 75$^{th}$ and 90$^{th}$ of the scattering coefficent distribution.

Figure 9 shows the time average of $d\sigma_s(D)/d\log D$ as function of the particle diameter. Next to it, the average number size distribution (blue line, right logarithmic axis) for the same time period is also shown. As we can see, particles smaller than 100 nm or larger than 3 μm do not contribute significantly to the scattering. 80% of the total scattering amount come from the size range between 208 and 1170 nm. Interestingly, the distribution is multimodal, having two main peaks around 260 and 860 nm. The median of the distribution

is at 550 nm which is much higher than the median of the number size distribution (64 nm), as expected, because scattering increases faster than linearly as function of the particle diameter. The average number size distribution is also multimodal with two distinct peaks around 40 nm and 140 nm. Considering the time evolution and not temporal averages we see, that these two peaks, as well as the two main peaks of the scattering coefficient size distribution, are often present simultaneously. In conclusion, the bimodality is not the product of time averaging of single modes appearing at different times.

Finally we investigate the effect of neglecting the imaginary part of the RI on the scattering coefficient. As we have seen in Section 3.5 including the eBC in the chemical composition adds at most an imaginary part of $\approx 4 \cdot 10^{-3}$i to the RI. We recalculated the average scattering coefficient size distribution adding this imaginary part to the RI. This gives us a highest possible estimate on the error we make if we would neglect the imaginary part of RI. It turns out that the relative difference of the scattering coefficient size distribution considering $4 \cdot 10^{-3}i$ RI instead of 0.0i never exceeds 1.7 % irrespective of the particle diameter.

### 3.8 Size dependence of the refractive index

To examine the dependence of $RI_{eff}$ on the given particle size distribution, we again have to restrict the time period to (01.12.2017-20.01.2018) when the LAS's particle losses were minimised. During this period we have an SMPS–LAS overlapping size range between 120 and 900 nm. If we calculate the temporal average over this complete time period, most of the noise is averaged out as well, so that we can use most of this overlapping size range for the RI fit. Moreover, the overall size distribution range can now be divided into 4 subranges suitable for separate $RI_{eff}$ calculations, representative for the corresponding subrange (Fig. 10).

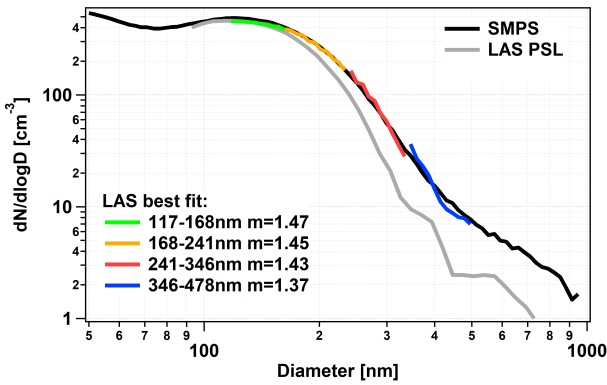

**Figure 10.** The average dry aerosol number size distribution measurements during December 2017 and January 2018 as measured by the SMPS (black line) and the LAS (gray line). The coloured lines show the 4 individual RI fits using 4 different particle size ranges.

Figure 10 shows the time averaged LAS (gray line) and SMPS (black line) number size distributions. We have chosen the following particle size ranges for the separate RI fit: 117–168 nm, 168–241 nm, 241–346 nm and 376–478 nm ensuring, that we have a similar number of size distribution measurement points for the fit procedure in each of the size ranges.

With the increasing particle size, we needed to apply a lower RI in order to have the best match between the LAS and the SMPS. In the first range we got an $RI_{eff}$ of 1.47, in the second 1.45, in the third 1.43 and in the fourth 1.37. According to Figure 10 the $RI_{eff}$ decreases slightly within the first 3 subranges of particle diameter ($RI_{eff}$ between 1.47 and 1.43), but more pronounced for the highest range ($RI_{eff}$ = 1.37)

The conspicuously lower $RI_{eff}$ in the highest investigated size range may originate from a significantly changing chemical composition. Interestingly, sea-salt particles should dominate this higher size range, but this would result in a higher $RI_{eff}$. Hence one may speculate about a coating of sea-salt particles in this special case (probably organic material with typically lower RI). The presence of a coating or a different aerosol source might also explain the bimodality of the scattering coefficient size distribution (Section 3.7). However, we have to keep in mind, that this is pure speculation and we have no proof of it.

## 4 Conclusions

We have calculated the real RI for dry natural aerosol at a coastal Antarctic measurement site using the overlapping size range of two instruments measuring the number size distribution in two different ways: optically and by electrical mobility. The yearly average ($\pm$Stdev) of the RI was calculated based on the data from almost a complete year and turned out to be 1.44 ($\pm$0.08). This average is in very good agreement with the RI value of 1.47 which we derived from filter based chemical composition measurements. The good agreement shows that at least for coastal Antarctica this method reliably delivers the RI values without the additional effort of a chemical characterization of the aerosol.

Based on this, we recommend this single, temporally constant refractive index value for modeling of aerosol optical properties. In this context we suggest supporting investigations to examine the validity of this approach and the usage of seasonal independent $RI_{eff}$ values for the Antarctic region.

In spite of the strong seasonal variability of the chemical composition at the measurement site (e.g. 86% sea-salt present in winter, 50% in summer), we could not identify a corresponding seasonal trend of the RI, which is in good agreement with RI derived from the chemical composition of the present aerosol. We conclude that the given high variability of the ionic composition of the aerosol typical for coastal Antarctica causes only minor variability in associated RI values. We could not find any significant influence from the wind direction either. We conclude that the general weather situation. just as local contamination, has no significant impact on $RI_{eff}$.

In forthcoming related investigations at Neumayer, a year-round optical closure experiment is planned. For this, the size range between 16 nm and 5 μm as well aerosol scattering coefficients by integrated nephelometer measurements will be employed.

*Data availability.* Data reported here are available at https://doi.pangaea.de/10.1594/PANGAEA.899429 and https://doi.pangaea.de/10.1594/PANGAEA.899430 for scientific purposes.

*Author contributions.* ZJ has performed the measurements, analysed and interpreted the data and wrote the manuscript. RW built up the measurement site, supervised the measurements and the data analysis and reviewed and edited the manuscript.

*Competing interests.* The authors declare that they have no conflict of interest.

*Acknowledgements.* The authors would like to thank all members of the 37[th] overwintering team of the Neumayer III station for their support and for being a great group. The first author of the paper would like to express her gratitude to her brother for his support during the harsh winter months in Antarctica.

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

# One year of aerosol refractive index measurement from a coastal Antarctic site

**Zsófia Jurányi**[1,a] **and Rolf Weller**[1]

[1]Alfred-Wegener-Institut Helmholtz Zentrum für Polar- und Meeresforschung, Bremerhaven, Germany
[a]now at: Institute for the Protection of Maritime Infrastructures, German Aerospace Center (DLR), Bremerhaven, Germany

**Correspondence:** Zsófia Jurányi (zsofia.juranyi@gmail.com)

**Abstract.**

Though the environmental conditions of the Weddell Sea region and Dronning Maud Land are still relatively stable compared to the fast-changing Antarctic Peninsula, we may suspect pronounced effects of global climate change for the near future (Thompson et al., 2011). Reducing the uncertainties in climate change modeling requires a better understanding of the aerosol optical properties, and for this we need accurate data on the aerosol refractive index (RI). Due to the remoteness of Antarctica only very few RI data are available from this region (Hogan et al., 1979; Virkkula et al., 2006; Shepherd et al., 2018). We calculate the real refractive index of natural atmospheric aerosols from number size distribution measurements at the German coastal Antarctic station Neumayer III. Given the high average scattering albedo of 0.992 (Weller et al., 2013), we assumed that the imaginary part of the RI is zero. Our method uses the overlapping size range (particle diameter D between 120 and 340 nm) of a scanning mobility particle sizer (SMPS), which sizes the particles by their electrical mobility, and a laser aerosol spectrometer (LAS), which sizes the particles by their optical scattering signal at 633nm wavelength.

Based on almost a complete year of measurement, the average effective refractive index ($RI_{eff}$, as we call our retrieved RI because of the used assumptions) for the dry aerosol particles turned out to be 1.44 with a standard deviation of 0.08, in a good agreement with the RI value of 1.47, which we derived from the chemical composition of bulk aerosol sampling measurements. At Neumayer the aerosol shows a pronounced seasonal pattern in both, number concentration and chemical composition. Despite this, the variability of the monthly averaged $RI_{eff}$ values remained between 1.40 and 1.50. Compared to the annual mean, two austral winter months (July and September) showed slightly but significantly increased values (1.50 and 1.47, respectively). The size dependency of the $RI_{eff}$ could be determined from time averaged LAS and SMPS number size distributions measured between December 2017 and January 2018. Here we calculated $RI_{eff}$ for four different particle size ranges and observed a slight decrease from 1.47 (D range 116–168 nm) to 1.37 (D range 346–478 nm).

We find no significant dependence of the derived $RI_{eff}$ values on the wind direction. Thus we conclude that $RI_{eff}$ is largely independent on the general weather situation, roughly classified in (i) advection of marine boundary layer air masses during easterly winds caused by passing cyclones in contrast to (ii) air mass transport from continental Antarctica under southern katabatic winds. Neumayer, the only relevant contamination source, is located 1.5 km north of the air chemistry observatory, where the measurements were performed. Given that northerly winds are almost absent, the potential impact of local contamination is minimized in general. Indeed our data show no impact of local contamination on $RI_{eff}$. Just in one case a temporary high contamination episode with diesel engines operating right next to the measurement site resulted in an unusual high $RI_{eff}$ of 1.59, probably caused by the high black carbon content of the exhaust fumes.

To conclude, our study revealed largely constant $RI_{eff}$ values throughout the year without any sign of seasonality. Therefore, it seems reasonable to use a single, constant $RI_{eff}$ value of 1.44 for modeling optical properties of natural, coastal Antarctic sub-µm aerosol.

## 1 Introduction

Atmospheric aerosols affect the radiative balance of planet Earth (e.g. Ramanathan et al., 2001): Directly by absorbing and scattering the sunlight (e.g. Schwartz, 1996) and indirectly through modifying the micro-physical properties of the clouds (e.g. Lohmann and Feichter, 2005). The current state of the scientific knowledge on the total (direct and indirect) aerosol effect is still considered low due to the complexity of these effects (IPCC, 2014).

The refractive index (RI) of the atmospheric aerosols is a key parameter calculating their absorption and scattering and therefore essential for the global modeling of the aerosol's radiative effects. Valenzuela et al. (2018) showed that there is still clearly a need for additional and accurate measurements of the RI. There are more existing optical software packages for the optical properties of the atmospheric particulate matter and these packages extensively use RI values of the different kind of aerosols. The OPAC (Optical Properties of Aerosols and Clouds, Hess et al., 1998) package is based on laboratory measurements, whereas the HITRAN-RI (HIgh-resolution TRANsmission Refractive Indices, Massie and Hervig, 2013) package uses both laboratory and field measurements for the different included components and allows comparisons between the products using the different RIs as well.

A common method to determine the RI of aerosol particles is an indirect method: The measurement of the absorption and/or scattering of the particles along with the knowledge of the particle's size. The absorption and the scattering of a single particle is determined by the particle's size, shape and RI. It is most often assumed that particles are spherical and for the theoretical calculations the Mie theory can be used.

Wex et al. (2009) determined the RI of secondary organic aerosol by selecting the particle size using a differential mobility analyser (DMA) and measuring the scattering signal using an optical particle counter (OPC). The same method was used by Hand and Kreidenweis (2002) on ambient aerosol. Additionally they combined the measurements from an aerodynamic particle sizer as well, in order to gain information on the particles' density. Bukowiecki et al. (2011); Zhang et al. (2013); Zieger et al. (2015) used the number size distribution with parallel nephelometer and aethalometer measurements to determine the RI of ambient aerosols. A very similar method was used by Virkkula et al. (2006) for the Antarctic site Aboa, assuming that here the imaginary part of the RI can be neglected.

Barkey et al. (2007) measured laboratory generated particles' number size distribution and light scattering by a polar nephelometer. They introduced an inversion algorithm to obtain the RI. A new and more exotic method is to use optical trapping combined with Mie spectroscopy to capture the RI of atmospheric aerosol samples in the 460–700 nm wavelength range by Shepherd et al. (2018). Cavity ring-down spectroscopy is a method to study the light extinction by aerosol particles. This method was used by Bluvshtein et al. (2012) who introduced an RI retrieval method by measuring the light extinction at two carefully selected size parameters. We have to keep in mind that all above mentioned methods are not direct measurements of the RI. All of these methods search for RI values that provide good agreement in a closure study between different measured quantities.

As we see there are plenty of existing aerosol RI measurements, but the majority of these measurements are based on laboratory generated particles and only few on ambient aerosols. And if we look for RI measurements from Antarctica we can only find very few available data. Hogan et al. (1979) collected aerosol particles at the South Pole in a size range between 0.3 and 12 µm during a 4-days period and put oils with known different RIs on them until they could not see the particles in the microscope (i.e. until the applied oil's RI matched the RI of the collected particles). They have found an RI of 1.54 for these samples. Virkkula et al. (2006) derived the RI (assuming a zero imaginary part) of the ambient aerosol at coastal Antarctica during a 12-days summer campaign and got values around 1.43–1.44. Insoluble organic aerosol collected at the Clean Air Sector Laboratory of the British Antarctic Survey station Halley was analysed by Shepherd et al. (2018). They obtained a RI of 1.47 for samples collected on 60 consecutive days during austral summer 2015.

In this paper we would like to present continuous data on the real RI at 633 nm wavelength of the dry ambient aerosol as derived from measurements of an optical particle counter and a scanning mobility particle sizer. To our knowledge this is the first time, that such long-term RI measurements of almost one year from Antarctica is presented. With this, our study aims at better understanding of the aerosol optical properties at a place where only very few such data are available with special focus on its temporal variability. Given the distinct seasonality of the aerosol composition (see Weller et al., 2008, Figs. 4 and 5 therein), we may likewise expect a seasonality of RI. To this end, continuous year-round data of RI are necessary, in particular regarding the lack of such measurements for the Antarctic realm.

## 2 Method

### 2.1 Sampling Site

The measurements presented in this paper were performed in the Air Chemistry Observatory (SPUSO from "Spurenstoffobservatorium") of the German Antarctic station of Neumayer III between February 2017 and January 2018. The SPUSO is situated at the coast of Antarctica on the Ekström shelf-ice close to Atka Bay. This observatory is a global site of the WHO's Global Atmosphere Watch programme (World Meteorological Organisation, 2016). Detailed description of the site and of the prevailing meteorological conditions were

already presented elsewhere (Wagenbach et al., 1988; Weller et al., 2008), here we only give a brief introduction to it.

The SPUSO lies 1.5 km south from the Neumayer III station and was built on the shelf-ice which moves approximately 120 m every year to the north. The edge of the shelf-ice and with this the sea is 7-to-21 km to the north. Due to the remoteness of the measurement site, anthropogenic pollution can barely reach it and the main aerosol source is the Southern Ocean. During the austral summer the sea next to the shelf-ice edge and in the close Atka bay is ice free, whereas during the long Antarctic winter the next open water can be as far as 100 km. Towards the inside of the continent, apart from some remote nunataks there is no ice-free surface.

The only possible contamination source is the Neumayer station itself, where most of the energy is provided by diesel engines. This is the reason why the SPUSO was built 1.5 km to the south of the station in a clean air sector and its power supply is provided through a cable from the main station. At this measurement site, northerly winds are almost never present and therefore most of the time we can have a contamination free sampling. The Neumayer station is completely isolated and not accessible during the winter season which lasts 9 months.

## 2.2 Experimental Setup

The aerosol is continuously sampled through our inlet system, which has its air intake approximately 8 m above the snow surface. The inlet has an aerodynamic cut-off diameter of 7–10 μm at windspeeds of 4–10 m s$^{-1}$ (Weller et al., 2008). Due to the heated measurement container and the low ambient temperatures, aerosol entering the measurement container is dry (relative humidity, RH≪30 %, most of the time even RH<10 %) without any additional drying. The inlet system is made of electropolished stainless steel, the individual instruments are connected to the inlet via stainless steel or/and conductive silicon tubing. The meteorological data used in this study (temperature, wind direction and speed and ambient RH) was measured directly on the roof of SPUSO.

The particle number size distribution was measured with two commercial instruments. A scanning mobility particle sizer (SMPS) consisting of an electrostatic classifier (TSI 3080) and a condensational particle counter (CPC, TSI 3776) measured in the 16–960 nm particle mobility diameter range. The SMPS was operated with 2.2 L min$^{-1}$ sheath flow and 0.3 L min$^{-1}$ sample flow. The other instrument was a laser aerosol spectrometer (LAS, TSI 3340) which detects and sizes the particles by measuring the intensity of their scattered light as they pass by the 633 nm Helium-Neon active cavity laser. The optical design and the high laser intensity enables the detection of single particles down to 90 nm diameter. The sample flow of the LAS was set to 0.05 L min$^{-1}$, the sheath flow was 0.65 L min$^{-1}$. The instrument measured in the size range of 90–5000 nm and was factory calibrated by Polysterene Latex (PSL) particles. Both the SMPS and LAS measured with a 10-minutes time resolution, however the LAS and the SMPS detects different particles at a time. The LAS counts all the particles which pass the laser beam whereas the SMPS performed two scans within the 10-minutes time period and is only able to detect one particle size at a time, dependent on the voltage that is currently set in the instrument. Therefore if the aerosol changes significantly within 10 minutes, differences can exist between the measurements of the two instruments as well.

The particle number concentration was measured by a commercial CPC (CPC, TSI 3775) with a one-minute time resolution. A Multi-Angle Absorption Photometer (MAAP, Thermo Scientific TM Model 5012) operating at a wavelength of 637 nm (Petzold and Schönlinner, 2004) was used to measure the aerosol absorption during the measurement campaign. The absorption values were converted into equivalent black carbon (eBC, Petzold et al., 2013) mass concentration using a mass absorption efficiency of 6.6 m$^2$g$^{-1}$, and were registered also once per minute. The ionic composition of the aerosol was measured by a low volume Teflon/Nylon filter system, and the filters are analysed by ion chromatography. The filters were changed daily but not every day at the same time and therefore the time resolution of the ionic composition varies with time. The average sampling flow was ≈3.5 m$^3$h$^{-1}$, the sampled air volume varied between 30 m$^3$ and 125 m$^3$ in 2017. The filter sampling is automatically switched off in case of a possible contamination (snow drift, northerly wind direction, wind velocities below 2 m s$^{-1}$ or above 20 m s$^{-1}$, and exceedingly high particle number concentrations), see details in Weller et al. (2008). In this study we used the following main ionic species: NH$_4^+$, Na$^+$ NO$_3^-$, non sea-salt SO$_4^{2-}$ and MSA$^-$ (methanesulphonate). The CPC and the MAAP are part of the continuous measurement program of GAW.

## 2.3 Correction of the LAS losses

We have collected data from both the LAS and SMPS instruments for almost one year (09.02.2017–20.01.2018). Unfortunately, during most of this time, the LAS was positioned horizontally too far away (ca. 3 m) from the inlet such that significant amount of particles were lost in the connecting tube. This problem was first discovered in November 2017. Right after, on the 23.11.2017, the instrument was repositioned right below the inlet in order to minimize the particle losses. For this study, we were particularly interested in the particle diameter range between 120 and 340 nm because we used the number size distribution data in this diameter range for the RI determination (see section 2.6). Therefore, it was important to check whether or not we are able to correct for the particle losses before November 2017 in this diameter range.

Measuring the losses in the sampling line which was used before November 2017 ("old" setup) was a challenging task.

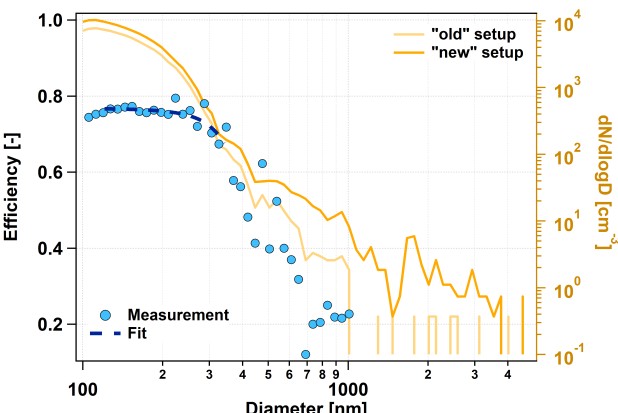

**Figure 1.** The quantification of the LAS losses in the sampling line. The two orange lines refer to the right axis and show the average room air number size distributions. "Old" setup: time average with the long horizontal tube, "new" setup: time average without the horizontal tube. The blue dots show the particle transmission efficiency through the tube, the dashed dark blue line shows a polynomial fit in the diameter range which was used for the RI calculation.

At our measurement site, no particle generator was available to perform tests with, and due to the location and isolation of the station, it was also impossible to receive any equipment for the test. Our best option was to use the room air of the measurement container to quantify the particle losses. The room air aerosol was measured by disconnecting the tubing from the inlet and sucking air from inside the measurement container. The room air provided only a low concentration, so that several hours of measurement were needed. One measurement cycle included the number size distribution measurement of the LAS of the room air aerosol in the "old" setup and right after removing the horizontal tube in the "new" setup, with the shorter, vertical tube. To make sure, that the aerosol source is stable enough during one cycle, the number size distribution measurement time was reduced to 2 times 60 s with some seconds in between to change between the setups.

All measured number size distributions were averaged separately for the "old" and the "new" setups, and the average number size distributions were compared. Figure 1 shows the results of this comparison. If one looks at them (Fig. 1, orange lines, right axis) or at the particle transmission efficiency (the ratio between the two size distributions, Fig. 1 blue dots, left axis) it is obvious that the losses in the "old" sampling line are significant. Almost all particles with diameters above 1 μm were lost, and therefore it is impossible to make any correction there. For this reason, the number size distribution up to 5 μm is only available after November 2017. In the diameter range of the RI determination of 120–340 nm, the efficiency is between 0.77 and 0.67. The losses are significant here as well, but we consider this still as correctable. To have a continuous correction factor, the

transmission efficiency (Fig. 1, blue dots) was fit within the diameter range of interest with a polynomial line. The blue dashed line shows this polynomial fit which was used for the correction.

### 2.4 Time averaging

Due to the low aerosol number concentration in Antarctica, we performed a base time averaging of one hour of all measured data. This one hour averaging still often resulted in too noisy number size distributions, such that an RI fit was impossible. The particle number concentration at our measurement site has a strong seasonal variability with much lower concentrations in winter than in summer. This strong seasonal variability is the reason why in summer a much shorter time averaging period is sufficient to enable a successful RI fit. To keep the highest possible time resolution of the derived RI, we have chosen the length of the time averaging to be time dependent. And this length was determined by the actual particle concentration.

After performing many tests, we found, that the one hour averaged SMPS number size distributions, recorded during a time period with an average number concentration of at least $400\,\mathrm{cm}^{-3}$, showed an adequate signal to noise ratio for the RI calculation and no further averaging was needed. For all other cases with lower concentrations the hourly averaged data was further averaged until the number of particles detected by the SMPS equaled or exceeded the particle number, which is detected during a one hour SMPS scan at $400\,\mathrm{cm}^{-3}$ particle concentration. In some extreme cases in winter, the measured data had to be averaged for 15 hours, whereas in summer most of the time the original one hour or sometimes 2-hours averaging time was needed. Due to this averaging method we have the highest possible time resolution though not constant, but changing with time, depending on the total particle number concentration. This changing time resolution had to be taken into account for all further time average or statistical calculations.

### 2.5 Recalculation of the LAS number size distribution

The LAS is factory calibrated using PSL particles having an RI of 1.588 (Eidhammer et al., 2008). In order to be able to recalculate the particle number size distribution for any other RI, we need to calculate the theoretical instrument response (TIR, the signal which the instrument measures) of the LAS for both PSL particles ($\mathrm{TIR}_{\mathrm{PSL}}$) and for particles with the specified RI ($\mathrm{TIR}_{\mathrm{RI}}$) as function of the particle diameter. This was done by a custom-written Mie code using the LAS wavelength of $\lambda = 633\,\mathrm{nm}$ and a detection angle $\Theta$ between 22 and 158 degrees with a geometry of a round detector shape.

The LAS delivers the number size distribution ($n(D)$) as the particle number concentration ($N(D)$) sorted into diameter bins: $n(D_i) = \frac{dN(D_i)}{d\log(D_i)}$, where i denotes the $i^{th}$ diameter

bin. These bins cover the whole measurement range of the instrument leaving no gaps. Each diameter bin has a lower and a higher boundary ($D_{i,\mathrm{lower}}$, $D_{i,\mathrm{higher}}$). These bin boundaries correspond to the PSL calibration of the LAS. In order to recalculate the number size distribution to another RI, all bin boundary diameters have to be recalculated. This recalculation can be done by using the previously calculated TIR values: (1) For a single, PSL calibration based bin diameter ($D_{i,\mathrm{PSL}}$) the instrument response $\mathrm{TIR}_{\mathrm{PSL}}(D_{i,\mathrm{PSL}})$ is looked up. (2) Now we look at the TIR values that are calculated using the desired RI. We search at which diameter ($D_{i,\mathrm{RI}}$) we get the same instrument response as for PSL: $\mathrm{TIR}_{\mathrm{RI}}(D_{i,\mathrm{RI}}) = \mathrm{TIR}_{\mathrm{PSL}}(D_{i,\mathrm{PSL}})$ and that diameter is the recalculated bin boundary diameter. We repeat this for every diameter bin.

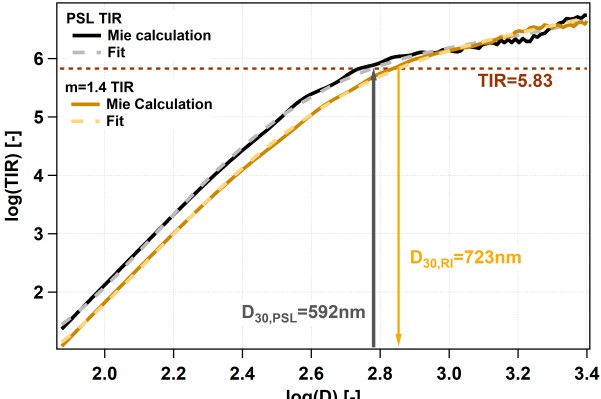

**Figure 2.** LAS Theoretical instrument responses for $m = 1.588 + 0i$ (black) and 1.40+0i (orange) as function of the particle diameter. Here we show an example, how an original LAS diameter bin border ($D_{30,\mathrm{PSL}}$) was recalculated to the target RI ($D_{30,\mathrm{RI}}$).

The diameter recalculation is not always straight-forward, because OPCs using a monochromatic laser often suffer from a non-monotonic instrument response at higher diameters (e.g., Hodkinson and Greenfield, 1965; Barnard and Harrison, 1988). This problem of non-monotonic instrument response was solved by smoothing the calculated instrumental response function by fitting a 5th grade polynomial to the logarithm of both $\mathrm{TIR}_{\mathrm{PSL}}$ and $\mathrm{TIR}_{\mathrm{RI}}$ functions. Figure 2 shows an example how a single bin boundary diameter ($D_{30,\mathrm{PSL}}$, the 30th diameter bin border) is recalculated using another RI ($m = 1.4+0i$). The Mie calculation (solid line) and the polynomial fit (dashed line) are shown for both RIs. The 30th diameter bin border is 592 nm in our setup, using the original PSL calibration. One can read from Figure 2 that a PSL particle of this size detected by the LAS results in the same TIR as a particle with the RI of 1.4 and the size of $D_{30,\mathrm{RI}} = 723$nm. The same procedure has to be used for every bin boundary diameter and every desired index of refraction. After having the recalculated diameter borders, we

can recalculate the number size distribution as well. If the original number size distribution is:

$$n_{\mathrm{PSL}}(D_{\mathrm{PSL}}) = \frac{dN(D_{\mathrm{PSL}})}{d\log(D_{\mathrm{PSL}})} \tag{1}$$

Then the recalculated number size distribution looks like this:

$$n_{\mathrm{RI}}(D_{\mathrm{RI}}) = \frac{dN(D_{\mathrm{RI}})}{d\log(D_{\mathrm{RI}})} = \frac{dN(D_{\mathrm{RI}})}{\log(D_{\mathrm{high,RI}}) - \log(D_{\mathrm{low,RI}})} \tag{2}$$

where $D_{\mathrm{high,RI}}$ is the upper and $D_{\mathrm{low,RI}}$ is the lower boundary of the recalculated diameter bin.

## 2.6 Calculation of the effective refractive index

In order to find the aerosol refractive index, the SMPS and the LAS data in the overlapping size range has to be matched. This matching was done by recalculating the LAS number size distribution using a set of different RIs and finding the one which matches the best the SMPS number size distribution at the overlapping size range. The following expression was used after Khlystov et al. (2004) to quantise the difference between the LAS and the SMPS distribution:

$$\chi(m) = \frac{1}{N} \cdot \sum_{i=N_{\min}}^{N_{\max}} \left[ \log\left(n_{\mathrm{SMPS}}(D_i)\right) - \log\left(n_{\mathrm{LAS}}(m, D_i)\right) \right]^2 \tag{3}$$

The SMPS and the LAS has an overlapping size range between 90 and 950 nm, however only the range between 120 and 340 nm was used for the fit. The SMPS number size distribution was too noisy above 340 nm and at the lowest diameters, the LAS does not have a detection efficiency of unity. The range of the RI was chosen to be 1.3–1.8 with 0.01 steps in between. The imaginary part of the RI was kept at 0 which is an acceptable assumption considering that the absorption is very low compared to the scattering at our measurement site, with an average single scattering albedo of 0.992 (Weller et al., 2013). The $\chi(m)$ function was determined for every single $m$ value, and the minimum of this function was searched. That $m$ where $\chi$ reaches its minimum is the $m$ value we look for and we interpret as the RI of the measured aerosol. Those cases were omitted where the $\chi$ function did not have an explicit minimum or exceeded a limit. After manual inspection of many fit procedures this limit was set to the value of 0.02. Such cases might occur if too much noise is present in the data or if the size distribution was varying too much during the time period of one scan. Next to this numerical criterion every single scan was manually checked as well.

The RI derived with our method is representative for the particle diameter range of 120–340 nm, which was used for the RI calculation. If we can assume that all particles in the number size distribution have the same RI, our calculated RI is the true RI. If the chemical composition of the aerosol is changing with the particle size, it is possible that the RI is also size dependent. Hence, our derived RI might differ from the average RI which corresponds to the complete aerosol population. In addition we assumed a spherical shape of the particles and a negligible imaginary part of the RI. Therefore we term our derived RI the effective refractive index ($RI_{eff}$) from now on, and for later conclusions we have to keep in mind that the ($RI_{eff}$) might not be the true RI of an individual particle.

## 3 Results and discussion

### 3.1 Verification of the LAS correction

In order to verify the used LAS correction (see Sec. 2.3), measurement of particles with known RI and spherical shape was necessary. The lack of any particle generator left us with not many possibilities. A commercial e-cigarette (Joytech eGo) was available at the station, and we used this to generate particles for the testing purpose. E-cigarette liquid contains glycerin, propylene glycol, water, nicotine and flavourings and the formed aerosol particles are spherical liquid droplets. Pratte et al. (2016) measured the RI of many e-cigarettes of different types and got values between 1.429 and 1.436, and therefore we assume that our generated test particles had an RI of 1.43.

We filled a plastic bag of ≈100 L volume with particle free air, then added 2–3 puffs of the e-cigarette smoke using a small, hand-operated air pump. After that, we let the aerosol particles coagulate in the bag for 10–15 minutes in order to let the particles reach the detection diameter range of the LAS. The e-cigarette test was done with the same setup as the "old" measurement setup using the long vertical tube.

We used the method introduced in the sections 2.5 and 2.6 to calculate the RI of this e-cigarette smoke, first with the uncorrected LAS data then with applying the above introduced (Sec. 2.3) LAS correction. These values can be compared to the e-cigarette smoke's literature RI value of 1.43 to check whether the LAS correction works well or not. For this fit we have chosen a slightly different particle size range of 110–220 nm because the form of the number size distribution was different from the ambient one.

Figure 3 shows the results of the e-cigarette experiment. Without using the LAS correction on the LAS data (green lines) we get an RI of 1.35 from the best fit. This value is significantly lower than the literature RI value of 1.43 suggesting that the LAS losses had a high influence on the retrieved RI and that a correction is necessary. Using the losses corrected LAS size distribution, the best fit between the SMPS and the LAS data (blue lines) resulted in the RI of 1.43 which agrees with the literature value. This verifies our LAS correction, and we applied it on all LAS data before November 2017.

### 3.2 Sensitivity of the RI calculation on the number size distribution measurement

The accuracy of our $RI_{eff}$ calculation mainly depends on the measured input data's uncertainty, which is the uncertainty of the number size distribution measurements in our case. Here, we discuss the sensitivity of the derived $RI_{eff}$ values introduced by the measurement uncertainty. An intercomparison between many mobility particle size spectrometers showed that all of the different investigated instruments measured within an uncertainty range of ±10% (Wiedensohler et al., 2012). We use this value for our SMPS, and assume that the LAS has the same uncertainty as well.

In order to investigate the effect of this measurement uncertainty we take the worst case scenarios, by either adding 10% to the particle number concentration measured by the SMPS and subtract 10% from the LAS, or the other way around. We calculated for one month measurement period the $RI_{eff}$ values using these modified number size distributions next to the original ones. Choosing 10% higher SMPS concentration and 10% lower LAS concentration resulted in lower calculated $RI_{eff}$. On average the values were 0.045 lower compared to the original values which translates into an average 3.1% error. The other scenario results in artificially high values, which turned out to be on average 0.050 and this means an error of 3.5%. This shows that even assuming the worst case scenario would cause an acceptable

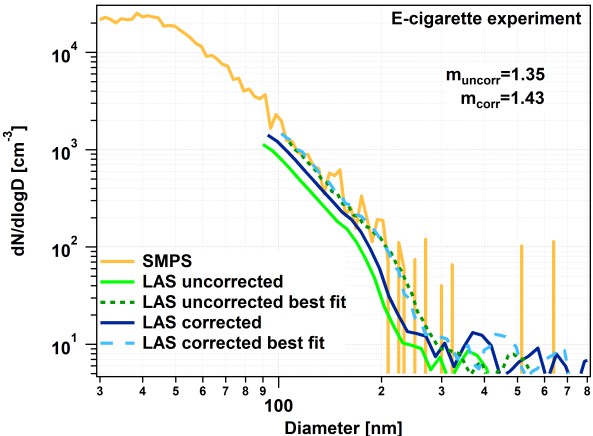

**Figure 3.** The E-cigarette experiment, showing the validation of our LAS correction. The orange line shows the measured SMPS number size distribution, the green lines the uncorrected LAS number size distribution (light: original, dark and dashed: best fit with $m_{uncorr}$ calculated RI) and the blue lines (dark: original, light and dashed: best fit with $m_{corr}$ calculated RI) are the losses corrected LAS number size distributions.

error, and most probably we can expect a lower uncertainty in reality.

### 3.3 RI calculation examples

Figure 4 shows four examples of the RI fitting procedure's performance in different cases. The first example (Fig. 4a) is from the summer season when the number concentration was high enough that a one-hour averaging period was reasonable. The orange line shows the measured SMPS scan, whereas the dark blue line shows the simultaneously measured LAS number size distribution with the factory calibration. The dark blue line lies below the SMPS line which indicates that the built-in calibration RI of 1.588 overestimates the prevailing RI. The fitting procedure verifies this and the best fit belongs to the recalculated LAS scan with the RI of 1.45 which we consider as the effective refractive index, $RI_{eff}$, of the dry aerosol at that time.

Figure 4b shows a similar situation from winter with much lower particle concentrations and therefore a longer averaging time of 11 hours. The obtained RI was quite low: 1.37 in this case. An uncommon example can be seen in figure 4c when the number size distribution was trimodal. The fit was successful again, the retrieved RI is 1.48. As the last example (Fig. 4d), we show a case where the fit was unsuccessful, we could not retrieve a valid RI. The fitting procedure returned a best fit, but the value of $\chi$ exceeded 0.02 and it is also clearly visible that this best solution does not fit very well the measured SMPS number size distribution. The reason why the fit did not work in this case was that the aerosol population was significantly changing within the duration of the SMPS scan. During the first half of the scan an aerosol plume with very high concentration reached the instruments. This appears in the SMPS scan as a very high fraction of small particles, because the SMPS selected and measured the smaller particles during the first half of the scan. Contrary, the LAS captures all particles with different diameters at the same time, and therefore this event appears as an elevated overall concentration. This was an extreme and exceptional situation where some unavoidable construction work was done around the SPUSO using machines powered by diesel engines.

### 3.4 Seasonal variability and mean value of the refractive index

We have collected data during almost a complete year (from 09.02.2017 to 20.01.2018), giving us the unique possibility to calculate the long-term $RI_{eff}$ and to analyze its seasonal variability. Figure 5 shows this seasonal variability, where some statistical values of the monthly $RI_{eff}$ are presented. The gray circles show the monthly mean values with the standard deviation (Stdev) as error bars, the black sticks the medians and the gray sticks the $25^{th}$ and $75^{th}$ percentiles. The orange bar chart belongs to the right axis and indicates the number of the $RI_{eff}$ values that could be retrieved for the corresponding month. The same data is also shown in Table 1 complemented with the yearly mean values.

The mean $RI_{eff}$ during our complete measurement period was 1.44 with a comparable median of 1.41. As already mentioned, there are only very few other RI measurements from Antarctica. Virkkula et al. (2006) calculated the RI values from number size distribution and scattering coefficient measurements at the Finnish Antarctic summer station Aboa. Aboa is situated approximately 300 km to the west of the Neumayer station and lies a little further away from the sea. These measurements were performed in the summer of 2000 during a 12-day period. They found a mean RI of 1.454 at $\lambda = 550$ nm and 1.460 at $\lambda = 700$nm wavelength excluding a nucleation event where unrealistically low values (lower than the RI of water) were derived. Our average RI values have a very good agreement with their average RI values, and this agreement is even better considering only our mean $RI_{eff}$ value from January (1.45).

Concerning the monthly averages, it is interesting, that in spite of the existing strong seasonal variability of both the aerosol concentration (Jaenicke et al., 1992; Weller et al., 2011) and chemical composition (Wagenbach et al., 1988) the RI does not or only slightly show a comparable behaviour: The monthly averages of $RI_{eff}$ remain quite constant and remain within the range of 1.40–1.50. There are two winter months with higher RIs: July with a mean of 1.50 and September with 1.47. These values are significantly different from the yearly mean (determined by using a statistical T-test with a significance level of 0.01). In both cases we have only relatively few data-points due to extremely low particle concentrations and therefore we can only speculate on the reason for the slightly higher values. In winter the fraction of sea salt is higher than in summer and sea salt has a slightly higher RI than the other salts present in the aerosol phase (see Sec. 3.5).

The monthly $RI_{eff}$ distributions are quite narrow. However, due to the needed long averaging time between 1 and 20 hours, a potential higher short-term variability may not be represented. Although the Stdev of $RI_{eff}$ comprising the whole measurement period is 0.08, we observed a statistically significant seasonality in the monthly data. The winter months (June to September) seem to have a higher scatter (Fig. 5 gray sticks) and higher Stdev values (0.11 in July vs. 0.03 in January, Fig. 5 error bars). We found a similar tendency in the chemical composition with higher variability during the austral winter compared to summer. This might be one reason for the higher scatter in the $RI_{eff}$ values, apart from probably higher uncertainty of the fitting method due to extremely low wintertime particle number concentrations.

### 3.5 Link to the chemical composition

The aerosol chemical composition shows a strong seasonal variation at our measurement site. The dominant aerosol component is sea-salt with around 50 % of the total mass in summer and 86 % in winter (Weller et al., 2008). While

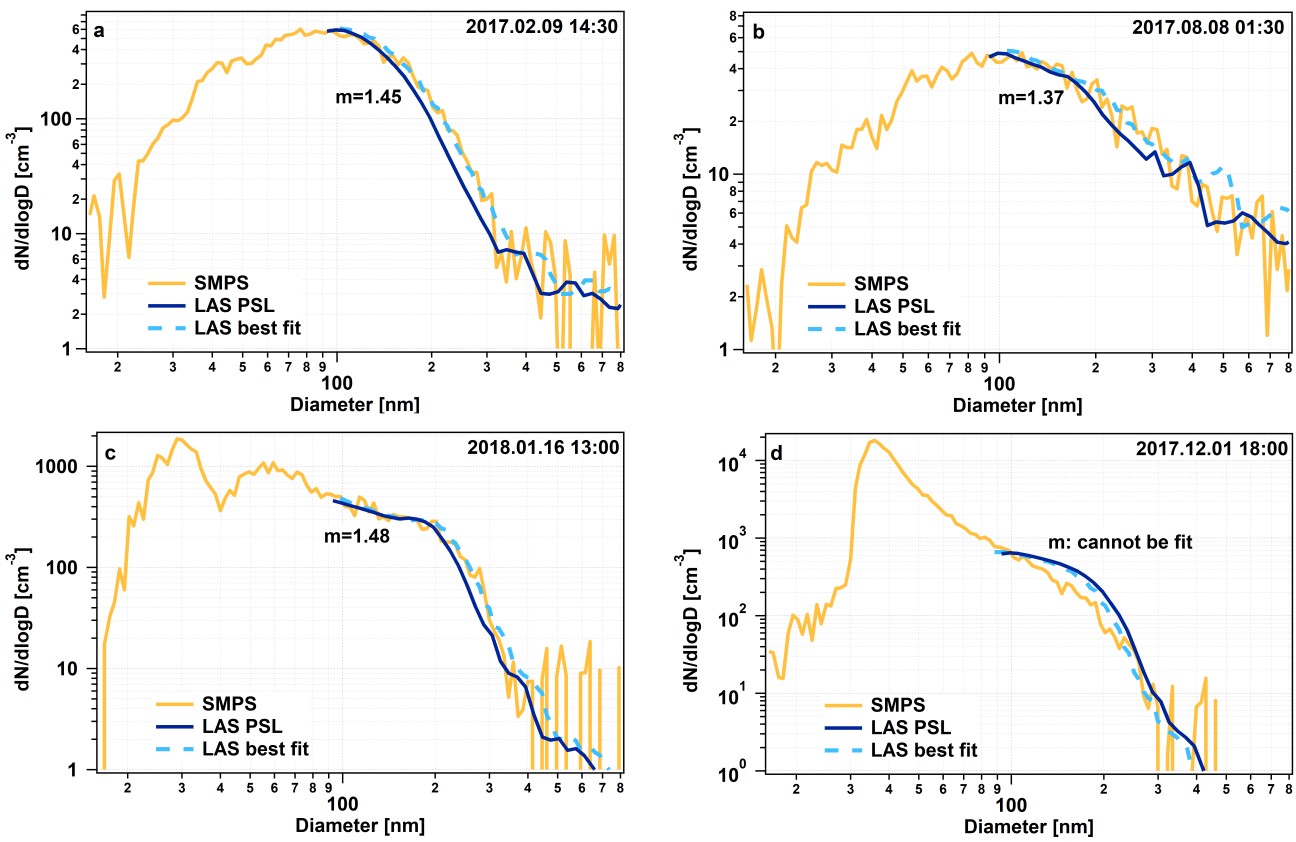

**Figure 4.** 4 examples on the refractive index fit performance. The orange line shows the measured SMPS number size distribution, whereas the blue lines (dark: PSL calibrated, light and dashed: best fit) show the LAS number size distributions.

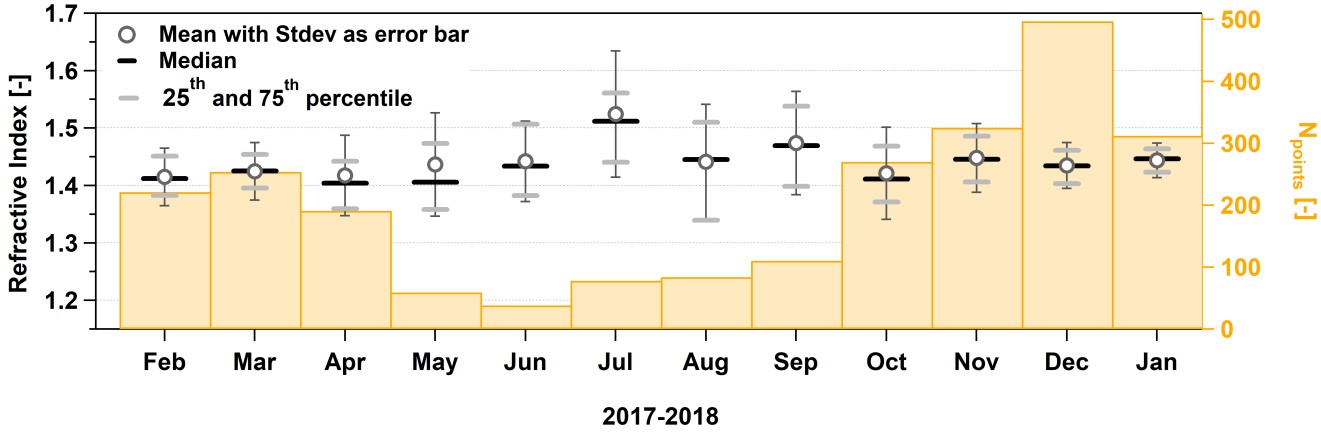

**Figure 5.** The monthly averages (with error bars as the standard deviation), medians and percentiles of $RI_{eff}$ from the coastal Antarctica, measured at $\lambda=633$ nm for dry aerosol particles. The orange bars refer to the right axis and show the number of successful RI retrievals in the corresponding month.

negligible during winter, biogenic sulphur aerosol reaches its annual maximum in austral summer between January and March (Minikin et al., 1998). At our investigated wavelength of 633 nm, sea-salt has an RI of 1.49 (Shettle and Fenn, 1979), sulfuric acid 1.42 (Palmer and Williams, 1975), ammonium sulphate 1.53 (Toon et al., 1976), ammonium bisulphate 1.47 (Chylek and Wong, 1995), sodium nitrate 1.46 (Cotterell et al., 2017), ammonium nitrate 1.52 (Toon et al.,

**Table 1.** The monthly and yearly ($\sum$) averages, standard deviations (Stdev), medians and percentiles of the $RI_{eff}$ from coastal Antarctica, measured at $\lambda$=633 nm for dry aerosol particles.

| Month | $25^{th}$ percentile | Median | $75^{th}$ percentile | Mean | Stdev | $N_{points}$ |
|---|---|---|---|---|---|---|
| Feb | 1.38 | 1.41 | 1.45 | 1.41 | 0.05 | 221 |
| Mar | 1.40 | 1.43 | 1.45 | 1.42 | 0.05 | 254 |
| Apr | 1.36 | 1.40 | 1.44 | 1.41 | 0.07 | 191 |
| May | 1.36 | 1.40 | 1.47 | 1.42 | 0.09 | 59 |
| Jun | 1.38 | 1.43 | 1.51 | 1.44 | 0.07 | 38 |
| Jul | 1.44 | 1.51 | 1.56 | 1.50 | 0.11 | 78 |
| Aug | 1.34 | 1.45 | 1.51 | 1.44 | 0.10 | 84 |
| Sep | 1.40 | 1.47 | 1.54 | 1.47 | 0.09 | 110 |
| Oct | 1.37 | 1.41 | 1.47 | 1.42 | 0.08 | 270 |
| Nov | 1.41 | 1.45 | 1.49 | 1.45 | 0.06 | 325 |
| Dec | 1.40 | 1.43 | 1.46 | 1.44 | 0.04 | 497 |
| Jan | 1.42 | 1.45 | 1.46 | 1.44 | 0.03 | 312 |
| $\sum$ | 1.37 | 1.41 | 1.46 | 1.44 | 0.08 | 2439 |

1976), MSA 1.43 (Virkkula et al., 2006) and black carbon 1.75+0.43i (Hess et al., 1998).

The chemical composition was determined from the daily filter measurements of the ionic composition and from the eBC measurement of the MAAP. The mass concentration of the dominant component of sea salt was calculated from the $Na^+$ ion. It was assumed that $NH_4^+$ is preferentially present as ammonium sulphate (($NH_4)_2SO_4$) and/or ammonium bisulphate ($NH_4HSO_4$) salt due to the high nss-$SO_4^{2-}$/$NH_4^+$ ratio of around 11.2±8 (annual mean ± Stdev). In addition, formation of ammonium nitrate ($NH_4NO_3$) has to be considered. Part of the nitrate can also be bound as $NaNO_3$. The remaining $SO_4^{2-}$ was assumed to be present as sulfuric acid.

We do not have any information on the organic carbon mass fraction for our measurement period, and therefore we could not include this component into the calculation. However, previous water soluble organic carbon (WSOC) mass concentration measurements (Weller et al., 2015) showed, that in the austral summer of 2011 the WSOC average mass fraction was less than 3% and therefore we believe that organic carbon does not have a significant influence on the resulting RI. Using this chemical composition and assuming that the aerosol is homogeneously and internally mixed, the RI can be calculated from the volume fraction and the RI of the individual components. The imaginary part of the RI was again neglected, which is a justified assumption, because the volume fraction of the eBC never exceeded 0.1 % in 2017. This amount of eBC would add at most a $\approx 4 \cdot 10^{-3}$i imaginary value to the RI.

The average RI calculated from the chemical composition in 2017 becomes 1.47, and is in a good agreement with the optically retrieved $RI_{eff}$ of 1.44. The reason for the slight discrepancy might be caused by the used assumptions. In addition and in contrast to the bulk chemical composition, the RI

calculation derived from the SMPS and OPC data are based on a limited size range between 120 and 340 nm. As discussed later in Section 3.7, RI changes slightly with the particle size.

Finally, we calculated RI separately for summer (November to February) and winter (March to October) from the aerosol chemical composition. We found higher RI values of 1.48 during austral winter compared to 1.45 during summer. This may be caused by the much higher sea salt aerosol portion during winter with the highest RI among the ionic compounds. Note also the significantly higher $RI_{eff}$ values for the winter months July and September (Fig. 5).

### 3.6 Impact of general weather situation and local contamination

Neumayer station is situated 1.5 km north of the measurement site, thus contamination during northerly winds, but also when the wind speeds are very low, has to be considered. We start with examining whether the actual wind direction influences our data in general, followed by a case study when diesel engines were operated right next to the measurement site. Contamination is mainly associated with high concentrations of black carbon. Black carbon has an RI of 1.75+0.43i (Hess et al., 1998) which is considerably higher than of any other natural chemical components of the aerosol. Note also the distinct imaginary part of the RI.

The prevailing wind direction at the SPUSO is east, associated with high wind speeds above 10 ms$^{-1}$, frequently exceeding even 20 ms$^{-1}$. Easterly wind directions, especially if they are accompanied with high wind speeds, are characteristic for the impact of passing cyclones and marine air entry. The second frequent wind direction is south, with wind speeds generally below 10 ms$^{-1}$. This weather situation is characteristic for advection of more continental

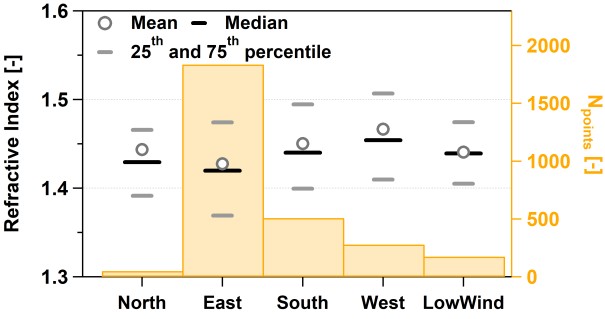

**Figure 6.** The averages, medians and percentiles of the $RI_{eff}$ from the coastal Antarctica separated by the wind direction, measured at $\lambda$=633 nm for dry aerosol particles. The orange bars refer to the right axis and show the number of successful RI retrievals.

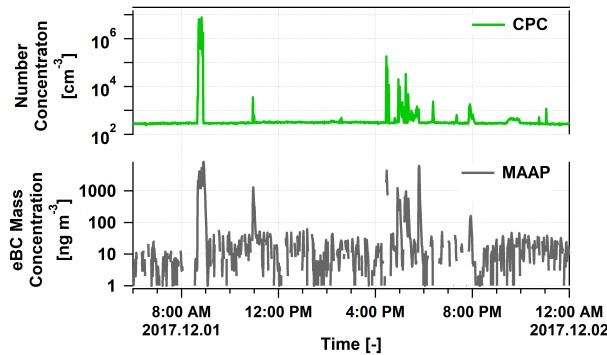

**Figure 7.** The particle number concentration (green) and the equivalent black carbon mass concentration (black) measured on 01.12.2017

air masses by katabatic winds. Westerly winds are usually caused by low-pressure systems in the southern Weddell region and associated with moderate winds speeds between 10 and 20 ms⁻¹. Northerly winds are virtually absent (König-
5 Langlo et al., 1998) and if present, mark a period of potential contamination from the station. We have separated the $RI_{eff}$ data according to different wind direction sectors to examine whether different air masses are associated with particles showing different RI. To this end, we defined the wind di-
10 rection sector between 315° and 45° as north, 45° and 135° as east, 135° and 225° as south and 225° and 315° as west. We categorized all data associated with wind speeds below 2 ms⁻¹ separately (LowWind in Figure 6).

Overall, our measurement period was representative and
15 meaningful for each individual sector, even for the inherently few data related to northerly wind directions. Figure 6 shows the $RI_{eff}$ values, sorted according to the mentioned categories. The gray circles show the time averages, the black sticks the medians, the gray sticks the 25th and 5th per-
20 centiles. In summary, no significant dependency of $RI_{eff}$ on the wind direction or wind speed is observable. We conclude that the general weather situation, just as local contamination, has no impact on $RI_{eff}$. Even adverse wind condition associated with potential contamination from the ex-
25 haust fumes of the main station did not cause any significant change of $RI_{eff}$.

In order to further investigate the problem of the contamination we performed a case study on a time period when planned contamination reached the SPUSO. This was the
30 same construction event which was already shown in Fig. 4d as an example for an unsuccessful fit when the aerosol was changing too fast. On the day of 01.12.2017, diesel engine powered machines were in operation in the very close vicinity of the measurement site.
Figure 7 shows the particle number concentration (green) and the black carbon mass concentration (black) as measured

by the CPC and MAAP, respectively during this construction episode. The highest concentrations were present during the morning and the late afternoon even exceeding $6 \cdot 10^6$ cm⁻³ and $8 \, \mu g \, m^{-3}$ which are 3–4 orders of magnitude higher 40 than the values without contamination (Weller et al., 2011, 2013). Unfortunately, these concentrations changed very fast, depending on whether the engine emissions were directly reaching our inlet, and therefore most of the time, we were not able to perform a fit for the RI. We have only one single 45 scan when the concentration was stable enough and elevated, allowing us to assume that we determined $RI_{eff}$ for a contaminated situation.

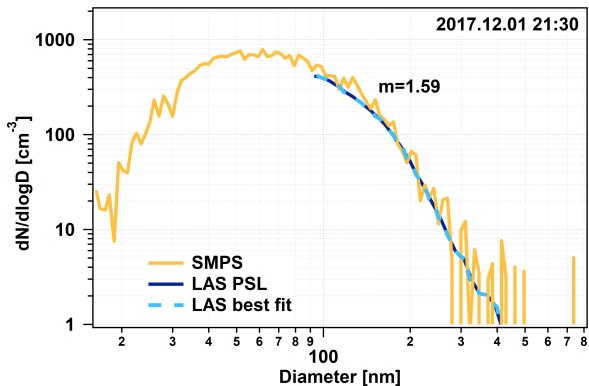

**Figure 8.** A successful RI fit from 01.12.2017 with high contamination present

Figure 8 shows this fit with the retrieved RI of 1.59. One can see that the original LAS scan fits already very well, 50 which means that the RI of the factory calibration of PSLs give us a good solution. This retrieved $RI_{eff}$ is significantly higher than the values we normally got. We can assume that the increased black carbon concentration caused this effect, and increased RI values might be an indicator for strong con- 55 tamination at this site. This time period, and any other time

period with known contamination was removed from the statistical calculations.

### 3.7 Size dependent contribution to the scattering

In the following we will calculate the contribution of the particles with different sizes to the scattering coefficient. Unfortunately, the LAS data was not usable above 600 nm during the time period when the particle losses were high, and therefore we can only do these calculations for an almost 2-months long summer period (01.12.2017-20.01.2018) when the LAS was installed right below the aerosol inlet. It was assumed that the derived $RI_{eff}$ is valid along the complete number size distribution (between 16 nm and 5000 nm) and that the particles are spherical and thus Mie calculation can be used for the determination of the single particle scattering at the wavelength of 633 nm. The scattering coefficient size distribution of the dry aerosol was calculated as follows:

$$\frac{d\sigma_s(D)}{d\log D} = C_s(D, \lambda, m) \cdot \frac{dN(D)}{d\log D} \qquad (4)$$

where $\sigma_s$ is the scattering coefficient in $m^{-1}$, $m$ is the derived, time dependent $RI_{eff}$ without a unit and $C_s$ is the scattering cross section of the individual particles in $m^2$. To calculate $C_s$ we used our custom written Mie code.

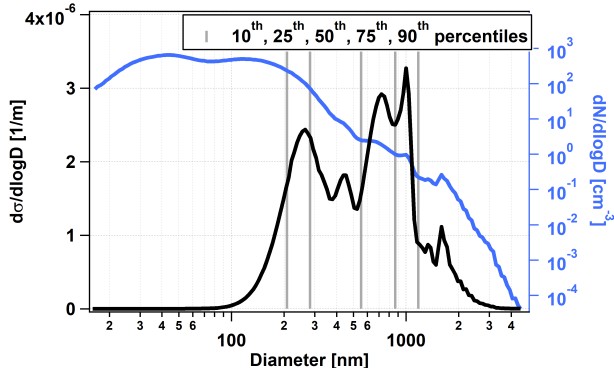

**Figure 9.** The average dry scattering coefficient size distribution (black line) at 633 nm wavelength and the corresponding particle number size distribution (blue line, right axis) as function of the particle diameter. The grey lines show the 10[th], 25[th], 50[th], 75[th] and 90[th] of the scattering coefficent distribution.

Figure 9 shows the time average of $d\sigma_s(D)/d\log D$ as function of the particle diameter. Next to it, the average number size distribution (blue line, right logarithmic axis) for the same time period is also shown. As we can see, particles smaller than 100 nm or larger than 3 μm do not contribute significantly to the scattering. 80% of the total scattering amount come from the size range between 208 and 1170 nm. Interestingly, the distribution is multimodal, having two main peaks around 260 and 860 nm. The median of the distribution

is at 550 nm which is much higher than the median of the number size distribution (64 nm), as expected, because scattering increases faster than linearly as function of the particle diameter. The average number size distribution is also multimodal with two distinct peaks around 40 nm and 140 nm. Considering the time evolution and not temporal averages we see, that these two peaks, as well as the two main peaks of the scattering coefficient size distribution, are often present simultaneously. In conclusion, the bimodality is not the product of time averaging of single modes appearing at different times.

Finally we investigate the effect of neglecting the imaginary part of the RI on the scattering coefficient. As we have seen in Section 3.5 including the eBC in the chemical composition adds at most an imaginary part of $\approx 4 \cdot 10^{-3}i$ to the RI. We recalculated the average scattering coefficient size distribution adding this imaginary part to the RI. This gives us a highest possible estimate on the error we make if we would neglect the imaginary part of RI. It turns out that the relative difference of the scattering coefficient size distribution considering $4 \cdot 10^{-3}i$ RI instead of 0.0i never exceeds 1.7% irrespective of the particle diameter.

### 3.8 Size dependence of the refractive index

To examine the dependence of $RI_{eff}$ on the given particle size distribution, we again have to restrict the time period to (01.12.2017-20.01.2018) when the LAS's particle losses were minimised. During this period we have an SMPS–LAS overlapping size range between 120 and 900 nm. If we calculate the temporal average over this complete time period, most of the noise is averaged out as well, so that we can use most of this overlapping size range for the RI fit. Moreover, the overall size distribution range can now be divided into 4 subranges suitable for separate $RI_{eff}$ calculations, representative for the corresponding subrange (Fig. 10).

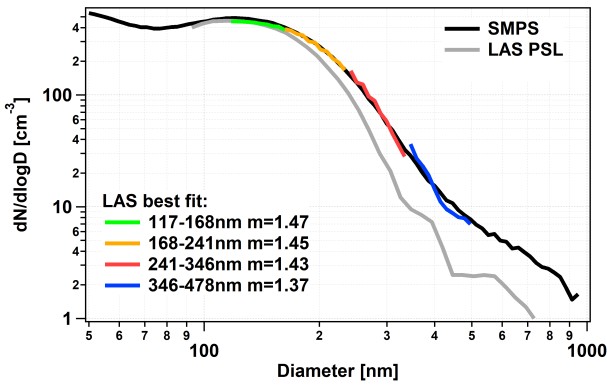

**Figure 10.** The average dry aerosol number size distribution measurements during December 2017 and January 2018 as measured by the SMPS (black line) and the LAS (gray line). The coloured lines show the 4 individual RI fits using 4 different particle size ranges.

Figure 10 shows the time averaged LAS (gray line) and SMPS (black line) number size distributions. We have chosen the following particle size ranges for the separate RI fit: 117–168 nm, 168–241 nm, 241–346 nm and 376–478 nm ensuring, that we have a similar number of size distribution measurement points for the fit procedure in each of the size ranges.

With the increasing particle size, we needed to apply a lower RI in order to have the best match between the LAS and the SMPS. In the first range we got an $RI_{eff}$ of 1.47, in the second 1.45, in the third 1.43 and in the fourth 1.37. According to Figure 10 the $RI_{eff}$ decreases slightly within the first 3 subranges of particle diameter ($RI_{eff}$ between 1.47 and 1.43), but more pronounced for the highest range ($RI_{eff}$ = 1.37)

The conspicuously lower $RI_{eff}$ in the highest investigated size range may originate from a significantly changing chemical composition. Interestingly, sea-salt particles should dominate this higher size range, but this would result in a higher $RI_{eff}$. Hence one may speculate about a coating of sea-salt particles in this special case (probably organic material with typically lower RI). The presence of a coating or a different aerosol source might also explain the bimodality of the scattering coefficient size distribution (Section 3.7). However, we have to keep in mind, that this is pure speculation and we have no proof of it.

## 4    Conclusions

We have calculated the real RI for dry natural aerosol at a coastal Antarctic measurement site using the overlapping size range of two instruments measuring the number size distribution in two different ways: optically and by electrical mobility. The yearly average (±Stdev) of the RI was calculated based on the data from almost a complete year and turned out to be 1.44 (±0.08). This average is in very good agreement with the RI value of 1.47 which we derived from filter based chemical composition measurements. The good agreement shows that at least for coastal Antarctica this method reliably delivers the RI values without the additional effort of a chemical characterization of the aerosol.

Based on this, we recommend this single, temporally constant refractive index value for modeling of aerosol optical properties. In this context we suggest supporting investigations to examine the validity of this approach and the usage of seasonal independent $RI_{eff}$ values for the Antarctic region.

In spite of the strong seasonal variability of the chemical composition at the measurement site (e.g. 86% sea-salt present in winter, 50% in summer), we could not identify a corresponding seasonal trend of the RI, which is in good agreement with RI derived from the chemical composition of the present aerosol. We conclude that the given high variability of the ionic composition of the aerosol typical for coastal Antarctica causes only minor variability in associated RI values. We could not find any significant influence from the wind direction either. We conclude that the general weather situation. just as local contamination, has no significant impact on $RI_{eff}$.

In forthcoming related investigations at Neumayer, a year-round optical closure experiment is planned. For this, the size range between 16 nm and 5 μm as well aerosol scattering coefficients by integrated nephelometer measurements will be employed.

*Data availability.* Data reported here are available at https://doi.pangaea.de/10.1594/PANGAEA.899429 and https://doi.pangaea.de/10.1594/PANGAEA.899430 for scientific purposes.

*Author contributions.* ZJ has performed the measurements, analysed and interpreted the data and wrote the manuscript. RW built up the measurement site, supervised the measurements and the data analysis and reviewed and edited the manuscript.

*Competing interests.* The authors declare that they have no conflict of interest.

*Acknowledgements.* The authors would like to thank all members of the 37[th] overwintering team of the Neumayer III station for their support and for being a great group. The first author of the paper would like to express her gratitude to her brother for his support during the harsh winter months in Antarctica.

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
