# Peer review of "One year of aerosol refractive index measurement from a coastal Antarctic site"

_Atmospheric Chemistry and Physics, 2019_

## Short Comment (SC1) · 29 May 2019

This short comment is to request that the authors "+0i" to the refractive indices reported in the abstract. For example, 1.50+0i instead of 1.50. I would also suggest adding a note about the high SSA of 0.99 measured at their sampling site to the abstract. After reading the abstract, I was left wondering about the strange formatting; after reading Section 2.6 of the manuscript I found a satisfying explanation.

---

## Referee Comment (RC1) · Anonymous Referee #1 · 21 Jun 2019

Aerosol refractive indices determined from two different size distributions measured at a coastal Antarctic site are described in this manuscript. The authors have done a commendable job of deriving these values under difficult measurement conditions and have attempted to account for calibrations and measurement losses with minimal supplies and creative means. Their results suggest that the refractive index was very stable over the period of the study, reflecting the low relative variability of aerosol composition at the site. The results are useful for understanding aerosol optical properties in extremely remote and clean regions, as well as method dependent variability in terms of sampling duration and sampling losses. The authors do not discuss size distribution information in this paper (perhaps this is the topic of a later analysis) but it would be interesting to understand size distribution variability, especially considering the scatter-

ing distribution results. Methods that determine refractive index from size distributions measurements are important because they provide estimates of refractive indices from which scattering coefficients can be determined, without the additional effort and cost of composition measurements needed to calculate refractive index. These methods also provide higher time resolution refractive indices than composition measurements typically can.

The paper could be strengthened by including some discussion of the implications of this work towards climate change estimates, which is one of their stated motivations for the study. There were several typographical errors in the text that I did not attempt to correct; I suggest another careful editing of the manuscript. I recommend publication after addressing comments listed below.

Specific comments

Page 1, Line 1-3: This sentence is a bit unclear. I think the authors are trying to state that reducing uncertainties in modeling evaluations of climate change require more accurate aerosol optical properties. It might also help to point out why it is important to have measurements of refractive index at the poles- is this because climate change is enhanced there relative to other locations? It would also help to state when these measurements were made earlier in the abstract.

Page 1, line 7: Do the authors refer to 2439 individual size distributions when they refer to measurement points?

Page 1, line 7: It is always helpful to also include wavelength and relative humidity conditions associated with optical property measurements. Sometimes people only read abstracts and figures.

Page 1, line 8: It would also help to include some uncertainty estimates or an estimate of standard deviation with the reported average RIeff.

Page 1, line 13-19: This paragraph is a bit unclear. I understood it better after I had

read the paper, but as part of the abstract it could benefit from clarification. Part of the issue is that the location of the site has not been described yet, so understanding the wind direction in respect to the site location and the Neumayer station is a bit confusing. It might be helpful to state the wind direction impact in a more generalized way, or provide more description of the site location first.

Page 1, line 20: Referring to the time-averaging here is also confusing without having read the paper first. It may not be necessary to include in the abstract. Are these differences larger than the uncertainties of the measurements?

Page 1, line 20: It would also help here to state something about the larger implications of this work, tying back to the point of the study so that the reader grasps the larger importance of the work.

Page 2, line 14: Probably the most common method would be from volume-weighted calculations of composition data because these are generally more available than detailed size distribution and scattering measurements.

Page 2, line 24: It is important to point out that these methods do not directly measure refractive index- they use closure studies between a variety of measurements to determine the refractive index that provides agreement.

Page 2, line 25: Remove "parallel their" for clarity.

Page 2, line 34: What does it mean, "Until the particles disappeared"?

Page 3, line 5: To what RH do these values correspond?

Page 3, line 5: It would also be helpful here to point out the importance of this work- why is it important to have yearlong estimates of RI from the Antarctic? What are the larger implications?

Page 3, line 10: Include the study time period earlier, it will help when considering the information provided in the next paragraph.

Page 3, line 31: Is RH and temperature measured? Define RH if it has not already been done.

Page 5, line 8: How was 'penetration efficiency' determined?

Page 6: Was there a quantitative measure by which "too noisy" was defined?

Page 6, line 7-8: This sentence is confusing, I suggest rewording.

Page 6, line 10: How was "good enough signal to noise ratio" defined?

Page 11, line 10: How long was the averaging time? I am not sure why rapidly changing aerosol should results in poor fit if both instruments are sampling the same aerosol at the same time? Was the aerosol changing faster than the SMPS could sample it? Did CPC data indicate this?

Page 11, line 20: Based on the stated estimated uncertainty ($\sim$3%), reporting this many significant digits seems unnecessary. This comment also holds for Table 1 and reporting of values throughout the paper. It would also help to report standard deviation for each month in Table 1.

Page 12, line 11: How was "significantly different from the yearly mean" determined?

Page 11, line 15: Providing standard deviations would help in discussing the lack of scatter in the data.

Page 13, line 4-5: Organic carbon was not measured, can the authors comment on the possible contribution to mass at the site? Has it been measured during previous studies?

Page 13: line 14: Typically, thermodynamics favor the formation of ammonium sulfate before ammonium nitrate, such that if there is enough ammonium available, it will neutralize sulfate before nitrate. What is the molar ratio of NH4/SO4 during the study? Were the aerosol acidic? (Seinfeld and Pandis, 1998)

Page 14, line 15: The RIeff values look somewhat lower with East winds. It would help to expand the scale in Figure 6 to values ranging from 1.35-1.55 to sees these differences more clearly. Are these differences greater than uncertainty in the values themselves?

Page 15. Line 3: The values in Figure 6 refer to an average value of all RIeff from the North. Did any individual distributions suggest contamination?

Page 16, line 12: Please provide units for scattering coefficient and a more detailed description of Cs. It must include a diameter-squared parameter.

Page 16, line 14. Is the scattering distribution in Figure 9 an average? Is the bimodal distribution is a function of the averaging of several different monomodal distributions at different times (suggesting interesting changes in the aerosol size distribution or refractive index).

Page 17: Figure 9 caption: What time period do these data correspond to? Is this an average of several distributions?

Page 17, line 9: This sentence is unclear.

Page 17, line 15-18: As Figure 2 shows, the instrument response at higher sizes shows a cross over region such that the instrument is unable to distinguish between refractive indices. Others have also shown this behavior at larger sizes (Garvey and Pinnick, 1983; Hand et al., 2000). Reporting refractive indices in these larger size ranges is probably not meaningful.

Page 18, line 15: I am not sure what the authors mean by "geographic borders of this value's validity"?

Page 19, line 14. Including experimental uncertainties here would help, as would re-stating the RI derived from composition data. Deriving Reff and the ability to calculate scattering coefficients using it and the measured size distributions, without the additional effort and cost of composition measurements, is an important benefit to this

analysis. Comparing the RI derived from composition in the context of experimental uncertainty can strengthen the arguments for the importance of this type of analysis. In addition, composition measurements are usually unavailable with the time resolution of size distribution measurements.

Page 19: Line 1: It would help to state the seasonal variability more strongly if the readers could comment whether the seasonal values or wind direction values were greater than the experimental uncertainty. As it stands, it appears somewhat subjective.

General comment: Please provide wavelength and RH on each of the figure and table captions- it can help the reader quickly orient themselves without having to scroll back through the text.

References:

Seinfeld and Pandis, Atmospheric Chemistry and Physics, 1998, John Wiley and Sonas, page 538.

D.M. Garvey and R.G. Pinnick (1983) Response characteristics of the Particle Measuring Systems Active Scattering Aerosol Spectrometer Probe (ASASP-X), Aerosol Science and Technology, 2:4, 477-488.

J.L. Hand, R. B. Ames, S. M. Kreidenweis, D. E. Day, and W. C. Malm (2000) Estimates of particle hygroscopicity during the Southeastern Aerosol and Visibility Study, Journal of the Air & Waste Management Association, 50, 677-685.

---

## Referee Comment (RC2) · Anonymous Referee #2 · 10 Jul 2019

Review of "One year of aerosol refractive index measurement from a coastal Antarctic site" submitted to ACPD by Z. Juranyi and R. Weller:

The manuscript presents a thorough derivation of the real part of the refractive index for atmospheric aerosol as encountered during the course of one year at the Antarctic station Neumayer.

The measurements and calculations as well as related caveats are all well described and reasonable, the techniques used are all sound. The resulting value is reasonable and of good use for climate modeling.

My main concern is that, in some parts more than in others, the language needs polishing, beyond what can be expected to be caught during the ACP-language editing

at the end of the publication process. I will not list all these occurrences where the English has to be approved, but give at least an already longish list in this review at "Technical comments". Also most of the "More general comments" concern issues that likely arose due to strangely formulated sentences. As such, I can recommend the manuscript for publication once these language issues will have been dealt with, and will give "minor revisions" although the list below is somewhat long and includes a few issues other than only linguistic.

——

More general comments:

page 5, line 5: Information on where exactly particles entered the tubing during these experiments would be good. Just underneath the roof, close to the inlet line?

page 6, line 1-2: You show this polynomial only up to 400nm - although the data (blue dots) go up to 1000nm - does this mean you only used particles up to 400nm? Please add an explanation and/or prolong the line in Fig. 1.

page 6, line 24 ff: I have an idea what you did, here, but I am not entirely sure - this could certainly be formulated much clearer. What I think you did is the following: (1) - calculate TIR for a fixed RI (2) - take the value from the TIR at the diameter of the PSL particles. I guess one confusion was due to your use of the word "bin boundary diameter". Maybe this could be defined once and then "LAS diameter" could be used instead, throughout the text, to make the text flow better? Also, this passage sounds as if there would basically only be a signal in one bin during a PSL calibration - this is most likely not the case. Describe this more clearly.

page 8, line 15-16: Concerning possible changes in particle composition: The way you did your derivation of RI, however, was to assume that the particle chemistry was the same for all particles in one measured size distribution? Please explicitly say this here somehow, as I got confused by your remark here.

page 8, line 29-30: "We used the method introduced in the sections 2.5 and 2.6 to determine the RI of this e-cigarette smoke." But in the paragraph above you said that the RI of the cigarette smoke was 1.43, based on literature (and if you would have had to determine it first you would run into issues with circular reasoning if you then would use this measurement to calibrate the LAS TIR). I assume this again is an issue with formulating the text. Please review.

page 9, line 4-6: Again confusing, so let me ask you again if this is what you did: When retrieving the RI for the uncorrected LAS data, you obtained an RI of 1.35, but when you corrected the measured LAS size distribution as described above and then retrieved the RI again, you got a value of 1.43, in agreement with literature. - If this is what you did, feel free to use my sentence here in the review instead of what you wrote. Your text here was hard to follow and it took me a while until I understood what you (likely) meant.

page 12, first paragraph of 3.5: I would recommend to start this paragraph differently – the first sentence states something that seems not to hold once one read the list of RIs: when looking at this list and the most abundant components of the aerosol, one wonders if this really can be in good agreement, since particularly sea salt and ammonium sulphate are clearly above the value you retrieved. This all becomes much clearer further down, but I recommend to avoid confusion and to remove this first sentence or replace it with a sentence that says what you are aiming at in 3.5.

page 17, line 2: Maybe add that you expect this because scattering scales with the diameter squared.

page 18, first paragraph: You spend most of the space in this paragraph on discussing why this one value does not make sense, and the reason basically is that the underlying data is corrupted. Maybe just do not present the blue line in the figure and say up front that due to a) the strange kink in the LAS distribution and b) due to the low particle number concentration at the larger particle diameters no useful value resulted. (I'd be

afraid that otherwise in the future someone might just grab that value from your figure without reading the text and use it.)

Also, this lowering for particles > ∼350 nm, together with the bimodality you showed in Fig. 9 - could this point towards two different sources for particles? This is something you could discuss here, instead.

———

Technical comments:

Page 1, line 11: June and September have (not has).

page 2, line 8: really "RI values of the" (or "for the . . .")?

page 2, line 20: Start a new sentence at "additionally" – these are two main clauses that are independent.

page 3, line 18: "can barely reach it, the main aerosol" better start a new sentence at "the" or combine these two sentences with an "and". (Again, these are two main clauses.)

page 3, line 19: Antarctic is always capitalized.

page 4, line 8: Exchange "makes" with "enables" and delete "possible" at the end of the sentence.

page 4, line 12: "is" is missing between "and" and "only".

page 4, line 13: Replace "the time" with "a time" (you are not talking about a specific time here).

page 4, line 19: Replace "in a minute" with "per minute".

page 4, line 21: Delete the "the" in "with the time".

page 5, line 4: Replace "such, that more" with ", so that several".

page 5, line 13: Replace "why we only have the complete number size distribution until 5 $\mu$m after . . ." with "why we have the complete number size distribution only up to 5 $\mu$m after . . .".

page 6, line 12-13: You wrote: "or exceeded the particle number detected during a one hour SMPS scan with 400 cm$-3$ concentration." – This sentence sounds quite strange, and I guess you could replace all that with "or exceeded 400 cm$-3$ " – or maybe "exceeded 400 cm$-3$ (as measured by the SMPS) for the duration of one hour"? But it's not clear to me if you refer to the concentration of the integrated particle number size distribution?

page 6, line 20: Replace "to any other RI" with "for any other RI".

page 6, line 26: Do you really mean an OPC (i.e., a counter) or rather an OPS (optical particle sizer)? (Check this also in the introduction, line 19 on page 2).

page 6, line 31: Put the "RI" in front of the parenthesis.

page 8, line 9-10: This again is a strangely formulated sentence.

page 8, line 17: "and that the imaginary part of the RI is negligible." You just said that before, a few lines above. But you could instead say: "and, as said above, the imaginary part of the RI is negligible."

page 8, line 20: "was" -> "were" (end of this line).

page 9, Fig. 3: Is the legend correct? The green dashed shoud be "uncorrected", right? Also, at first this was a bit confusing, as the fits are not close to the data - maybe change the naming convention, here and in the text, maybe "RI-fit" instead of just "fit"?

page 10, line 5: Replance "count with" with "expect".

page 10, Fig. 4: It is slightly counter-intuitive that you give the "original" LAS curve in deep blue, which is close to the color in which you also give the resulting RI (m). I'd suggest to use the same color for the dashed line (which is the final result) and the RI

(m), and a different color for the "original" LAC curve.

page 10, line 8: Replace "about" with "of".

page 13, line 20-21: "The slight difference may come next to the used assumptions from the fact that we used …" – sounds quite strange, not English at all. Please reformulate.

page 13, line 26: Replace "among the" with "compared to the other".

page 13, line 26-27: "This might explain as well why we have gained with the fitting procedure a slightly higher RIeff values for two winter months." – Again strange sentence, the sequence of the words is somewhat off. Please reformulate.

page 14, line 3: Delete "stay" and add a comma before "caused".

page 14, line 4: A space is missing after "site".

page 14, line 10-12: The use of the word "points" here is a bit stange – maybe better "data"?

page 14, line 15: "independent on from which direction the wind blew" -> replace with "independent of the wind direction".

page 15, line 5: Add "be" between "not" and "fit".

page 16, line 5: Add "ly" to "different".

page 17, line 4: "We can only use again the" -> "Again, we can only use, ..."

page 17, line 8: Replace "high enough" with "sufficiently many".

page 17, line 9-10: Confusing formulation. A better way to say that: "The resulting RI values will then be valid for the respective particle sizes."

page 18, line 3: "0.0018 is also close to the limit of 0.02 …" There is more than a factor of 10 between the values you give here. Check!

page 18, line 2: The first word here should be plural ("instruments"). Author contributions: "Juranyi has" (not "have").

---

## Referee Comment (RC3) · Anonymous Referee #3 · 5 Aug 2019

GENERAL REMARKS

The study presented in the manuscript investigates the complex index of refraction for aerosol sampled at a coastal Antarctic site. The period covered by this study spans a full annual cycle. The experiment and the analyses are very carefully conducted, given the difficult situation at the sampling site with very limited access to, e.g., reference materials for instrument calibration and aerosol laboratory equipment. Overall, the study is scientifically sound and makes a significant contribution to the research field of aerosol impacts on climate. The manuscript is well organized and fits very well into the scope of the journal.

My main concern is discussed below in the specific comments. Furthermore, before publication the language requires careful inspection. Then, few minor revisions need

to be considered which are listed below. In my report below I focus on the issues not yet mentioned by the other referees.

SPECIFIC COMMENTS

The authors report on aerosol refractive index observations but never mentioned that the index of refraction is a complex number. Particularly, the imaginary part of the refractive index constitutes the light-absorbing properties of the sampled aerosol. As Weller et al. (2013) reported, there is a small but significant fraction of lights-absorbing material contained in the aerosol in Antarctica. However, the authors never refer to this observation in a quantitative manner, nor they stated the assumption of a zero imaginary part of the refractive index. Furthermore, the scattering cross-section as calculated by Mie or Rayleigh-Debye-Gans theories depends on the square of the complex refractive index which includes the imaginary part.

I request a discussion of the uncertainties in calculating the real part of the refractive index, when neglecting the imaginary part. The effect may be small but it should be mentioned since the imaginary part plays a crucial role in the aerosol radiation interaction.

MINOR ISSUES

Figure 3: I assume that the dashed green line refers to the LAS uncorrected best fit, please add.

Figure 10: I propose to specify LAS original as LAS (m = 1.59); the term "original" suggests that data were modified, which is, however, not the case.

When reporting on the black carbon mass concentration determined by the MAAP, the authors should use the today accepted terminology of "equivalent black carbon" (eBC); see Petzold et al. (2013).

REFEENCES

Petzold, A., Ogren, J. A., Fiebig, M., Laj, P., Li, S.-M., Baltensperger, U., Holzer-Popp, T., Kinne, S., Pappalardo, G., Sugimoto, N., Wehrli, C., Wiedensohler, A., and Zhang, X.-Y.: Recommendations for reporting "black carbon" measurements, Atmos. Chem. Phys., 13, 8365–8379, doi: 10.5194/acp-13-8365-2013, 2013.

Weller, R., Minikin, A., Petzold, A., Wagenbach, D., and König-Langlo, G.: Characterization of long-term and seasonal variations of black carbon (BC) concentrations at Neumayer, Antarctica, Atmos. Chem. Phys., 13, 1579-1590, doi: 10.5194/acp-13-1579-2013, 2013.

---

## Author Comment (AC1) · 10 Sep 2019

The authors would like to thank Anonymous Referee #1 who helped us improve our manuscript.

reviewer comment: The paper could be strengthened by including some discussion of the implications of this work towards climate change estimates, which is one of their stated motivations for the study.

answer: Our derived refractive index values refer to dry aerosol particles, as they are derived from dry aerosol number size distribution measurements. We are not really able to extend the discussion towards climate change estimates, because we have no sufficient information on the water uptake (hygroscopicity) of the aerosol particles from

our measurement site yet. Therefore we could only speculate on the real ambient light scattering and absorption coefficients, which we do not want. Hygroscopicity measurements are planned for this austral summer (winter 2019-2020) at our measurement site which will hopefully allow us to make such a study.

reviewer comment: There were several typographical errors in the text that I did not attempt to correct; I suggest another careful editing of the manuscript.

answer: A careful editing was done to the manuscript.

reviewer comment: Page 1, Line 1-3: This sentence is a bit unclear. I think the authors are trying to state that reducing uncertainties in modeling evaluations of climate change require more accurate aerosol optical properties. It might also help to point out why it is important to have measurements of refractive index at the poles- is this because climate change is enhanced there relative to other locations? It would also help to state when these measurements were made earlier in the abstract.

answer: The sentence was reformulated, now it reads: "Though the environmental conditions of the Weddell Sea region and Dronning Maud Land (DML) are still relatively stable compared to the fast-changing Antarctic Peninsula, we may suspect pronounced effects of global climate change for the near future (Thompson et al. 2011). Reducing the uncertainties in climate change modeling requires inter alia a better understanding of the aerosol optical properties, and for this, we need accurate data on the aerosol refractive index (RI). Due to the remoteness of Antarctica only very few RI data are available from this region (Hogan et al. 1979, Virkkula et al. 2006, Shepherd et al. 2018).

reviewer comment: Page 1, line 7: Do the authors refer to 2439 individual size distributions when they refer to measurement points?

answer: We referred to 2439 averaged size distributions, it was removed from the manuscript.

[Figure]

reviewer comment: Page 1, line 7: It is always helpful to also include wavelength and relative humidity conditions associated with optical property measurements. Sometimes people only read abstracts and figures.

answer: The wavelength and the RH condition are added into the abstract.

reviewer comment: Page 1, line 8: It would also help to include some uncertainty estimates or an estimate of standard deviation with the reported average RIeff.

answer: The standard deviation value of RIeff was added to the abstract.

reviewer comment: Page 1, line 13-19: This paragraph is a bit unclear. I understood it better after I had read the paper, but as part of the abstract it could benefit from clarification. Part of the issue is that the location of the site has not been described yet, so understanding the wind direction in respect to the site location and the Neumayer station is a bit confusing. It might be helpful to state the wind direction impact in a more generalized way, or provide more description of the site location first.

answer: The paragraph was rewritten: "We find no significant dependence of the derived RIeff values on the wind direction. Thus, we conclude that RIeff is largely independent on the general weather situation, roughly classified in (i) advection of marine boundary layer air masses during easterly winds caused by passing cyclones in contrast to (ii) air mass transport from continental Antarctica under southern katabatic winds. Neumayer, the only relevant contamination source, is located 1.5 km north of the air chemistry observatory, where the measurements were performed. Given that northerly winds are almost absent, the potential impact of local contamination is minimized in general. Indeed our data show no impact of local contamination on RIeff. Just in one case, a temporary high contamination episode with diesel engines operating right next to the measurement site resulted in an unusual high RIeff of 1.59, probably caused by the high black carbon content of the exhaust fumes."

reviewer comment: Page 1, line 20: Referring to the time-averaging here is also confusing without having read the paper first. It may not be necessary to include in the abstract. Are these differences larger than the uncertainties of the measurements?

answer: The sentence was removed from the abstract.

reviewer comment: Page 1, line 20: It would also help here to state something about the larger implications of this work, trying back to the point of the study so that the reader grasps the larger importance of the work.

answer: This additional text was added to the abstract: "To conclude, our study revealed largely constant RIeff values throughout the year without any sign of seasonality. Therefore, it seems reasonable to use a single, constant RIeff value of 1.44 for modeling optical properties of natural, coastal Antarctic sub-$\mu$m aerosol."

reviewer comment: Page 2, line 14: Probably the most common method would be from volume-weighted calculations of composition data because these are generally more available than detailed size distribution and scattering measurements.

answer: The reviewer is wright, "The most common method" was changed to "A common method"

reviewer comment: Page 2, line 24: It is important to point out that these methods do not directly measure refractive index- they use closure studies between a variety of measurements to determine the refractive index that provides agreement.

answer: The following sentence was added to the manuscript: "We have to keep in mind that all above mentioned methods are not direct measurements of the RI. All of these methods search for RI values that provide good agreement in a closure study between different measured quantities."

reviewer comment: Page 2, line 25: Remove "parallel their" for clarity.

answer: Was removed.

reviewer comment: Page 2, line 34: What does it mean, "Until the particles disappeared"?

answer: The text was changed to: "until they could not see the particles in the microscope (i.e. until the applied oil's RI matched the RI of the collected particles)."

reviewer comment: Page 3, line 5: To what RH do these values correspond?

answer: To dry aerosol, the line was modified to: "In this paper we would like to present continuous data on the real RI at 633nm wavelength of the dry ambient aerosol as derived..."

reviewer comment: Page 3, line 5: It would also be helpful here to point out the importance of this work- why is it important to have yearlong estimates of RI from the Antarctic? What are the larger implications?

answer: The following text was added: "With this, our study aims at better understanding of the aerosol optical properties at a place where only very few such data are available with special focus on its temporal variability. Given the distinct seasonality of the aerosol composition (see Weller et al., 2008, Figs. 4 and 5 therein), we may likewise expect a seasonality of RI. To this end, continuous year-round data of RI are necessary, in particular regarding the lack of such measurements for the Antarctic realm."

reviewer comment: Page 3, line 10: Include the study time period earlier, it will help when considering the information provided in the next paragraph.

answer: The sentence was modified to: "The measurements presented in this paper were performed in the Air Chemistry Observatory (SPUSO from "Spurenstoffobservatorium") of the German Antarctic station of Neumayer III between February 2017 and January 2018."

reviewer comment: Page 3, line 31: Is RH and temperature measured? Define RH if it has not already been done.

answer: The RH of the aerosol entering the instruments was calculated from the measured outdoor temperature, RH and the measured indoor temperature. A definition to RH and the following sentence was added to the text: "The meteorological data used in this study (temperature, wind direction and speed and ambient RH) was measured directly on the roof of SPUSO."

reviewer comment: Page 5, line 8: How was 'penetration efficiency' determined?

answer: it is the ratio between the measured number size distribution with and without the tubing which causes the losses. The explanation was added to the text and it was renamed to particle transmission efficiency.

reviewer comment: Page 6: Was there a quantitative measure by which "too noisy" was defined?

answer: We did not have a real quantitative measure, just the fact that no successful fit was possible, which became possible with further averaging.

reviewer comment: Page 6, line 7-8: This sentence is confusing, I suggest rewording.

answer: done, the sentence now reads: "This strong seasonal variability is the reason why in summer a much shorter time averaging period is sufficient to enable a successful RIeff fit. To keep the highest possible time resolution of the derived RI, we have chosen the length of the time averaging to be time dependent. And this length was determined by the actual particle concentration."

reviewer comment: Page 6, line 10: How was "good enough signal to noise ratio" defined?

answer: Such that a RIeff fit was possible, was added to the text.

reviewer comment: Page 11, line 10: How long was the averaging time? I am not sure why rapidly changing aerosol should results in poor fit if both instruments are sampling the same aerosol at the same time? Was the aerosol changing faster than the SMPS

could sample it? Did CPC data indicate this?

answer: The averaging time was variable, at least one hour but up to 14 hours. But that is not the problem. The problem is if the aerosol changes significantly within one scan, which lasted 6 minutes for both instruments. The two instruments measure the number size distribution differently. The LAS captures all of the particles within this 6-minutes period, and therefore delivers an average number size distribution for this time period. The SMPS on the other hand scans through the different diameters during 6 minutes and therefore captures only one size at a time. Therefore e.g. when during the 6 minutes of a single scan the aerosol concentration doubles and the form of it remains constant, the LAS returns the average number size distribution, whereas the SMPS returns a skewed number size distribution with too low concentrations at the low diameters and too high at the high diameters (assuming that the SMPS is operated in an upscan mode). And yes, it could be also seen in the CPC data, that during the beginning of the scan the concentration was much higher than later. The text was changed to: "The reason why the fit did not work in this case was that the aerosol population was significantly changing within the duration of the SMPS scan. During the first half of the scan an aerosol plume with very high concentration reached the instruments. This appears in the SMPS scan as a very high fraction of small particles, because during the first half of the scan, the SMPS selected and measured the smaller particles. Contrary, the LAS captures all particles with different diameters at the same time, and therefore this event appears as an elevated overall concentration. This was an extreme and exceptional situation where some unavoidable construction was done around the SPUSO using machines powered by diesel engines."

reviewer comment: Page 11, line 20: Based on the stated estimated uncertainty (~3%), reporting this many significant digits seems unnecessary. This comment also holds for Table 1 and reporting of values throughout the paper. It would also help to report standard deviation for each month in Table 1.

answer: done

reviewer comment: Page 12, line 11: How was "significantly different from the yearly mean" determined?

answer: it was determined by a statistical T-test with a significance level of 0.01. The text now reads: "These values are significantly different from the yearly mean (determined by using a statistical T-test with a significance level of 0.01)."

reviewer comment: Page 11, line 15: Providing standard deviations would help in discussing the lack of scatter in the data.

answer: the standard deviations are added now.

reviewer comment: Page 13, line 4-5: Organic carbon was not measured, can the authors comment on the possible contribution to mass at the site? Has it been measured during previous studies?

answer: Water soluble organic carbon (WSOC) was measured on filter samples during the austral summer in 2011 at our measurement site. This analysis showed that that the WSOC mass fraction was on average less than 3% of the total mass and therefore we believe that excluding organic carbon from the chemistry based calculation does not influence significantly the results. This was added to the text as well: " We do not have any information on the organic carbon mass fraction for our measurement period, and therefore we could not include this component into the calculation. However, previous water soluble organic carbon (WSOC) mass concentration measurements (Weller et al. 2015) showed, that in the austral summer of 2011 the WSOC average mass fraction was less than 3% and therefore we believe that organic carbon does not have a significant influence on the resulting RI. "

reviewer comment: Page 13: line 14: Typically, thermodynamics favor the formation of ammonium sulfate before ammonium nitrate, such that if there is enough ammonium available, it will neutralize sulfate before nitrate. What is the molar ratio of NH4/SO4 during the study? Were the aerosol acidic?

[Figure]

answer: The reviewer's objection is correct. The nss-SO42-/NH4+ ratio is 11.2±8 (annual mean ± Stdev; the summer ratio is significantly higher) and we can generally assume acidic aerosol (at least during summer). Thus formation of ammoniumsulphate ((NH4)2SO4) is more plausible. However, given the high nss-SO42- excess (corresponding to a high H2SO4 excess), partly ammoniumbisulphate (NH4HSO4) may also be formed. The RI of NH4HSO4 is lower (Chálek et al., 1995) than that of NH4NO3 (1.473 instead of 1.52), but the latter would be comparable to that of (NH4)2SO4 (1.53; Tang, 1996). Unfortunately, in our case such a detailed chemical characterization is not possible, by neither our bulk aerosol nor our size segregated aerosol measurements. We considered this in the revised version of the manuscript (see revised Section 3.5).

reviewer comment: Page 14, line 15: The RIeff values look somewhat lower with East winds. It would help to expand the scale in Figure 6 to values ranging from 1.35-1.55 to sees these differences more clearly. Are these differences greater than uncertainty in the values themselves?

answer: The figure was expanded. Using the same significance test as used before showed us that the mean values are not significantly different from the yearly mean.

reviewer comment: Page 15. Line 3: The values in Figure 6 refer to an average value of all RIeff from the North. Did any individual distributions suggest contamination?

answer: No, no obvious contamination could be identified among these scans. Such scans when there was a known contamination present, such as the scan in figure 8, was removed from the data analysis.

reviewer comment: Page 16, line 12: Please provide units for scattering coefficient and a more detailed description of Cs. It must include a diameter-squared parameter.

answer: Units are provided. The scattering cross section is a well-known physical property, it has a diameter-squared unit, as it is now included in the text as well. The authors do not know what detail about Cs should be included additionally. We included

the method of calculation now in the text as well (our custom written Mie Code).

reviewer comment: Page 16, line 14. Is the scattering distribution in Figure 9 an average? Is the bimodal distribution is a function of the averaging of several different monomodal distributions at different times (suggesting interesting changes in the aerosol size distribution or refractive index).

answer: Yes, it is an average. The sentence of Line 14 states it: "Figure 9 shows the time average of dsigma/dlogD as function of the particle diameter" Both modes of the bimodal distributions (of both scattering coefficient and the number size distribution) are most of the time present simultaneously. It was also added to the text: "Considering the time evolution and not temporal averages we see, that these two peaks, as well as the two main peaks of the scattering coefficient size distribution, are often present simultaneously. In conclusion, the bimodality is not the product of time averaging of single modes appearing at different times."

reviewer comment: Page 17: Figure 9 caption: What time period do these data correspond to? Is this an average of several distributions?

answer: Yes. Now, the figure caption states as well, that the data is an average.

reviewer comment: Page 17, line 9: This sentence is unclear

answer: The sentence: "The resulted RI values will describe the particles with the particle sizes of the corresponding size range." was changed to: "Moreover, the overall size distribution range can now be divided into 4 subranges suitable for separate RIeff calculations, representative for the corresponding subrange (Fig 10.)."

reviewer comment: Page 17, line 15-18: As Figure 2 shows, the instrument response at higher sizes shows a cross over region such that the instrument is unable to distinguish between refractive indices. Others have also shown this behavior at larger sizes (Garvey and Pinnick, 1983; Hand et al., 2000). Reporting refractive indices in these larger size ranges is probably not meaningful.

answer: Yes, the reviewer is right about the cross over region. As it can be seen in Figure 2, the m=1.4 and m=1.59 lines meet around log(D)=3 which is D=1000nm. Our highest diameter range for the size dependency investigation was 478-710nm. Even at the largest diameter of 710nm (log(710nm)=2.85) the two example instrument response curves are still very well separated from each other (almost at the orange arrow in Figure 2). Therefore we think that the RI calculation even at our highest diameter range is meaningful. As the request of Reviewer #2, we removed this highest size range anyhow for another reason.

reviewer comment: Page 18, line 15: I am not sure what the authors mean by "geographic borders of this value's validity"?

answer: the geographic border was used to express, to question where exactly can a single (season independent) RI value of 1.44 be used. Is it only our measurement site? Or is it also valid 50 km away? May it be valid for other coastal Antarctic sites? Or maybe even everywhere in Antarctica?

The text was changed to: "Based on this, we recommend this single, temporally constant refractive index value for modeling of aerosol optical properties. In this context we suggest supporting investigations to examine the validity of this approach and the usage of seasonal independent RIeff values for the Antarctic region."

reviewer comment: Page 18, line 14. Including experimental uncertainties here would help, as would restating the RI derived from composition data. Deriving Reff and the ability to calculate scattering coefficients using it and the measured size distributions, without the additional effort and cost of composition measurements, is an important benefit to this analysis. Comparing the RI derived from composition in the context of experimental uncertainty can strengthen the arguments for the importance of this type of analysis. In addition, composition measurements are usually unavailable with the time resolution of size distribution measurements.

answer: The standard deviation of the RIeff and the chemical composition RI is now

included in the conclusions.

reviewer comment: Page 19: Line 1: It would help to state the seasonal variability more strongly if the readers could comment whether the seasonal values or wind direction values were greater than the experimental uncertainty. As it stands, it appears somewhat subjective.

answer: The text now reads: "In spite of the strong seasonal variability of the chemical composition at the measurement site (e.g. 86% sea-salt present in winter, 50% in summer) ..."

reviewer comment: General comment: Please provide wavelength and RH on each of the figure and table captions- it can help the reader quickly orient themselves without having to scroll back through the text.

answer: The values were provided.

References Chylek, P., and J. Wong, Effect of absorbing aerosols on global radiation budget, Geophysical Research Letters, 22 (8), 929-931, 1995. Tang, I.N., Chemical and size effects of hygroscopic aerosols on light scattering coefficients, Journal of Geophysical Research, 101 (D14), 19,245-19,250, 1996.  

---

## Author Comment (AC3) · 10 Sep 2019

reviewer comment: The authors report on aerosol refractive index observations but never mentioned that the index of refraction is a complex number. Particularly, the imaginary part of the refractive index constitutes the light-absorbing properties of the sampled aerosol. As Weller et al. (2013) reported, there is a small but significant fraction of lights-absorbing material contained in the aerosol in Antarctica. However, the authors never refer to this observation in a quantitative manner, nor they stated the assumption of a zero imaginary part of the refractive index. Furthermore, the scattering cross-section as calculated by Mie or Rayleigh-Debye-Gans theories depends on the square of the complex refractive index which includes the imaginary part. I request a discussion of the uncertainties in calculating the real part of the refractive index, when

neglecting the imaginary part. The effect may be small but it should be mentioned since the imaginary part plays a crucial role in the aerosol radiation interaction.

answer: First of all we would like to thank anonymous referee #3 for his/her helpful comments. We agree about the importance of the imaginary part of the refractive index and the light absorption. However, we do not agree, that we did not mention the assumption of a zero imaginary part of the refractive index: Page 8, Line 7-9: " The imaginary part of the RI was kept at 0 which is an acceptable assumption considering that the absorption is very low compared to the scattering at our measurement site, average single scattering albedo at Neumayer is 0.992 (Weller et al., 2013)." Page 8, Line 16-17: "The other assumption we use is that the aerosol particles are spherical and that the imaginary part of the RI is negligible." Page 13, Line 17-18: "The imaginary part of the RI was again neglected, which is surely a justified assumption, because the volume fraction of the BC never exceeded 0.1% in 2017." But, we agree, that a thorough discussion on the effect of the neglected imaginary part of the RI improves the manuscript. And therefore, we modified the text as follows: Abstract "Given the high average scattering albedo of 0.992 (Weller et al. 2013), we assumed that the imaginary part of the RI is zero." Section 3.5 "The imaginary part of the RI was again neglected, which is a justified assumption, because the volume fraction of the eBC never exceeded 0.1% in 2017. This amout of eBC would add at most a $\sim$4x10-3i imaginary value of the RI." Section 3.7 "Finally we investigate the effect of neglecting the imaginary part of the RI on the scattering coefficient. As we have seen in Section 3.5 including the eBC in the chemical composition adds at most an imaginary part of $\sim$4x10$^{\wedge}$-3i to the RI. We recalculated the average scattering coefficient size distribution adding this imaginary part to the RI. This gives us a highest possible estimate on the error we make if we would neglect the imaginary part of RI. It turns out that the relative difference of the scattering coefficient size distribution considering 4x10$^{\wedge}$-3i RI instead of 0.0i never exceeds 1.7% irrespective of the particle diameter."

reviewer comment: Figure 3: I assume that the dashed green line refers to the LAS

uncorrected best fit, please add.

answer: Yes. The figure was corrected.

reviewer comment: Figure 10: I propose to specify LAS original as LAS (m = 1.59); the term "original" suggests that data were modified, which is, however, not the case.

answer: Thanks for the good suggestion! It was adopted for the other figures as well.

reviewer comment: When reporting on the black carbon mass concentration determined by the MAAP, the authors should use the today accepted terminology of "equivalent black carbon" (eBC); see Petzold et al. (2013).

answer: The terminology was adopted in the text, and the reference was added.

---

## Author Comment (AC4) · 10 Sep 2019

Thank you Joel for the comment! It was made clear in the abstract, that we assumed a zero imaginary part of the RI and calculated only the real part. Since we only calculate the real part we do not think that it is needed to add a +0i to every value. We have added the single scattering albedo value to the abstract as well.

"Given the high average scattering albedo of 0.992 (Weller et al. 2013), we assumed that the imaginary part of the RI is zero."

---

## Author Response (AR1)

**Response to Reviewer 1:**

The authors would like to thank Anonymous Referee #1 who helped us improve our manuscript.

According to the comments, we reconsidered and rectified our manuscript (ms). Below, we give a point to point reply. For convenience and to avoid an unnecessary inflation of this response letter, all corresponding changes in our revised ms are accordingly indicated. We refrain from listing all revised marginal fragments here. However, we present all essentially revised parts of the ms straight below our response (*"in quotation marks and in italics"*).

*reviewer comment:* The paper could be strengthened by including some discussion of the implications of this work towards climate change estimates, which is one of their stated motivations for the study.

*answer:* Our derived refractive index values refer to dry aerosol particles, as they are derived from dry aerosol number size distribution measurements. Unfortunately, we are not able to extend the discussion towards climate change estimates, because we have no sufficient information on the water uptake (hygroscopicity) of the aerosol particles from our measurement site yet. Therefore, we could only speculate on the real ambient light scattering and absorption coefficients, which we do not want. We hope that the planned hygroscopicity measurements this austral summer (2019-2020) at our measurement site will allow us to make such a study.

*reviewer comment:* There were several typographical errors in the text that I did not attempt to correct; I suggest another careful editing of the manuscript.

answer: A careful editing and inspection by an native speaker was performed.

*reviewer comment:* Page 1, Line 1-3: This sentence is a bit unclear. I think the authors are trying to state that reducing uncertainties in modeling evaluations of climate change require more accurate aerosol optical properties. It might also help to point out why it is important to have measurements of refractive index at the poles- is this because climate change is enhanced there relative to other locations? It would also help to state when these measurements were made earlier in the abstract.

answer: The sentence was rewritten, now it reads:

"Though the environmental conditions of the Weddell Sea region and Dronning Maud Land (DML) are still relatively stable compared to the fast-changing Antarctic Peninsula, we may suspect pronounced effects of global climate change for the near future (Thompson et al. 2011). Reducing the uncertainties in climate change modeling requires inter alia a better understanding of the aerosol optical properties, and for this, we need accurate data on the aerosol refractive index (RI). Due to the remoteness of Antarctica only very few RI data are available from this region (Hogan et al. 1979, Virkkula et al. 2006, Shepherd et al. 2018).

*reviewer comment:* Page 1, line 7: Do the authors refer to 2439 individual size distributions when they refer to measurement points?

answer: We referred to 2439 averaged size distributions, but it was removed now from the manuscript.

*reviewer comment:* Page 1, line 7: It is always helpful to also include wavelength and relative humidity conditions associated with optical property measurements. Sometimes people only read abstracts and figures.

answer: The wavelength and the RH condition are added into the abstract.

*reviewer comment:* Page 1, line 8: It would also help to include some uncertainty estimates or an estimate of standard deviation with the reported average RIeff.

answer: The standard deviation value of RIeff was added to the abstract.

*reviewer comment:* Page 1, line 13-19: This paragraph is a bit unclear. I understood it better after I had read the paper, but as part of the abstract it could benefit from clarification. Part of the issue is that the location of the site has not been described yet, so understanding the wind direction in respect to the site location and the Neumayer station is a bit confusing. It might be helpful to state the wind direction impact in a more generalized way, or provide more description of the site location first.

**answer: We revised this paragraph:**

"We find no significant dependence of the derived RIeff values on the wind direction. Thus, we conclude that RIeff is largely independent on the general weather situation, roughly classified in (i) advection of marine boundary layer air masses during easterly winds caused by passing cyclones in contrast to (ii) air mass transport from continental Antarctica under southern katabatic winds. Neumayer, the only relevant contamination source, is located 1.5 km north of the air chemistry observatory, where the measurements were performed. Given that northerly winds are almost absent, the potential impact of local contamination is minimized in general. Indeed our data show no impact of local contamination on RIeff. Just in one case, a temporary high contamination episode with diesel engines operating right next to the measurement site resulted in an unusual high RIeff of 1.59, probably caused by the high black carbon content of the exhaust fumes."

*reviewer comment:* Page 1, line 20: Referring to the time-averaging here is also confusing without having read the paper first. It may not be necessary to include in the abstract. Are these differences larger than the uncertainties of the measurements?

answer: We now remove this confusing sentence from the abstract.

*reviewer comment:* Page 1, line 20: It would also help here to state something about the larger implications of this work, trying back to the point of the study so that the reader grasps the larger importance of the work.

answer: To this end we added the following text to the abstract:

"To conclude, our study revealed largely constant  $Rl_{eff}$  values throughout the year without any sign of seasonality. Therefore, it seems reasonable to use a single, constant  $Rl_{eff}$  value of 1.44 for modeling optical properties of natural, coastal Antarctic sub- $\mu$ m aerosol."

*reviewer comment*: Page 2, line 14: Probably the most common method would be from volumeweighted calculations of composition data because these are generally more available than detailed size distribution and scattering measurements.

answer: The reviewer is right, "The most common method" was changed to "A common method"

*reviewer comment:* Page 2, line 24: It is important to point out that these methods do not directly measure refractive index- they use closure studies between a variety of measurements to determine the refractive index that provides agreement.

answer: The following sentence was added to the manuscript:

"We have to keep in mind that all above mentioned methods are not direct measurements of the RI. All of these methods search for RI values that provide good agreement in a closure study between different measured quantities."

reviewer comment: Page 2, line 25: Remove "parallel their" for clarity.

answer: We removed this term in the revised manuscript.

reviewer comment: Page 2, line 34: What does it mean, "Until the particles disappeared"?

answer: We changed the text to:

"until they could not see the particles in the microscope (i.e. until the applied oil's RI matched the RI of the collected particles)."

reviewer comment: Page 3, line 5: To what RH do these values correspond?

answer: All these values refer to dry aerosol, thus we modified the text to:

"In this paper we would like to present continuous data on the real RI at 633nm wavelength of the dry ambient aerosol as derived..."

*reviewer comment:* Page 3, line 5: It would also be helpful here to point out the importance of this work- why is it important to have yearlong estimates of RI from the Antarctic? What are the larger implications?

answer: The following text was added to clear this point:

"With this, our study aims at better understanding of the aerosol optical properties at a place where only very few such data are available with special focus on its temporal variability. Given the distinct seasonality of the aerosol composition (see Weller et al., 2008, Figs. 4 and 5 therein), we may likewise expect a seasonality of RI. To this end, continuous year-round data of RI are necessary, in particular regarding the lack of such measurements for the Antarctic realm."

*reviewer comment:* Page 3, line 10: Include the study time period earlier, it will help when considering the information provided in the next paragraph.

answer: We modified this sentence to:

"The measurements presented in this paper were performed in the Air Chemistry Observatory (SPUSO from "Spurenstoffobservatorium") of the German Antarctic station of Neumayer III between February 2017 and January 2018."

*reviewer comment:* Page 3, line 31: Is RH and temperature measured? Define RH if it has not already been done.

*answer:* The RH of the aerosol entering the instruments was calculated from the measured outdoor temperature, RH and the measured indoor temperature. To clarify this point, we added a definition for RH and the following sentence to the text:

"The meteorological data used in this study (temperature, wind direction and speed and ambient RH) was measured directly on the roof of SPUSO."

reviewer comment: Page 5, line 8: How was 'penetration efficiency' determined?

*answer:* It is the ratio between the measured number size distribution with and without the tubing which causes the losses. We added the explanation to the text and renamed it to particle transmission efficiency.

reviewer comment: Page 6: Was there a quantitative measure by which "too noisy" was defined?

*answer:* We did not have a real quantitative measure, just the fact that no successful fit was possible, which became possible with further averaging.

reviewer comment: Page 6, line 7-8: This sentence is confusing, I suggest rewording.

answer: We reworded this confusing sentence and now it reads:

"This strong seasonal variability is the reason why in summer a much shorter time averaging period is sufficient to enable a successful RIeff fit. To keep the highest possible time resolution of the derived RI, we have chosen the length of the time averaging to be time dependent. And this length was determined by the actual particle concentration."

*reviewer comment:* Page 6, line 10: How was "good enough signal to noise ratio" defined? *answer:* Such that a meaningful RIeff fit was possible. We now added this note to the text.

*reviewer comment:* Page 11, line 10: How long was the averaging time? I am not sure why rapidly changing aerosol should results in poor fit if both instruments are sampling the same aerosol at the same time? Was the aerosol changing faster than the SMPS could sample it? Did CPC data indicate this?

*answer:* The averaging time was variable, one hour at minimum, but up to 14 hours occasionally. Anyway, this is not the decisive point. The problem is if the aerosol changes significantly within one scan, which took 6 minutes for each instrument. The two instruments measure the number size distribution differently. The LAS captures all of the particles within this 6-minutes period, and therefore delivers an average number size distribution for this period. The SMPS on the other hand scans through the different diameters during 6 minutes and therefore captures only one size range/bin at a given time interval. More precisely: if for instance during the 6 minutes of a single scan the aerosol concentration increases appreciably but the shape of the distribution remains constant, the LAS returns the average number size distribution. In contrast, the SMPS returns a skewed number size distribution with too low concentrations in the lower, but too high concentrations in the upper size range (assuming that the SMPS is operated in an up-scan mode). We could also observe in the CPC data, when during the beginning of the scan the concentration was significantly different compared to the end of the scan. Accordingly, we changed the text to:

"The reason why the fit did not work in this case was that the aerosol population was significantly changing within the duration of the SMPS scan. During the first half of the scan an aerosol plume with very high concentration reached the instruments. This appears in the SMPS scan as a very high fraction

of small particles, because during the first half of the scan, the SMPS selected and measured the smaller particles. Contrary, the LAS captures all particles with different diameters at the same time, and therefore this event appears as an elevated overall concentration. This was an extreme and exceptional situation where some unavoidable construction was done around the SPUSO using machines powered by diesel engines."

*reviewer comment:* Page 11, line 20: Based on the stated estimated uncertainty (~3%), reporting this many significant digits seems unnecessary. This comment also holds for Table 1 and reporting of values throughout the paper. It would also help to report standard deviation for each month in Table 1

*answer:* We removed insignificant digits and supplied the corresponding standard deviation in the revised Table 1.

*reviewer comment:* Page 12, line 11: How was "significantly different from the yearly mean" determined?

*answer:*We determined the significance by a statistical T-test with a significance level of 0.01. The revised text now reads:

"These values are significantly different from the yearly mean (determined by using a statistical T-test with a significance level of 0.01)."

*reviewer comment:* Page 11, line 15: Providing standard deviations would help in discussing the lack of scatter in the data.

Answer: We added standard deviations in the revised text.

*reviewer comment:* Page 13, line 4-5: Organic carbon was not measured, can the authors comment on the possible contribution to mass at the site? Has it been measured during previous studies?

*answer:* Water soluble organic carbon (WSOC) was measured on filter samples during the austral summer in 2011 at our measurement site. This analysis revealed that the WSOC mass fraction (except methane sulphonate!) was on average less than 3% of the total mass and therefore we believe that excluding organic carbon from the chemistry based calculation does not influence significantly the results. We added this finding in the revised version:

"We do not have any information on the organic carbon mass fraction for our measurement period, and therefore we could not include this component into the calculation. However, previous water soluble organic carbon (WSOC) mass concentration measurements (Weller et al. 2015) showed, that in the austral summer of 2011 the WSOC average mass fraction was less than 3% and therefore we believe that organic carbon does not have a significant influence on the resulting RI. "

*reviewer comment:* Page 13: line 14: Typically, thermodynamics favor the formation of ammonium sulfate before ammonium nitrate, such that if there is enough ammonium available, it will neutralize sulfate before nitrate. What is the molar ratio of NH4/SO4 during the study? Were the aerosol acidic?

*answer*: The reviewer's objection is correct. The nss-SO42-/NH4+ ratio is 11.2±8 (annual mean ± Stdev; the summer ratio is significantly higher) and we can generally assume acidic aerosol (at least during summer). Thus formation of ammoniumsulphate ((NH4)2SO4) is more plausible. However, given the high nss-SO42- excess (corresponding to a high H2SO4 excess), partly ammoniumbisulphate (NH4HSO4) may also be formed. The RI of NH4HSO4 is lower (Chýlek et al., 1995) than that of NH4NO3 (1.473 instead of 1.52), but the latter would be comparable to that of (NH4)2SO4 (1.53; Tang, 1996). Unfortunately, in

our case such a detailed chemical characterization is not possible, by neither our bulk aerosol nor our size segregated aerosol measurements. We considered this in the revised version of the manuscript (see revised Section 3.5).

*reviewer comment:* Page 14, line 15: The Rleff values look somewhat lower with East winds. It would help to expand the scale in Figure 6 to values ranging from 1.35-1.55 to sees these differences more clearly. Are these differences greater than uncertainty in the values themselves?

*answer:* We expanded the scale appropriately. Using the same significance test as used before showed us that the mean values are not significantly different from the yearly mean.

*reviewer comment:* Page 15. Line 3: The values in Figure 6 refer to an average value of all RIeff from the North. Did any individual distributions suggest contamination?

*answer:* No obvious contamination could be identified among these scans. If contamination was recognized (such as the scan in Figure 8), we removed those data from further evaluation.

*reviewer comment:* Page 16, line 12: Please provide units for scattering coefficient and a more detailed description of Cs. It must include a diameter-squared parameter.

*answer:* We provided units in the revised version. The scattering cross section has a diameter-squared unit, as it is now included in the text as well. The authors do not know what detail about Cs should be included additionally, because Cs is a well-known quantity. However, we mentioned the calculation method now in the text as well (i.e. a custom written Mie Code).

*reviewer comment:* Page 16, line 14. Is the scattering distribution in Figure 9 an average? Is the bimodal distribution is a function of the averaging of several different monomodal distributions at different times (suggesting interesting changes in the aerosol size distribution or refractive index).

*answer:* Yes, it is an average value, see line 14: "Figure 9 shows the time average of  $d\sigma/dlogD$  as function of the particle diameter". Both modes of the bimodal distributions (of both scattering coefficient and the number size distribution) are most of the time present simultaneously. In addition, we added to the main text:

"Considering the time evolution and not temporal averages we see, that these two peaks, as well as the two main peaks of the scattering coefficient size distribution, are often present simultaneously. In conclusion, the bimodality is not the product of time averaging of single modes appearing at different times."

*reviewer comment:* Page 17: Figure 9 caption: What time period do these data correspond to? Is this an average of several distributions?

answer: Yes, it is an average of several distributions which is now stated in the figure caption.

reviewer comment: Page 17, line 9: This sentence is unclear

answer: Accordingly we changed the confusing sentence to:

"Moreover, the overall size distribution range can now be divided into 4 subranges suitable for separate RIeff calculations, representative for the corresponding subrange (Fig 10.)."

*reviewer comment:* Page 17, line 15-18: As Figure 2 shows, the instrument response at higher sizes shows a cross over region such that the instrument is unable to distinguish between refractive indices. Others have also shown this behavior at larger sizes (Garvey and Pinnick, 1983; Hand et al., 2000). Reporting refractive indices in these larger size ranges is probably not meaningful.

answer: Yes, the reviewer is right about the cross over region. As it can be seen in Figure 2, the m=1.4 and m=1.59 traces meet around log(D)=3 which is D=1000 nm. Our widest diameter range for size dependency investigations was 478-710 nm. Even at the largest diameter of 710 nm (log(710nm)=2.85) the two example instrument response curves are still very well separated from each other (almost at the orange arrow in Figure 2). Therefore, we think that the RI calculation is meaningful even at our highest diameter range. As the request of Reviewer #2, we removed this highest size range anyhow for another reason.

*reviewer comment:* Page 18, line 15: I am not sure what the authors mean by "geographic borders of this value's validity"?

*answer:* We used the admittedly unclear notion "geographic border" to implicitly formulate the question: Where exactly can a single (seasonal independent) RI value of 1.44 be used? Is it valid only for our measurement site or also for other coastal or even continental Antarctic sites? We clarified this passage now to:

"Based on this, we recommend this single, temporally constant refractive index value for modeling of aerosol optical properties. In this context we suggest supporting investigations to examine the validity of this approach and the usage of seasonal independent RIeff values for the Antarctic region."

*reviewer comment:* Page 18, line 14. Including experimental uncertainties here would help, as would restating the RI derived from composition data. Deriving Reff and the ability to calculate scattering coefficients using it and the measured size distributions, without the additional effort and cost of composition measurements, is an important benefit to this analysis. Comparing the RI derived from composition in the context of experimental uncertainty can strengthen the arguments for the importance of this type of analysis. In addition, composition measurements are usually unavailable with the time resolution of size distribution measurements.

*answer:* we include now the standard deviation of the  $RI_{eff}$  and the chemical composition RI in the conclusions.

*reviewer comment:* Page 19: Line 1: It would help to state the seasonal variability more strongly if the readers could comment whether the seasonal values or wind direction values were greater than the experimental uncertainty. As it stands, it appears somewhat subjective.

*answer*: The text now reads: "In spite of the strong seasonal variability of the chemical composition at the measurement site (e.g. 86% sea-salt present in winter, 50% in summer) ..."

*reviewer comment:* General comment: Please provide wavelength and RH on each of the figure and table captions- it can help the reader quickly orient themselves without having to scroll back through the text.

answer: We provide now wavelength and RH throughout.

**References**

Chylek, P., and J. Wong, Effect of absorbing aerosols on global radiation budget, Geophysical Research Letters, 22 (8), 929-931, 1995.

Tang, I.N., Chemical and size effects of hygroscopic aerosols on light scattering coefficients, Journal of Geophysical Research, 101 (D14), 19,245-19,250, 1996.

**Response to Reviewer 2:**

The authors would like to thank Anonymous Referee #2 for their helpful comments and suggestions.

According to the comments, we reconsidered and rectified our manuscript (ms). Below, we give a point to point reply. For convenience and to avoid an unnecessary inflation of this response letter, all corresponding changes in our revised ms are accordingly indicated. We refrain from listing all revised marginal fragments here. However, we present all essentially revised parts of the ms straight below our response (*"in quotation marks and in italics"*).

*reviewer comment:* My main concern is that, in some parts more than in others, the language needs polishing, beyond what can be expected to be caught during the ACP-language editing at the end of the publication process. I will not list all these occurrences where the English has to be approved, but give at least an already longish list in this review at "Technical comments".

answer: A thorough language editing of the text was performed assisted by a native speaker.

*reviewer comment*: page 5, line 5: Information on where exactly particles entered the tubing during these experiments would be good. Just underneath the roof, close to the inlet line?

*answer:* For these experiments the instrument was repositioned and the inlet line where the instrument was connected before was closed and the tubing was removed from the inlet. The particles entered the tubing from somewhere middle of the measurement container. The following sentence was added to the page 5, line 4:

"The room air was measured by disconnecting the tubing from the inlet and sucking air from inside the measurement container."

*reviewer comment:* page 6, line 1-2: You show this polynomial only up to 400nm - although the data (blue dots) go up to 1000nm - does this mean you only used particles up to 400nm? Please add an explanation and/or prolong the line in Fig. 1.

*answer:* Yes, we used a polynomial fit only in the size range between 120 nm and 340 nm, as it was stated two lines before: "For the RI fit only this size range of the number size distribution was used." For clarification the text now reads:

"In the diameter range of the RI determination of 120–340 nm, the efficiency is between 0.77 and 0.67. The losses are significant here as well, but we consider this still as correctable. To have a continuous correction factor, the transmission efficiency (Fig. 1, blue dots) was fit within the diameter range of interest a polynomial line."

*reviewer comment:* page 6, line 24 ff: I have an idea what you did, here, but I am not entirely sure – this could certainly be formulated much clearer. What I think you did is the following: (1) - calculate TIR for a fixed RI (2) - take the value from the TIR at the diameter of the PSL particles. I guess one confusion was due to your use of the word "bin boundary diameter". Maybe this could be defined once and then "LAS diameter" could be used instead, throughout the text, to make the text flow better? Also, this passage sounds as if there would basically only be a signal in one bin during a PSL calibration - this is most likely not the case. Describe this more clearly.

answer: The revised text now reads:

"...(TIR, the signal which the instrument measures) of the LAS for both PSL particles (TIRPSL) and for particles with the desired RI (TIRRI) as function of the particle diameter".

"The LAS delivers the number size distribution (n(D)) as the particle number concentration (N(D)) sorted into diameter bins:  $n(D_i) = dN(D_i)/dlog(D_i)$ , where i denotes the ith diameter bin. These bins cover the whole measurement range of the instrument leaving no gaps. Each diameter bin has a lower and a higher boundary ( $D_{i,lower}$ ,  $D_{i,higher}$ ). These diameter bin boundaries correspond to the PSL calibration of the LAS. In order to recalculate the number size distribution to another RI, all bin boundary diameter has to be recalculated. This recalculation can be done by using the previously calculated TIR values: (1) For a single PSL calibration based bin diameter ( $D_{i,PSL}$ ) the instrument response TIRPSL( $D_{i,PSL}$ ) is looked up. (2) Now we look at the TIR values that are calculated for the desired RI. We search at which diameter ( $D_{i,Ri}$ ) we get the same instrument response as for PSL (TIR\_RI( $D_{i,RI}$ )=TIRPSL( $D_{i,PSL}$ )) and that diameter is the recalculated bin boundary diameter. We repeat this for every diameter bin."

"The diameter recalculation is not always straightforward, because OPCs using a monochromatic laser often suffer from a non-monotonic instrument response at higher diameters (e.g., Hodkinson and Greenfield, 1965; Barnard and Harrison, 1988). This problem of non-monotonic instrument response was solved by smoothing the calculated instrumental response function by fitting a 5th grade polynomial to the logarithm of both TIR\_PSL and TIR\_RI functions. Figure 2 shows an example how a single bin boundary diameter (D30\_PSL, the 30th diameter bin border) is recalculated using another (m=1.4+0i) RI."

*reviewer comment:* page 8, line 15-16: Concerning possible changes in particle composition: The way you did your derivation of RI, however, was to assume that the particle chemistry was the same for all particles in one measured size distribution? Please explicitly say this here somehow, as I got confused by your remark here.

*answer:* We did not assume the same chemical composition for all particles in one measured size distribution for the RI derivation! In contrast, we derive an RI which matches only the real aerosol RI if all the particles have the same chemical composition in the measured size distribution (actually only the same RI, particles with different chemical composition still might have the same RI). Otherwise, if the aerosol population is described with a single RI value, it is some kind of an average value. Since we derived the RI using only the number size distribution in the 120-340nm particle size range, the derived RI corresponds to this size range as well, and has no information on the particles with diameters outside of this size range. If the RI changes significantly with the size, our derived RI might not be equal to the average RI considering the whole aerosol population. To clarify this point, the text now reads:

"The RI derived with our method is representative for the size range of 120–340 nm, which was used for the RI calculation. If we can assume that all particles in the number size distribution have the same RI, our calculated RI is the true RI. If the chemical composition of the aerosol is changing with the particle size, it is possible that the RI is also size dependent. Hence, our derived RI might differ from the average RI that corresponds for the complete aerosol population. In addition, we assumed a spherical shape of the particles and a negligible imaginary part of the RI. Therefore we term our derived RI the effective refractive index (Rleff) from now on, and for later conclusions we have to keep in mind that the RIeff might not be the true RI of an individual particle."

*reviewer comment:* page 8, line 29-30: "We used the method introduced in the sections 2.5 and 2.6 to determine the RI of this e-cigarette smoke." But in the paragraph above you said that the RI of the cigarette smoke was 1.43, based on literature (and if you would have had to determine it first you would run into issues with circular reasoning if you then would use this measurement to calibrate the LAS TIR). I assume this again is an issue with formulating the text. Please review.

*answer:* In this section (as noted in the titel) we wanted to verify our RI calculation method and especially the particle loss correction. This is the reason why we searched for an aerosol source, which

has a known refractive index. If we now derive in the same way a RI value with our method and this RI agrees well with the literature value of the test aerosol (e-cigarette smoke in our case), then we know that our method (calculation and loss correction) works well. This is what we did here. We modified the text for clarification:

"We used the method introduced in the sections 2.5 and 2.6 to calculate the RI of this e-cigarette smoke, first with the uncorrected LAS data then with applying the above introduced (Section 2.3) LAS correction. These values can be compared to the e-cigarette smoke's literature RI value of 1.43 to check whether the LAS correction works well or not."

*reviewer comment:* page 9, line 4-6: Again confusing, so let me ask you again if this is what you did: When retrieving the RI for the uncorrected LAS data, you obtained an RI of 1.35, but when you corrected the measured LAS size distribution as described above and then retrieved the RI again, you got a value of 1.43, in agreement with literature. - If this is what you did, feel free to use my sentence here in the review instead of what you wrote. Your text here was hard to follow and it took me a while until I understood what you (likely) meant.

*answer:* Yes, this is exactly what we did. This point should be now clear after the following changes in the text:

"Without using the LAS correction on the LAS data (green lines) we get an RI of 1.35 from the best fit. This value is significantly lower than the literature RI value of 1.43 suggesting that the LAS losses had a high influence on the retrieved RI and that a correction is necessary. When we corrected the measured LAS size distribution as described above (Section 2.3), the best fit between the SMPS and the LAS data (blue lines) resulted in the RI of 1.43, which is in agreement with the literature value. This verifies our LAS correction, and we applied it on all LAS data before November 2017."

reviewer comment: page 12, first paragraph of 3.5: I would recommend to start this paragraph differently – the first sentence states something that seems not to hold once one read the list of RIs: when looking at this list and the most abundant components of the aerosol, one wonders if this really can be in good agreement, since particularly sea salt and ammonium sulphate are clearly above the value you retrieved. This all becomes much clearer further down, but I recommend to avoid confusion and to remove this first sentence or replace it with a sentence that says what you are aiming at in 3.5.

answer: The first part of the sentence was replaced:

"The aerosol chemical composition shows a strong seasonal variation at our measurement site. The dominant aerosol component is sea-salt with around 50% of the total mass in summer..."

*reviewer comment:* page 17, line 2: Maybe add that you expect this because scattering scales with the diameter squared

answer: We added this note:

".. as expected, because scattering increases faster than linearly as function of the particle diameter."

*reviewer comment:* page 18, first paragraph: You spend most of the space in this paragraph on discussing why this one value does not make sense, and the reason basically is that the underlying data is corrupted. Maybe just do not present the blue line in the figure and say up front that due to a) the strange kink in the LAS distribution and b) due to the low particle number concentration at the larger particle diameters no useful value resulted. (I'd be afraid that otherwise in the future someone might just grab that value from your figure without reading the text and use it.) Also, this lowering for

particles >~350 nm, together with the bimodality you showed in Fig. 9 - could this point towards two different sources for particles? This is something you could discuss here, instead.

answer: We now removed the blue line. We modified the paragraph accordingly:

"The conspicuously lower RIeff in the highest investigated size range may originate from a significantly changing chemical composition. Interestingly, sea-salt particles should dominate this higher size range, but this would result in a higher RIeff. Hence, one may speculate about a coating of sea-salt particles in this special case (probably organic material with typically lower RI). The presence of a coating or a different aerosol source might also explain the bimodality of the scattering coefficient size distribution (Section 3.8)."

*reviewer comment:* page 6, line 26: Do you really mean an OPC (i.e., a counter) or rather an OPS (optical particle sizer)? (Check this also in the introduction, line 19 on page 2).

*answer:* Optical particle counters (OPCs) are not only counting but also sizing the particles. They are just for some historical reason called counters. Optical particle sizer is more recent name for the same instrument. The older name, in our opinion, is better known and therefore we would like to leave it as it is.

reviewer comment: page 8, line 9-10: This again is a strangely formulated sentence.

answer: We changed the sentence:

"The Chi function was determined for every single m value, and the minimum of this function was searched. The m value, where Chi reaches its minimum is the m we look for and we interpret as the RI of the measured aerosol."

We especially thank the reviewer for numerous technical comments! We agree with all suggested corrections and modified the text accordingly throughout.

**Response to Reviewer 3:**

The authors would like to thank Anonymous Referee #3 for their helpful comments and suggestions.

According to the comments, we reconsidered and rectified our manuscript (ms). Below, we give a point to point reply. For convenience and to avoid an unnecessary inflation of this response letter, all corresponding changes in our revised ms are accordingly indicated. We refrain from listing all revised marginal fragments here. However, we present all essentially revised parts of the ms straight below our response ("*in quotation marks and in italics*").

*reviewer comment:* The authors report on aerosol refractive index observations but never mentioned that the index of refraction is a complex number. Particularly, the imaginary part of the refractive index constitutes the light-absorbing properties of the sampled aerosol. As Weller et al. (2013) reported, there is a small but significant fraction of lights-absorbing material contained in the aerosol in Antarctica. However, the authors never refer to this observation in a quantitative manner, nor they stated the assumption of a zero imaginary part of the refractive index. Furthermore, the scattering cross-section as calculated by Mie or Rayleigh-Debye-Gans theories depends on the square of the complex refractive index which includes the imaginary part. I request a discussion of the uncertainties in calculating the real part of the refractive index, when neglecting the imaginary part. The effect may be small but it should be mentioned since the imaginary part plays a crucial role in the aerosol radiation interaction.

*answer*: We agree about the importance of the imaginary part of the refractive index and the light absorption. However, we do not agree, that we did not mention the assumption of a zero imaginary part of the refractive index (see: Page 8, Line 7-9: "The imaginary part of the RI was kept at 0 which is an acceptable assumption considering that the absorption is very low compared to the scattering at our measurement site, average single scattering albedo at Neumayer is 0.992 (Weller et al., 2013). Page 8, Line 16-17: "The other assumption we use is that the aerosol particles are spherical and that the imaginary part of the RI is negligible." Page 13, Line 17-18: "The imaginary part of the RI was again neglected, which is surely a justified assumption, because the volume fraction of the BC never exceeded 0.1% in 2017."

Nevertheless, we agree, that a thorough discussion on the effect of the neglected imaginary part of the RI improves the manuscript. Accordingly, we modified and supplemented the text as follows:

Abstract

"Given the high average scattering albedo of 0.992 (Weller et al. 2013), we assumed that the imaginary part of the RI is zero."

**Section 3.5**

"The imaginary part of the RI was again neglected, which is a justified assumption, because the volume fraction of the eBC never exceeded 0.1% in 2017. This amount of eBC would add at most a ~4x10-3i imaginary value of the RI."

**Section 3.7**

"Finally we investigate the effect of neglecting the imaginary part of the RI on the scattering coefficient. As we have seen in Section 3.5 including the eBC in the chemical composition adds at most an imaginary part of  $^{4}x10^{-3}i$  to the RI. We recalculated the average scattering coefficient size distribution adding this imaginary part to the RI. This gives us a highest possible estimate on the error we make if we would neglect the imaginary part of RI. It turns out that the relative difference of the scattering coefficient size distribution considering 4x10-3i RI instead of 0.0i never exceeds 1.7% irrespective of the particle diameter."

*reviewer comment:* Figure 3: I assume that the dashed green line refers to the LAS uncorrected best fit, please add.

answer: Yes, the reviewer is right and we corrected the figure accordingly.

*reviewer comment:* Figure 10: I propose to specify LAS original as LAS (m = 1.59); the term "original" suggests that data were modified, which is, however, not the case.

answer: Thanks for this reasonable suggestion that we adopt now for this and the other figures as well.

*reviewer comment:* When reporting on the black carbon mass concentration determined by the MAAP, the authors should use the today accepted terminology of "equivalent black carbon" (eBC); see Petzold et al. (2013).

*answer:* Again, we adopted the reviewers suggestion in the revised version of our manuscript and included the reference.

**One year of aerosol refractive index measurement from a coastal Antarctic site**

**Zsófia Juránvi1,a and Rolf Weller1**

1Alfred-Wegener-Institut Helmholtz Zentrum für Polar- und Meeresforschung, Bremerhaven, Germany anow at: Institute for the Protection of Maritime Infrastructures, German Aerospace Center (DLR), Bremerhaven, Germany

Correspondence: Zsófia Jurányi (zsofia.juranyi@gmail.com)

**Abstract. Climate change model evaluations need**

Though the environmental conditions of the Weddell Sea region and Dronning Maud Land are still relatively stable compared to the fast-changing Antarctic Peninsula, we

- 5 may suspect pronounced effects of global climate change for the near future (Thompson et al., 2011). Reducing the uncertainties in climate change modeling requires inter alia a better understanding of the atmospheric aerosols' optical properties and with this of the aerosol optical 10 properties, and for this we need accurate data on the
- aerosol refractive index (RI)of atmospheric aerosols as well. Due to the remoteness of Antarctica only a very few data on the refractive index exists from there. In this paper we very few RI data are available from this region
- We calculate the real refractive index of natural atmospheric aerosols from number size distribution measurements at a coastal Antarctic measurement site. In our calculations we used the German coastal Antarctic station Neumayer
- 20 III. Given the high average scattering albedo of 0.992 (Weller et al., 2013), we assumed that the imaginary part of the RI is zero. Our method uses the overlapping size range (120-340 particle diameter D between 120 and 340 nm) of a scanning mobility particle sizer (SMPS), which sizes the
- 25 particles by their electrical mobility, and a laser aerosol spectrometer (LAS), which sizes the particles by their optical scattering signal at 633nm wavelength.

Based on almost a complete year of measurementand 2439 measurement points, the average effective refractive 30 index (RIeff), as we call our retrieved RI because of the used assumptions) for the dry aerosol particles turned out to be 1.44 . This is with a standard deviation of 0.08, in a good agreement with the RI value of 1.47, which we derived from the chemical composition filter of bulk

aerosol sampling measurements. At our measurement site 35 the aerosol has a very characteristic Neumayer the aerosol shows a pronounced seasonal pattern in both, number concentration and chemical composition. Despite this, we could not identify any significant seasonal variability in the the variability of the monthly averaged RIeff, the monthly 40 averages remain within the range of values remained between 1.40 -1.50. Two and 1.50. Compared to the annual mean, two austral winter months June and Septemberhas a slightly higher average (July and September) showed slightly but significantly increased values (1.50 and 1.47, respectively). 45 The size dependency of the RIeff could be determined from time averaged LAS and SMPS number size distributions measured between December 2017 and January 2018. Here  $_{15}$  (Hogan et al., 1979; Virkkula et al., 2006; Shepherd et al., 2018). we calculated RIeff for four different particle size ranges and observed a slight decrease from 1.47 (D range 116–168 nm) 50 to 1.37 (D range 346–478 nm).

> We could not identify any influence of the occurring wind direction on the retrieved find no significant dependence of the derived RIeff values on the wind direction. Thus we conclude that RIeff either. For the few examples of 55 north winds coming from the Neumayerstation (occurs very rarely, this is the reason why the measurement site was built to the south), we don't see different values than for the other wind directions. During an artificial, is largely independent on the general weather situation, 60 roughly classified in (i) advection of marine boundary layer air masses during easterly winds caused by passing cyclones in contrast to (ii) air mass transport from continental Antarctica under southern katabatic winds. Neumayer, the only relevant contamination source, is located 1.5 km north 65 of the air chemistry observatory, where the measurements were performed. Given that northerly winds are almost absent, the potential impact of local contamination is

minimized in general. Indeed our data show no impact of local contamination on RIeff. Just in one case a temporary high contamination episode, when diesel engines were operated with diesel engines operating right next to the mea-

- 5 surement site , we had an hour of constant conditions such that one RI fit was possible. This fit resulted in an unusual high RIeff. of 1.59, which is most probably due to probably caused by the high black carbon content of the diesel engine emission. Therefore, we also assume that even 10 during northerly wind directions we did not have significant
- influence from the Neumayer stationexhaust fumes.

During a shorter period between 2017 December and 2018 January we used the time averaged LAS and SMPS number size distributions to get some information on the

- 15 size dependency of the refractive index. The RITo conclude, our study revealed largely constant RIeff was fit in 5 different particle size ranges, and we have found a slight decrease of the values throughout the year without any sign of seasonality. Therefore, it seems reasonable to use
- 20 a single, constant RIeff with the particle size from 1.47 in the 116-168 to 1.37 in the 346-478 range value of 1.44 for modeling optical properties of natural, coastal Antarctic sub-µm aerosol.

**Introduction 1**

- 25 Atmospheric aerosols affect the radiative balance 2001): of planet Earth (e.g. Ramanathan et al., directly—Directly by absorbing and scattering the sunlight (Schwartz, 1996) (e.g. Schwartz, 1996) and indirectly through modifying the micro-30 physical properties of clouds the (Lohmann and Feichter, 2005)(e.g. Lohmann and Feichter, 2005)used by Bluvshtein et al. (2012) who introduced an RI re-The current state of the scientific knowledge on the total (direct and indirect) aerosol effect is still considered low due to the complexity of these effects (IPCC, 2014).
- The refractive index (RI) of the atmospheric aerosols is a key parameter calculating their absorption and scattering and therefore essential for the global modeling of the aerosol's radiative effects. Valenzuela et al. (2018) showed that there is still clearly a need for additional and accurate measurements
- 40 of the RI. There are more existing optical software packages for the optical properties of the atmospheric particulate matter and these packages extensively use RI values of the different kind of aerosols. The OPAC (Optical Properties of Aerosols and Clouds, Hess et al., 1998) package is
- 45 based on laboratory measurements, whereas the HITRAN-RI (HIgh-resolution TRANsmission Refractive Indices, Massie and Hervig, 2013) package uses both laboratory and field measurements for the different included components and allows comparisons between the products using the different

50 RIs as well. Valenzuela et al. (2018) showed us that there is

still clearly a need for additional and accurate measurements of the RI.

The most A common method to determine the RI of aerosol particles is an indirect method: the The measurement of the absorption and/or scattering of the particles along with 55 the knowledge of the particle's size. The absorption and the scattering of a single particle is determined by the particle's size, shape and RI. It is most often assumed that particles are spherical and for the theoretical calculations the Mie theory can be used. 60

Wex et al. (2009) determined the RI of secondary organic aerosol by selecting the particle size using a differential mobility analyser (DMA) and measuring the scattering signal using an optical particle counter (OPC). The same method was used by Hand and Kreidenweis (2002) 65 on ambient aerosol, additionally. Additionally they combined the measurements from an aerodynamic particle sizer as well, in order to gain information on the particles' density. Bukowiecki et al. (2011); Zhang et al. (2013); Zieger et al. (2015) used the number size distribution with parallel 70 nephelometer and aethalometer measurements to determine the RI of ambient aerosols. A very similar method was used by Virkkula et al. (2006), where it was assumed that for the Antarctic site Aboa, assuming that here the imaginary part of the RI can be neglected and therefore no absorption data was 75 used.

Barkey et al. (2007) measured laboratory generated particles' number size distribution and parallel their light scattering by a polar nephelometer. They introduced an inversion algorithm to obtain the RI. A new and more exotic method 80 is to use optical trapping combined with Mie spectroscopy to capture the RI of atmospheric aerosol samples in the 460-700 nm wavelength range by Shepherd et al. (2018). Cavity ring-down spectroscopy is a commonly used method to study the light extinction by aerosol particles. This method was 85 trieval method by measuring the light extinction at two carefully selected size parameters. We have to keep in mind that all above mentioned methods are not direct measurements of the RI. All of these methods search for RI values that 90 provide good agreement in a closure study between different measured quantities.

As we see there are plenty of existing measurements on aerosol RI aerosol RI measurements, but the majority of these measurements are based on laboratory generated par-95 ticles and only less few on ambient aerosols. And if we look for RI measurements from Antarctica we can only find very few available data. Hogan et al. (1979) collected aerosol particles at the South Pole in a size range between 0.3 and  $12 \,\mu m$ during a 4-days period and put oils with known different 100 RIs on them until the particles disappeared they could not see the particles in the microscope (i.e. until the applied oil's RI matched the RI of the collected particles). They have found an RI of 1.54 for these samples. Virkkula et al. (2006) derived the RI (assuming a zero imaginary part) of the am- 105

bient aerosol at coastal Antarctica during a 12-days summer campaign and got values around 1.43–1.44. Insoluble organic aerosol collected at the Clean Air Sector Laboratory of the British Antarctic Survey station Halley was analysed by Sherkard et al. (2018). The sector sector are able of the sector and the sector area able of the sector and the sector area able of the sector able of the sector area able of the

5 by Shepherd et al. (2018). The samples were They obtained a RI of 1.47 for samples collected on 60 consecutive days during the austral summer of 2015 and they got an RI of 1.47. austral summer 2015.

In this paper we would like to present continuous data 10 on the real RI at 633 nm wavelength of the dry ambient aerosol as derived from measurements of an optical particle counter and a scanning mobility particle sizer. To our knowledge this is the first time, that such long-term RI measurements of almost one year from Antarctica is presented.

[revised manuscript text omitted]

Thermo ESM AndersenScientific TM Model 5012) operating at a wavelength of 630637 nm (Petzold and Schönlinner, 2004) was used to measure the BC mass concentration aerosol absorption during the measurement campaign. The

- 5 absorption values were converted into BC equivalent black carbon (eBC, Petzold et al., 2013) mass concentration using a mass absorption efficiency of  $6.6 \,\mathrm{m^2g^{-1}}$ , and were registered also once in a per minute. The ionic composition of the aerosol is was measured by a low volume Teflon/Nylon
- 10 filter system, and the filters are analysed by ion chromatography. The filters were changed daily but not every day at the same time and therefore the time resolution of the ionic composition varies with the time. The average sampling flow was  $\approx 3.5 \,\mathrm{m^3 h^{-1}}$ , the sampled air volume var-
- $_{15}$  ied between 30 m3 and 125 m3 in 2017. The filter sampling is automatically switched off in case of a possible contamination (snow drift, wind coming from the Neumayer station, low windspeed, too high particle concentration or too high windspeednortherly wind direction, wind velocities
- 20 below  $2 \text{ m s}^{-1}$  or above  $20 \text{ m s}^{-1}$ , and exceedingly high particle number concentrations), see details in Weller et al. (2008). In this study we used the following main ionic species: NH4+, Na+ NO3-, non sea-salt SO42- and MSA- (methanesulphonate). The CPC , the MAAP and the filter
- 25 measurements and the MAAP are part of the continuous measurement program of GAW.

**2.3 Correction of the LAS losses**

We have collected data from both the LAS and SMPS instruments for almost one year (09.02.2017–20.01.2018). Unfor-30 tunately, during most of this time, the LAS was positioned horizontally too far away (ca. 3 m) from the inlet such - that significant amount of particles were lost in the connecting tube. This problem was first discovered in November 2017. Right after, on the 23.11.2017, the instrument was reposi-35 tioned right below the inlet in order to minimize the particle

- losses. We used the number size distribution data in the For this study, we were particularly interested in the particle diameter range between 120 and 340 nm because we used the number size distribution data in this diameter range for the
- 40 RI determination (see section 2.6)for the RI determination, therefore. Therefore, it was important to check whether or not we are able to correct for the particle losses before November 2017 in this diameter range.

Measuring the losses in the sampling line which was used 45 before November 2017 ("old" setup) was a challenging task. Our At our measurement site, the SPUSO, did not have any kind of particle generator no particle generator was available to perform tests with, and due to the location and isolation of the station, it was also impossible to receive any equipment

50 for the test. Our best option was to use the room air of the measurement container to quantify the losses. This particle source particle losses. The room air aerosol was measured by disconnecting the tubing from the inlet and sucking air from

Figure 1. The quantification of the LAS losses in the sampling line. The two orange lines belong refer to the right axis and show the average room air number size distributions. "Old" setup: time average with the long horizontal tube, "new" setup: time average without the horizontal tube. The blue dots show the penetration-particle transmission efficiency through the tube, the dashed dark blue line shows a polynomial fit in the diameter range which was used for the RI calculation.

inside the measurement container. The room air provided only a low concentration<del>such, that more</del>, so that several hours of measurement were needed. One measurement cycle included the number size distribution measurement with of the LAS of the room air aerosol in the "old" setup and right after removing the horizontal tube another measurement in the "new" setup, with the shorter, vertical tube. To make sure, that the aerosol source is stable enough during one cycle, the setup was changed every number size distribution measurement time was reduced to 2 times 60 s with some seconds in between to change between the setups.

All measured number size distributions were averaged 65 separately for the "old" and the "new" setups, and the average number size distributions were compared. Figure 1 shows the results of this average number size distribution comparison. If one looks at it them (Fig. 1, orange lines, right axis) or the penetration efficiency (at the particle 70 transmission efficiency (the ratio between the two size distributions, Fig. 1 - blue dots, left axis) it is obvious that the losses in the "old" sampling line are significant, almost . Almost all particles with diameters above 1 µm were lost, and therefore it is impossible to make any correction there. 75 This is the reason, why we only For this reason, we have the complete number size distribution until up to  $5 \,\mu m$  only after November 2017 for this study. However, in In the diameter range of the RI determination of 120-340 nm, the efficiency is between 0.77 and 0.67. The losses are significant here as 80 well, but we consider this still as correctable. To have a continuous correction factor, the transmission efficiency (Fig. 1, blue dots) was fit within the diameter range of interest with a

polynomial line. The blue dashed line shows this polynomial fit which was used for the correction.

**2.4 Time averaging**

Due to the low aerosol number concentration in Antarctica, 5 we performed a base time averaging of one hour of all measured data. This one hour averaging still often resulted in too noisy number size distributions, such that an RI fit was impossible. The particle number concentration at our measurement site has a strong seasonal variability with much lower

- 10 concentrations in winter than in summer. This is why we decided to perform on top of the one hour time averaging a particle concentration dependent time averaging as well in order to keep strong seasonal variability is the time resolution as high as possible reason why in summer a much shorter time
- 15 averaging period is sufficient to enable a successful RI fit. To keep the highest possible time resolution of the derived RI, we have chosen the length of the time averaging to be time dependent. And this length was determined by the actual particle concentration.
- 20 After performing many tests, we concluded found, that the one hour averaged SMPS number size distributions, that were recorded during a time period with an average number concentration of at least  $400 \text{ cm}^{-3}$ had a good enough , showed an adequate signal to noise ratio for the RI cal-
- 25 culation and no further averaging was needed. For all other cases with lower concentrations the hourly averaged data was further averaged until the number of the detected particles particles detected by the SMPS equaled or exceeded the particle number, which is detected during a one hour SMPS
  30 scan with at 400 cm-3 concentration. With this averaging method in particle concentration. In some extreme cases in winter, the measured data had to be averaged for 15 hours, whereas in summer most of the time the original one hour or maybe sometimes 2-hours averaging time was needed. Due
  35 to this averaging method we have the highest possible time
- resolution but it is though not constant, changing in timebut changing with time, depending on the total particle number concentration. This changing time resolution had to be taken into account for all further time average or statistical calcu-40 lations.

**2.5 Recalculation of the LAS number size distribution**

The LAS is factory calibrated using PSL particles having an RI of 1.588 (Eidhammer et al., 2008). In order to be able to recalculate the particle number size distribution to for any 45 other RI, we need to calculate the theoretical instrument response (TIR, the signal which the instrument measures) of the LAS for both PSL and the desired particle RI particles (TIRPSL) and for particles with the specified RI (TIRRI) as function of the particle diameter. This was done by a custom-50 written Mie code using the LAS wavelength of  $\lambda = 633$  nm and a detection angle  $\Theta$  between 22 and 158 degrees with a geometry of a round detector shape.

The LAS diameter bin boundaries corresponding delivers the number size distribution (n(D)) as the particle number concentration (N(D)) sorted into diameter bins: 55  $n(D_i) = \frac{dN(D_i)}{d\log(D_r)}$ , where i denotes the ith diameter bin. These bins cover the whole measurement range of the instrument leaving no gaps. Each diameter bin has a lower and a higher boundary (Di, lower, Di, higher). These bin boundaries correspond to the PSL calibration can be transformed to a diameter at the target RIby searching for the PSL calibration TIR value in of the LAS. In order to recalculate the number size distribution to another RI, all bin boundary diameter has to be recalculated. This recalculation can be done by using the previously calculated TIR values: 65 (1) For a single, PSL calibration based bin diameter  $(D_{i, PSL})$ the instrument response  $\text{TIR}_{\text{PSL}}(D_{i,\text{PSL}})$  is looked up. (2) Now we look at the TIR values calculated with the target Rland looking up the corresponding diameter. However, this problem that are calculated using the desired RI. We 70 search at which diameter  $(D_{i,RI})$  we get the same instrument response as for PSL:  $\text{TIR}_{\text{RI}}(D_{i,\text{RI}}) = \text{TIR}_{\text{PSL}}(D_{i,\text{PSL}})$  and that diameter is the recalculated bin boundary diameter. We repeat this for every diameter bin.

**Figure 2.** LAS Theoretical instrument responses for m = 1.588 + 0i(black) and 1.40+0i (orange) as function of the particle diameter. Here we show an example, how an original LAS diameter bin border (D30,RSL) was recalculated to the target RI (D30,RI).

The diameter recalculation is not always straight-forward, 75 because OPCs using a monochromatic laser often suffer from a non-monotonic instrument response at higher diameters (e.g., Hodkinson and Greenfield, 1965; Barnard and Harrison, 1988).

LAS Theoretical instrument responses for m = 1.588 + 0i 80 (black) and 1.40+0i (orange) as function of the particle diameter. Here we show an example, how an original LAS diameter bin border (D30) was recalculated to the target RI ( $D'_{30}$ ) is shown.

This problem of non-monotonic instrument response was solved by smoothing the calculated instrumental response function . The smoothing was done by fitting a 5th grade polynomial to the logarithm of both PSL and target RI 5 TIRTIRPSL and TIRRI functions. Figure 2 shows an example how a single bin boundary diameter ( $D_{3030,PSL}$ , the 30th diameter bin border) is recalculated using another RI (m = 1.4+0i)RI. The Mie calculation (solid line) and the polynomial fit (dashed line) are shown for both RIs. The 10 30th diameter bin border is 592 nm in our setup, using the original PSL calibration. One can read from Figure 2 that a PSL particle of this size detected by the LAS results in the same TIR as a particle with the RI of 1.4 and the size of  $D_{30,RI} = 723$  nm. The same procedure has to be used for 15 every bin boundary diameter and every desired index of refraction. After having the recalculated diameter borders, we can recalculate the number size distribution as well. If the

$$n_{\underline{\text{PSL}}}(D_{\underline{\text{PSL}}}) = \frac{dN(D)}{d\log(D)} \frac{dN(D_{\text{PSL}})}{d\log(D_{\text{PSL}})}$$
(1)

original number size distribution is:

20 Then the recalculated number size distribution looks like this:

$$n'_{\underline{\mathrm{RI}}}(D'_{\underline{\mathrm{RI}}}) = \frac{dN(D')}{d\log(D')} \frac{dN(D_{\mathrm{RI}})}{d\log(D_{\mathrm{RI}})} = \frac{dN(D')}{\log(D'_{\mathrm{high}}) - \log(D'_{\mathrm{low}})}$$
(2)

where  $\frac{D'_{\text{high}}}{D_{\text{bigh},\text{RL}}}$  is the upper and  $\frac{D'_{\text{low}}}{D_{\text{low},\text{RL}}}$  is the lower boundary of the recalculated diameter bin.

**25 2.6 Calculation of the effective refractive index**

In order to find the aerosol refractive index, the SMPS and the LAS data in the overlapping size range has to be matched. This matching was done by recalculating the LAS number size distribution using a set of different RIs and finding the 30 one which matches the best the SMPS number size distribution at the overlapping size range. The following expression was used after Khlystov et al. (2004) to quantise the difference between the LAS and the SMPS distribution:

$$\chi(m) = \frac{1}{N} \cdot \sum_{i=N_{\min}}^{N_{\max}} \left[ \log\left( n_{\text{SMPS}}\left( D_{i} \right) \right) - \log\left( n_{\text{LAS}}\left( m, D_{i} \right) \right) \right]^{2}$$
(3)

The SMPS and the LAS has an overlapping size range between 90 and 950 nm, however only the range between 120 and 340 nm was used for the fit<del>due to the very low particle concentration</del>. The SMPS number size distribution was too noisy over 340 nm and at the lowest diameters, the LAS does not have a detection efficiency of unity. The range of the RI 40 was chosen to be 1.3-1.8 with 0.01 steps in between. The imaginary part of the RI was kept at 0 which is an acceptable assumption considering that the absorption is very low compared to the scattering at our measurement site, with an average single scattering albedo at Neumayer is of 0.992 (Weller 45 et al., 2013). The  $\chi(m)$  function was determined for every single m value<del>and the aerosol RI is the</del>, and the minimum of this function was searched. That m value where the where  $\chi$  function reaches its minimum is the *m* value we look for and we interpret as the RI of the measured aerosol. Those 50 cases were omitted where the  $\chi$  function did not have an explicit minimum or exceeded a limit. After manual inspection of many fit procedures this limit was set to the value of 0.02. Such cases might occur if too much noise is present in the data or if the size distribution was varying too much during 55 the time period of one scan. Next to this numerical criterion every single scan was manually checked as well.

The RI derived with our method is representative for the used overlapping size particle diameter range of 120- $340 \,\mathrm{nm}$ , which was used for the RI calculation. If we can 60 assume that all particles in the number size distribution have the same RI, our calculated RI is the true RI. If the chemical composition of the aerosol is changing with the particle size, it is also possible that the RI is also size dependent. This we have to keep in mind for later conclusions. 65 The other assumption we use is that the aerosol particles are lopherical and they the offende, our derived RI might differ from the average RI which corresponds to the complete aerosol population. In addition we assumed a spherical shape of the particles and a negligible imaginary part of the RIis 70 negligible. Due to these assumptions we call the. Therefore we term our derived RI the effective refractive index (RIeff) from now on, and for later conclusions we have to keep in mind that the (RIeff) might not be the true RI of an individual particle. 75

**3 Results and discussion**

**3.1 Verification of the LAS correction**

In order to verify the used LAS correction (see Sec. 2.3), measurements measurement of particles with known RI and spherical shape was necessary. The lack of any particle generator left us with not many possibilities. A commercial ecigarette (Joytech eGo) was available at the station, and we used this to generate particles for the testing purpose. Ecigarette liquid contains glycerin, propylene glycol, water, nicotine and flavourings and the formed aerosol particles are spherical liquid droplets. Pratte et al. (2016) measured the RI of many e-cigarettes of different types and got values between 1.429 and 1.436, and therefore we assume that our generated test particles had an RI of 1.43.

---

## Author Response (AR2)

**Response to Referee #2**

The authors would like to thank again Anonymous Referee #2 for her/his helpful comments and suggestions. Below, we give a point to point reply.

- page 1, line 7: "inter alia" – please reconsider inserting these two words; I personally would suggest to remove them

We deleted the term "inter alia"

- page 1, line 9: add "atmospheric" before "aerosol"

We think, it is clear that atmospheric aerosol is meant here!

- page 3, line 93: exchange "sees" with "detected"

**Is replaced.**

- page 4, line 76-78: The sentence sounds awkward. I suggest an alternative formulation: "For this reason, the number size distribution up to 5  $\mu$ m is only available after November 2017."

We changed this part accordingly.

page 5, line 64: exchange "diameter has" with "diameters have"

**Is corrected.**

page 6, line 39: exchange "over" with "above"

**Is corrected.**

page 14, line 4: Explaining this low RI based on the existence of coated sea salt particles sounds quite far fetched, and I wonder if not also the statistics are quite bad in the sense that low particle number concentrations in the related size range cause high uncertainties. If you can agree with me on that, then please finish this chapter with a respective note.

[revised manuscript text omitted]
 100 µm is not available after between the size of the RI determination of 100 µm is not available.
- 30 120–340 nm, the efficiency is between 0.77 and 0.67. The losses are significant here as well, but we consider this still as correctable. To have a continuous correction factor, the

transmission efficiency (Fig. 1, blue dots) was fit within the diameter range of interest with a polynomial line. The blue dashed line shows this polynomial fit which was used for the 35 correction.

**2.4 Time averaging**

Due to the low aerosol number concentration in Antarctica, we performed a base time averaging of one hour of all measured data. This one hour averaging still often resulted in too noisy number size distributions, such that an RI fit was impossible. The particle number concentration at our measurement site has a strong seasonal variability with much lower concentrations in winter than in summer. This strong seasonal variability is the reason why in summer a much shorter time averaging period is sufficient to enable a successful RI fit. To keep the highest possible time resolution of the derived RI, we have chosen the length of the time averaging to be time dependent. And this length was determined by the actual particle concentration.

After performing many tests, we found, that the one hour averaged SMPS number size distributions, recorded during a time period with an average number concentration of at least  $400 \,\mathrm{cm}^{-3}$ , showed an adequate signal to noise ratio for the RI calculation and no further averaging was needed. For all 55 other cases with lower concentrations the hourly averaged data was further averaged until the number of particles detected by the SMPS equaled or exceeded the particle number, which is detected during a one hour SMPS scan at  $400 \,\mathrm{cm}^{-3}$ particle concentration. In some extreme cases in winter, the 60 measured data had to be averaged for 15 hours, whereas in summer most of the time the original one hour or sometimes 2-hours averaging time was needed. Due to this averaging method we have the highest possible time resolution though not constant, but changing with time, depending on the total 65 particle number concentration. This changing time resolution had to be taken into account for all further time average or statistical calculations.

**2.5 Recalculation of the LAS number size distribution**

The LAS is factory calibrated using PSL particles having an 70 RI of 1.588 (Eidhammer et al., 2008). In order to be able to recalculate the particle number size distribution for any other RI, we need to calculate the theoretical instrument response (TIR, the signal which the instrument measures) of the LAS for both PSL particles (TIRPSL) and for particles 75 with the specified RI (TIRRI) as function of the particle diameter. This was done by a custom-written Mie code using the LAS wavelength of  $\lambda = 633$  nm and a detection angle  $\Theta$  between 22 and 158 degrees with a geometry of a round detector shape.

The LAS delivers the number size distribution (n(D)) as the particle number concentration (N(D)) sorted into diameter bins:  $n(D_i) = \frac{dN(D_i)}{d\log(D_i)}$ , where i denotes the ith diameter bin. These bins cover the whole measurement range of the instrument leaving no gaps. Each diameter bin has a lower and a higher boundary ( $D_{i,\text{lower}}$ ,  $D_{i,\text{higher}}$ ). These bin boundaries correspond to the PSL calibration of the LAS. In or-

- 5 der to recalculate the number size distribution to another RI, all bin boundary diameters have to be recalculated. This recalculation can be done by using the previously calculated TIR values: (1) For a single, PSL calibration based bin diameter ( $D_{i,PSL}$ ) the instrument response TIRPSL( $D_{i,PSL}$ ) is
- 10 looked up. (2) Now we look at the TIR values that are calculated using the desired RI. We search at which diameter  $(D_{i,\text{RI}})$  we get the same instrument response as for PSL:  $\text{TIR}_{\text{RI}}(D_{i,\text{RI}}) = \text{TIR}_{\text{PSL}}(D_{i,\text{PSL}})$  and that diameter is the recalculated bin boundary diameter. We repeat this for every 15 diameter bin.

**Figure 2.** LAS Theoretical instrument responses for m = 1.588 + 0i (black) and 1.40+0i (orange) as function of the particle diameter. Here we show an example, how an original LAS diameter bin border (D30,PSL) was recalculated to the target RI ( $D_{30,PSL}$ ).

The diameter recalculation is not always straight-forward, because OPCs using a monochromatic laser often suffer from a non-monotonic instrument response at higher diameters (e.g., Hodkinson and Greenfield, 1965; Barnard and Harri-20 son, 1988). This problem of non-monotonic instrument response was solved by smoothing the calculated instrumental response function by fitting a  $5^{th}$  grade polynomial to the logarithm of both  $TIR_{PSL}$  and  $TIR_{RI}$  functions. Figure 2 shows an example how a single bin boundary diameter  $_{\rm ^{25}}$  (D\_{\rm 30,PSL}, the 30^{\rm th} diameter bin border) is recalculated using another RI (m = 1.4+0i). The Mie calculation (solid line) and the polynomial fit (dashed line) are shown for both RIs. The  $30^{\rm th}$  diameter bin border is  $592\,{\rm nm}$  in our setup, using the original PSL calibration. One can read from Figure 2 that 30 a PSL particle of this size detected by the LAS results in the same TIR as a particle with the RI of 1.4 and the size of  $D_{30,RI} = 723$ nm. The same procedure has to be used for every bin boundary diameter and every desired index of refraction. After having the recalculated diameter borders, we

can recalculate the number size distribution as well. If the 35 original number size distribution is:

$$n_{\rm PSL}(D_{\rm PSL}) = \frac{dN(D_{\rm PSL})}{d\log(D_{\rm PSL})} \tag{1}$$

Then the recalculated number size distribution looks like this:

$$n_{\rm RI}(D_{\rm RI}) = \frac{dN(D_{\rm RI})}{d\log(D_{\rm RI})} = \frac{dN(D_{\rm RI})}{\log(D_{\rm high,RI}) - \log(D_{\rm low,RI})}$$
(2) 40

[revised manuscript text omitted]